# Sample Efficient Robust Offline Self-Play for Model-based Reinforcement Learning

## Abstract

Multi-agent reinforcement learning (MARL), as a thriving field, explores how multiple agents independently make decisions in a shared dynamic environment. Due to environmental uncertainties, policies in MARL must remain robust to tackle the sim-to-real gap. Although robust RL has been extensively explored in single-agent settings, it has seldom received attention in self-play, where strategic interactions heighten uncertainties. We focus on robust two-player zero-sum Markov games (TZMGs) in offline RL, specifically on tabular robust TZMGs (RTZMGs) with a given uncertainty set. To address sample scarcity, we introduce a model-based algorithm (*RTZ-VI-LCB*) for RTZMGs, which integrates robust value iteration considering uncertainty level and applies a data-driven penalty to the robust value estimates. We establish the finite-sample complexity of RTZ-VI-LCB by accounting for distribution shifts in the historical dataset. Our algorithm is capable of learning under partial coverage and environmental uncertainty. An information-theoretic lower bound is developed to show that learning RTZMGs is at least as difficult as standard TZMGs when the uncertainty level is sufficiently small. This confirms the tightness of our algorithm's sample complexity, which is optimal regarding both state and action spaces. To the best of our knowledge, our algorithm is the first to attain this optimality and establishes a new benchmark for offline RTZMGs. We also extend our algorithm to multi-agent general-sum Markov games, achieving a breakthrough in breaking the curse of multiagency.

## 1 Introduction

Multi-agent reinforcement learning (MARL), which aims to develop algorithms for multiple agents to learn and make decisions in dynamic environments, has gained significant attention in areas such as game playing (Silver et al., 2017), autonomous driving (Bhalla et al., 2020), and Path Planning (Cao et al., 2020). Under the constraints on time or resources, a key challenge in applying MARL to real-world scenarios is the restricted ability to interact or explore the environment. Offline MARL, also named as batch MARL, addresses this issue by utilizing historical data collected from past interactions, often generated by unknown behavior policies. Researchers hope that this data can offer valuable insights into the optimal policy without the need for further exploration (Lambert et al., 2022). Beyond seeking to maximize the expected total rewards, a critical challenge lies in addressing environmental uncertainties stemming from model mismatches, system noise, and the gap between simulation and real-world situations. Standard MARL algorithms that train in ideal conditions are highly sensitive and prone to catastrophic failure when faced with even small adversarial perturbations in the deployment environment (Zhang et al., 2020; Yeh et al., 2021; Zeng et al., 2022). However, historical data is often gathered under the assumption of model stability, which is unrealistic due to the time-varying and non-stationary nature of real-world systems. Thus, the robust guarantee is critical in offline settings, leading to the formulation of offline robust MARL.

As a specific setting of MARL, two-player zero-sum Markov games (TZMGs) are a fascinating area of research, thus leading the field of robust TZMGs (RTZMGs) following from robust MARL. The inherent solution concepts for RTZMGs encompass equilibria not just between the two players but also between their adversaries, who select the worst-case environments from a predefined uncertainty set for each player. This structure inherently offers greater robustness and stability when facing unmodeled disruptions. Despite recent efforts (Kardeş et al., 2011; Blanchet et al., 2024; Zhang et al., 2020; Ma et al., 2023), there is still a lack of fundamental understanding in learning for

RTZMGs. For a tabular RTZMG with horizon length $H$, states $S$, actions $\{A, B\}$, and uncertainty sizes $\{\sigma^+, \sigma^-\}$ for the two players, the best sample complexity for offline setting so far is achieved by $\text{P}^2\text{M}^2\text{PO}$ (Blanchet et al., 2024) with a near-optimal sample complexity on $H$, $S$, $\{A, B\}$, where however the influence of uncertainty levels is overlooked. Notably, historical data often only offers partial and limited coverage of the state-action space, leading to poor estimates of model parameters and, in turn, unreliable policy learning outcomes. We summarize previous works and present them along with our results in Table 1. Consequently, current solutions lack an algorithm with optimal sample complexity under partial coverage. Thus, we explore the unresolved question as follows:

*Can we achieve effective sample complexity with robustness to learn Nash policy under partial and limited coverage in TZMGs simultaneously?*

Table 1: A comparison between RTZ-VI-LCB and $\text{P}^2\text{M}^2\text{PO}$ (Blanchet et al., 2024) on finding an $\varepsilon$-optimal robust Nash policy in finite-horizon offline RTZMGs with $f(\sigma^+, \sigma^-, H) = \min\left\{\frac{(H\sigma^+ - 1 + (1-\sigma^+)^H)}{(\sigma^+)^2}, \frac{(H\sigma^- - 1 + (1-\sigma^-)^H)}{(\sigma^-)^2}, H\right\}$, where the uncertainty set is quantified by total variation (TV) distance. The sample complexities omit all logarithmic factors.

| Algorithm | Sample complexity | Uncertainty level |
|---|---|---|
| $\text{P}^2\text{M}^2\text{PO}$ | $\frac{C_{\text{r}} H^5 S^2 AB}{\varepsilon^2}$ | not consider |
| **RTZ-VI-LCB (Ours)** | $\frac{C_{\text{r}}^\star H^4 S(A+B)}{\varepsilon^2} f(\sigma^+, \sigma^-, H)$ | full range |
| Lower bound | $\frac{C_{\text{r}}^\star S H^4 (A+B)}{\varepsilon^2}$ | $\min\{\sigma^+, \sigma^-\} \lesssim \frac{1}{H}$ |
| Lower bound | $\frac{C_{\text{r}}^\star S H^3 (A+B)}{\varepsilon^2 \min\{\sigma^+, \sigma^-\}}$ | $\min\{\sigma^+, \sigma^-\} \gtrsim \frac{1}{H}$ |

## 1.1 CONTRIBUTION

We aim to understand and achieve effective sample complexity under partial convergence in RTZMGs. Our contributions are outlined as follows.

- We introduce a concept to evaluate the quality of historical data, which is the robust unilateral clipped concentrability coefficient $C_{\text{r}}^\star \in \left[\frac{1}{S(A+B)}, \infty\right)$. This coefficient captures the distribution shift between the behavior policy $(\mu^{\text{n}}, \nu^{\text{n}})$ and the single optimal robust policies $(\mu, \nu^\star)$ and $(\mu^\star, \nu)$ under model perturbations, without requiring full coverage of the state-action space by the behavior policy. In contrast, $\text{P}^2\text{M}^2\text{PO}$ (Blanchet et al., 2024) measures distribution mismatch using the maximum density ratio $C_{\text{r}}$, which is less tight than our robust unilateral clipped concentrability coefficient $C_{\text{r}}^\star$.

- We design a new model-based algorithm for offline RTZMGs, an optimistic variant of robust value iteration (VI) for RTZMGs named RTZ-VI-LCB. Specifically, RTZ-VI-LCB incorporates a plug-in estimator of the nominal transition kernel (Iyengar, 2005) and introduces a data-informed penalty to the robust value estimates. armed with TV distance, we show that this algorithm achieves an $\varepsilon$-optimal robust Nash equilibrium (NE) policy up to some logarithmic factor as long as the sample size exceeds $\widetilde{O}\left(\frac{C_{\text{r}}^\star H^4 S(A+B)}{\varepsilon^2} \min\left\{\frac{(H\sigma^+ - 1 + (1-\sigma^+)^H)}{(\sigma^+)^2}, \frac{(H\sigma^- - 1 + (1-\sigma^-)^H)}{(\sigma^-)^2}, H\right\}\right)$ after a burn-in cost independent of $\varepsilon$. To the best of our knowledge, this is the first time optimal dependency on state $S$ and actions $\{A, B\}$ has been achieved for offline RTZMGs.

- In addition to the upper bound, we derive information-theoretic lower bounds across various uncertainty levels, independent of the specific distance metric applied. We show that there exists an algorithm requiring at least $\Omega\left(\frac{C_{\text{r}}^\star S H^4 (A+B)}{\varepsilon^2}\right)$ samples to find an $\varepsilon$-optimal robust NE policy when the uncertainty level $\min\{\sigma^+, \sigma^-\} \lesssim \frac{1}{H}$, and at least $\Omega\left(\frac{C_{\text{r}}^\star S H^3 (A+B)}{\varepsilon^2 \min\{\sigma^+, \sigma^-\}}\right)$ samples when $\min\{\sigma^+, \sigma^-\} \gtrsim \frac{1}{H}$. This indicates that learning RTZMGs is at least as challenging as standard TZMGs (Jin et al., 2022) when the uncertainty is sufficiently small. Besides, we confirm the optimality of RTZ-VI-LCB across different uncertainty levels of the critical parameters, i.e., state $S$ and actions $\{A, B\}$, except for the finite-horizon $H$.

- We design an extended algorithm of RTZ-VI-LCB for robust multi-player general-sum Markov games (named Multi-RTZ-VI-LCB) and achieve an $\varepsilon$-optimal robust NE policy in $\widetilde{O}\left(\frac{C_r^\star H^4 S \sum_{i=1}^m A_i}{\varepsilon^2} \min\left\{\left\{\frac{(H\sigma_i - 1 + (1-\sigma_i)^H)}{(\sigma_i)^2}\right\}_{i=1}^m, H\right\}\right)$ samples with $M$ players and $A_i$ actions and uncertainty size $\sigma_i$ per player.

## 1.2 RELATED WORK

In this section, we review a curated selection of related research, with an emphasis on provably efficient RL algorithms in the tabular setting, as these are the most pertinent to our work.

**Finite-sample studies of standard TZMGs.** Markov games (MGs), or called stochastic games, were first proposed in the early 1950s (Shapley, 1953). Since then, extensive research has been conducted, and MARL has gained significant attention (Oroojlooy & Hajinezhad, 2023), particularly around Nash equilibrium (Littman, 1994; Lee et al., 2020). Numerous MARL algorithms with provable convergence and asymptotic guarantees have been developed (Rashid et al., 2020). More recent work has focused on creating algorithms for standard MARL with non-asymptotic guarantees through finite-sample analysis. In this area, most efforts to compute Nash equilibria are focused on TZMGs. The studies in (Bai & Jin, 2020) and (Xie et al., Jun. 2022) were the first to provide non-asymptotic sample complexity guarantees for model-based (e.g., VI-Explore and VI-ULCB) and model-free algorithms (e.g., OMNI-VI). Further improvements in sample complexity have been explored (Cui et al., 2023; Chen et al., 2022; Liu et al., July 2021; Feng et al., 2023; Li et al., 2024c).

**Robustness in MARL.** Although progress has been made in standard MARL, existing algorithms may struggle when faced with environmental disturbances or uncertainties, leading to significantly deviated equilibria. Increasing research now focuses on enhancing MARL robustness against uncertainties in different parts of MGs (Vial et al., 2022), including state (Zhou & Liu, 2023), environment (reward and transition dynamics), agent types (Zhang et al., 2021), and other agents' policies (Kannan et al., 2023). A typical method to address robustness against uncertainties of the environment is distributionally robust optimization (DRO), which is a method predominantly explored in supervised learning (Bertsimas et al., 2018; Gao, 2023; Blanchet & Murthy, 2019). The application of DRO to manage model uncertainty in single-agent RL (Iyengar, 2005) has attracted considerable attention. However, when extended to MARL, researchers formulated the problem as robust MGs armed with DRO and developed a relatively understudied field with only a few proven algorithms (Blanchet et al., 2024; Kardeş et al., 2011; Ma et al., 2023; Zhang et al., 2020; Shi et al., 2024b). Thus, relevant algorithms based on partial coverage of datasets while considering the uncertainty level are lacking.

**Single-agent robust RL.** In single-agent RL, addressing uncertainties of environments using DRO—such as robust Markov decision processes (MDPs) and distributionally robust dynamic programming—has attracted considerable interest in both theoretical research and practical applications (Badrinath & Kalathil, 2021; Goyal & Grand-Clement, 2023). Recent work has focused on the finite-sample performance of provable robust RL algorithms, exploring different divergence functions for uncertainty sets, various sampling mechanisms, and related challenges (Yang et al., 2023; Blanchet et al., 2024; Shi et al., 2024a). Studies on robust MDPs, particularly relevant here, use uncertainty sets based on TV distance (Liu & Xu, 2024) or Kullback-Leibler (KL) divergence (Shi & Chi, 2024) in tabular settings. It has been shown that addressing robust MDPs does not demand more samples compared with those needed for standard MDPs (Shi et al., 2024a). However, RTZMGs present additional complexities beyond those in robust single-agent RL.

## 2 PROBLEM FORMULATION

We focus on offline RTZMGs in this paper, which is a robust version of standard TZMGs taking environmental uncertainties into consideration. RTZMGs form a broader class than standard TZMGs, accommodating various prescribed environmental uncertainty sets. Along with this setting, we investigate an efficient algorithm to achieve robustness and optimal sample complexity on action $\{A, B\}$ without requiring full coverage of the state-action space. An RTZMG under the finite-horizon setting can be defined as $\mathcal{MG}_r = \left\{\mathcal{S}, \mathcal{A}, \mathcal{B}, \mathcal{U}_\rho^{\sigma^+}\left(P^0\right), \mathcal{U}_\rho^{\sigma^-}\left(P^0\right), r, H\right\}$, where

$\mathcal{S} := \{1, \cdots, S\}$ is the state space of size $S$; ($\mathcal{A} := \{1, \cdots, A\}, \mathcal{B} := \{1, \cdots, B\}$) denotes the action spaces of the max-player and the min-player with sizes $A$ and $B$, respectively; $H$ is the horizon length; $r = \{r_h\}_{h=1}^H$ represents the immediate reward obtained at time step $h$. Specifically, $r_h(s, a, b)$ is assumed to be deterministic on a state-action pair $(s, a, b)$ and falls within the range $[0, 1]$. In RTZMGs, this reward can represent both the gain of the max-player and the loss of the min-player. A crucial difference from standard TZMGs is that, rather than assuming a fixed transition kernel, both players in RTZMGs expect that the transition kernel could be chosen arbitrarily from specified uncertainty sets, $\mathcal{U}_\rho^{\sigma^+}(P^0)$ and $\mathcal{U}_\rho^{\sigma^-}(P^0)$, respectively. These uncertainty sets are centered on a nominal kernel $P^0 : \mathcal{S} \times \mathcal{A} \times \mathcal{B} \mapsto \Delta(\mathcal{S})$, with their size and shape defined by a distance metric $\rho$ and radius parameters $\sigma^+ > 0$ and $\sigma^- > 0$. To accommodate individual robustness preferences, the max-player and min-player can independently define their uncertainty sets $\mathcal{U}_\rho^{\sigma^+}(P^0)$ and $\mathcal{U}_\rho^{\sigma^-}(P^0)$, selecting different sizes ($\sigma^+ > 0$ and $\sigma^- > 0$) and potentially different divergence functions ($\rho$) for shaping the sets. In this paper, we consider the same divergence function for both players.

**Uncertainty set with *two-player-wise* $(s, a, b)$-*rectangularity*.** We define the transition kernel uncertainty sets $\mathcal{U}_\rho^{\sigma^+}(P^0)$ and $\mathcal{U}_\rho^{\sigma^-}(P^0)$ for RTZMGs. Inspired by the *rectangularity* condition used in robust single-agent RL (Shi et al., 2024a; Iyengar, 2005), we adapt this concept to a two-player setting, termed *two-player-wise* $(s, a, b)$-*rectangularity*. The adaptation enhances computational tractability and facilitates the robust version of Bellman recursions. It permits each player to select its uncertainty set independently, which can be decomposed for each state-action pair into a product of subsets. Consequently, the uncertainty sets $\mathcal{U}_\rho^{\sigma^+}(P^0)$ and $\mathcal{U}_\rho^{\sigma^-}(P^0)$ for the two players, adhering to *two-player-wise* $(s, a, b)$-*rectangularity*, are mathematically defined as:

$$\mathcal{U}_\rho^{\sigma^+}(P^0) := \otimes \, \mathcal{U}_\rho^{\sigma^+}(P^0_{h,s,a,b}), \qquad \mathcal{U}_\rho^{\sigma^-}(P^0) := \otimes \, \mathcal{U}_\rho^{\sigma^-}(P^0_{h,s,a,b}), \tag{1}$$

where

$$\mathcal{U}_\rho^{\sigma^+}(P^0_{h,s,a,b}) := \left\{ P_{h,s,a,b} \in \Delta(\mathcal{S}) : \rho\left(P_{h,s,a,b}, P^0_{h,s,a,b}\right) \le \sigma^+ \right\}.$$

Here, $\otimes$ represents the Cartesian product. The uncertainty set for min-player can be defined similarly. We define a vector of the transition kernel $P$ or $P^0$ at any state-action pair $(s, a, b)$ as

$$P_{h,s,a,b} := P_h(\cdot \mid s, a, b) \in \mathbb{R}^{1 \times S}, \qquad P^0_{h,s,a,b} := P^0_h(\cdot \mid s, a, b) \in \mathbb{R}^{1 \times S}. \tag{2}$$

Here, the distance function $\rho$ for each player's uncertainty set can be selected from various options that quantify differences between probability vectors. These include $f$-divergences (such as KL divergence, TV distance, and chi-square) (Yang et al., 2022), the Wasserstein distance (Xu et al., 2023), and $\ell_q$ norms (Clavier et al., 2023).

**Robust value functions.** In RTZMGs, players seek to optimize their worst-case performance across all possible transition kernels within their respective uncertainty sets $\mathcal{U}_\rho^{\sigma^+}(P^0)$ and $\mathcal{U}_\rho^{\sigma^-}(P^0)$. For any product policy $(\mu \times \nu) \in \Delta(\mathcal{A} \times \mathcal{B})$, the max-player's worst-case performance at time step $h$ is quantified by the *robust value function* $V_h^{\mu,\nu,\sigma^+}$ and the *robust Q-function* $Q_h^{\mu,\nu,\sigma^+}$ for all $(h, s, a, b) \in [H] \times \mathcal{S} \times \mathcal{A} \times \mathcal{B}$, defined as:

$$V_h^{\mu,\nu,\sigma^+}(s) := \inf_{P \in \mathcal{U}_\rho^{\sigma^+}(P^0)} V_h^{\mu,\nu,P}(s) \quad \text{and} \quad Q_h^{\mu,\nu,\sigma^+}(s, a, b) := \inf_{P \in \mathcal{U}_\rho^{\sigma^+}(P^0)} Q_h^{\mu,\nu,P}; \tag{3}$$

$$V_h^{\mu,\nu,\sigma^-}(s) := \sup_{P \in \mathcal{U}_\rho^{\sigma^-}(P^0)} V_h^{\mu,\nu,P}(s) \quad \text{and} \quad Q_h^{\mu,\nu,\sigma^-}(s, a, b) := \sup_{P \in \mathcal{U}_\rho^{\sigma^-}(P^0)} Q_h^{\mu,\nu,P}, \tag{4}$$

where

$$V_h^{\mu,\nu,P}(s) := \mathbb{E}_{\mu,\nu,P}\left[ \sum_{t=h}^H r_t(s_t, a_t, b_t) \mid s_h = s \right];$$

$$Q_h^{\mu,\nu,P}(s, a, b) := \mathbb{E}_{\mu,\nu,P}\left[ \sum_{t=h}^H r_t(s_t, a_t, b_t) \mid s_h = s, a_h = a, b_h = b \right].$$

**Offline dataset.** Let $\mathcal{D}$ be a dataset consisting of $K$ episodes under independence, with each episode produced by implementing a behavior policy $\{\mu_h^{\mathsf{n}}, \nu_h^{\mathsf{n}}\}_{h=1}^H$ in a nominal MDP $\mathcal{M}^0 = \left(\mathcal{S}, \mathcal{A}, \mathcal{B}, H, P^0 := \{P_h^0\}_{h=1}^H, \{r_h\}_{h=1}^H\right)$. For $1 \leq k \leq K$, the $k$-th episode $\left(s_1^k, a_1^k, b_1^k, \ldots, s_H^k, a_H^k, b_H^k, s_{H+1}^k\right)$ is generated as follows:

$$s_1^k \sim \varrho^{\mathsf{n}}, \quad a_h^k \sim \mu_h^{\mathsf{n}}(\cdot \,|\, s_h^k), \quad b_h^k \sim \nu_h^{\mathsf{n}}(\cdot \,|\, s_h^k), \quad s_{h+1}^k \sim P_h^0(\cdot \,|\, s_h^k, a_h^k, b_h^k), \quad 1 \leq h \leq H. \quad (5)$$

Throughout this paper, let $\varrho^{\mathsf{n}}$ denote the initial distribution related to a historical dataset. We use the short-hand notation for the occupancy distribution w.r.t. the behavior policy $(\mu^{\mathsf{n}}, \nu^{\mathsf{n}})$ as: $\forall (h, s, a, b) \in [H] \times \mathcal{S} \times \mathcal{A} \times \mathcal{B}$,

$$d_h^{\mathsf{n}, P^0}(s) = d_h^{\mu^{\mathsf{n}}, \nu^{\mathsf{n}}, P^0}(s) := \mathbb{P}(s_h = s \,|\, s_1 \sim \varrho^{\mathsf{n}}, \mu^{\mathsf{n}}, \nu^{\mathsf{n}}, P^0); \quad (6a)$$

$$d_h^{\mathsf{n}, P^0}(s, a, b) = d_h^{\mu^{\mathsf{n}}, \nu^{\mathsf{n}}, P^0}(s, a, b) := \mathbb{P}(s_h = s \,|\, s_1 \sim \varrho^{\mathsf{n}}, \mu^{\mathsf{n}}, \nu^{\mathsf{n}}, P^0) \, \mu_h^{\mathsf{n}}(a \,|\, s) \, \nu_h^{\mathsf{n}}(b \,|\, s). \quad (6b)$$

Similarly, for any product policy $(\mu, \nu)$, there is, $\forall (h, s, a, b) \in [H] \times \mathcal{S} \times \mathcal{A} \times \mathcal{B}$

$$d_h^{\mu, \nu, P}(s) := \mathbb{P}(s_h = s \,|\, s_1 \sim \varrho, \mu, \nu, P); \quad (7a)$$

$$d_h^{\mu, \nu, P}(s, a, b) := \mathbb{P}(s_h = s \,|\, s_1 \sim \varrho, \mu, \nu, P) \, \mu_h(a \,|\, s) \, \nu_h(b \,|\, s). \quad (7b)$$

**Robust Bellman equations.** RTZMGs include a robust version of the Bellman equation, referred to as the *robust Bellman equation*. The robust value functions $V_h^{\mu, \nu, \sigma^+}(s)$ for max-player in RTZMGs, associated with any product policy $(\mu, \nu)$, satisfy: $\forall (h, s) \in [H] \times \mathcal{S}$,

$$V_h^{\mu, \nu, \sigma^+}(s) = \mathbb{E}_{a \sim \mu_h(a), b \sim \nu_h(a)} \left[ r_h(s, a, b) + \inf_{P \in \mathcal{U}_\rho^{\sigma^+}(P_{h,s,a,b}^0)} P V_{h+1}^{\mu, \nu, \sigma^+} \right]. \quad (8)$$

$V_h^{\mu, \nu, \sigma^-}(s)$ for min-player can be obtained similarly. We highlight that the robust Bellman equations are intrinsically connected to the *two-player-wise $(s, a, b)$-rectangularity* condition (see (1)) applied to the uncertainty set. This condition separates the dependencies of uncertainty subsets among different time steps, the players, and state-action pairs, thus leading to the Bellman recursion.

**Optimal robust policy.** We further define the maximum robust value function with fixed opponent policy for each player as: $\forall (h, s) \in [H] \times \mathcal{S}$,

$$V_h^{\star, \nu, \sigma^+}(s) := \max_{\mu : \mathcal{S} \times [H] \mapsto \Delta(\mathcal{A})} V_h^{\mu, \nu, \sigma^+}(s) = \max_{\mu : \mathcal{S} \times [H] \mapsto \Delta(\mathcal{A})} \inf_{P \in \mathcal{U}_\rho^{\sigma^+}(P^0)} V_h^{\mu, \nu, P}(s). \quad (9)$$

Optimal robust policy for min-player can be obtained similarly. As proved by Blanchet et al. (2024), there is at least one policy referred to as $\mu_h^\star(s) : \mathcal{S} \times [H] \mapsto \Delta(\mathcal{A})$ (for the max-player) and $\nu_h^\star(s) : \mathcal{S} \times [H] \mapsto \Delta(\mathcal{B})$ (for the min-player), corresponding to as the *robust best-response policy*. These policies can simultaneously achieve $V_h^{\star, \nu, \sigma^+}(s)$ (for the max-player) and $V_h^{\mu, \star, \sigma^-}(s)$ (for the min-player) for all $s \in \mathcal{S}$ and $h \in [H]$.

**Robust Nash equilibrium.** In RTZMGs, the dynamics expand beyond traditional TZMGs to involve four participants: two players and two adversaries determining the worst-case transitions. Therefore, finding an equilibrium becomes central in RTZMGs due to potentially conflicting objectives. We introduce the robust variant of standard solution concepts—robust NE for RTZMGs. A product policy $(\mu, \nu)$ is considered a *robust NE* if

$$\forall (s) \in \mathcal{S}, \quad V_h^{\star, \nu, \sigma^+}(s) = V_h^{\star, \sigma^+}(s); \quad V_h^{\mu, \star, \sigma^-}(s) = V_h^{\star, \sigma^-}(s). \quad (10)$$

A robust NE signifies that given the product policy $(\mu, \nu)$ of the opponents, no player can enhance their outcome by deviating from their current policy unilaterally when each player accounts for the worst-case scenario within their uncertainty set $\mathcal{U}_\rho^{\sigma^+}(P^0)$ or $\mathcal{U}_\rho^{\sigma^-}(P^0)$.

Since finding exact robust equilibria can be complex and may not always be feasible, practitioners often seek approximate equilibria. In this context, a product policy $(\mu \times \nu) \in \Delta(\mathcal{A} \times \mathcal{B})$ can be termed an $\varepsilon$-*robust NE* if

$$\text{Gap}(\mu, \nu) := \max \left\{ V_1^{\star, \nu, \sigma^+}(\varrho) - V_1^{\star, \sigma^+}(\varrho), \ V_1^{\star, \sigma^-}(\varrho) - V_1^{\mu, \star, \sigma^-}(\varrho) \right\} \leq \varepsilon, \quad (11)$$

where

$$V_1^{\star,\nu,\sigma^+}(\varrho) = \mathbb{E}_{s\sim\varrho} V_1^{\star,\nu,\sigma^+}(s), \qquad \text{and} \qquad V_1^{\star,\sigma^+}(\varrho) = \mathbb{E}_{s\sim\varrho} V_1^{\star,\sigma^+}(s).$$

The definitions of $V_1^{\mu,\star,\sigma^-}(\varrho)$ and $V_1^{\star,\sigma^-}(\varrho)$ can be obtained similarly. The existence of robust NE has been proved for general divergence functions in the uncertainty set by Blanchet et al. (2024).

**Learning objective**    With a dataset collected from the nominal environment, our objective is to find a solution among the $\varepsilon$-robust NEs for the RTZMG $\mathcal{MG}_r$ with respect to a specified uncertainty set $\mathcal{U}(P^0)$ around the nominal kernel, while minimizing the number of samples required under partial coverage of the state-action space.

## 3    ALGORITHM DESIGN

In this section, we propose an efficient model-based algorithm RTZ-VI-LCB to achieve robustness and optimal sample complexity on action $\{A, B\}$. This algorithm is designed for offline RTZMGs within the finite-horizon setting.

### 3.1    BUILDING AN EMPIRICAL NOMINAL MDP

According to the empirical frequencies of state transitions, we can naturally construct an empirical estimate $\widehat{P}^0 = \{\widehat{P}_h^0\}_{h=1}^H$ of $P^0$, where

$$\widehat{P}_h^0\left(s' \mid s, a, b\right) = \begin{cases} \frac{1}{N_h(s,a,b)} \sum_{i=1}^N \mathbb{1}\left\{(s_i, a_i, b_i, s_i') = (s, a, b, s')\right\}, & \text{if } N_h\left(s, a, b\right) > 0; \\ \frac{1}{S}, & \text{if } N_h\left(s, a, b\right) = 0, \end{cases} \quad (12)$$

$$\widehat{r}_h\left(s, a, b\right) = \begin{cases} r_h\left(s, a, b\right), & \text{if } N_h\left(s, a, b\right) > 0; \\ 0, & \text{if } N_h\left(s, a, b\right) = 0, \end{cases} \quad (13)$$

for any $(h, s, a, b, s') \in [H] \times \mathcal{S} \times \mathcal{A} \times \mathcal{B} \times \mathcal{S}$. Besides, $N_h(s, a, b)$ represents the total number of sample transitions from $(s, a, b)$ at step $h$, and

$$N_h(s, a, b) := \sum_{i=1}^N \mathbb{1}\left\{(s_i, a_i, b_i) = (s, a, b)\right\}. \quad (14)$$

---

**Algorithm 1:** Two-stage subsampling technique for RTZ-VI-LCB.

1 **Input:** Dataset $\mathcal{D}$, probability $\delta$.
2 **Step 1: Data Partitioning.** Split $\mathcal{D}$ into two equal-sized subsets, $\mathcal{D}^\mathsf{m}$ and $\mathcal{D}^\mathsf{a}$, each containing $K/2$ trajectories.
3 **Step 2: Defining Transition Bounds.** For step $h$ and state $s$, denote the number of transitions from $\mathcal{D}^\mathsf{m}$ (resp. $\mathcal{D}^\mathsf{a}$) as $N_h^\mathsf{m}(s)$ (resp. $N_h^\mathsf{a}(s)$). Construct the trimmed count as:

$$N_h^\mathsf{t}(s) := \max\left\{ N_h^\mathsf{a}(s) - 10\sqrt{N_h^\mathsf{a}(s)\log\frac{HS}{\delta}},\, 0 \right\}; \quad (15)$$

4 **Step 3: Generating Subsampled Dataset.** Randomly sample transitions (quadruples of the form $(s, a, b, h, s')$) from $\mathcal{D}^\mathsf{m}$ uniformly. For each $(s, h) \in \mathcal{S} \times [H]$, include $\min\{N_h^\mathsf{t}(s), N_h^\mathsf{m}(s)\}$ transitions in the new dataset $\mathcal{D}^\mathsf{t}$.
5 **Output:** Set $\mathcal{D}_0 = \mathcal{D}^\mathsf{t}$.

---

Although it is feasible to decompose the historical dataset $\mathcal{D}$ into sample transitions, the dependencies between transitions within the same episode introduce complexities in our analysis. To address this issue, Li et al. (2024a) introduced a two-fold subsampling method for single-agent RL to preprocess $\mathcal{D}$, thereby reducing statistical dependencies and producing a distributionally equivalent dataset $\mathcal{D}_0$ with independent samples. We adapt this method to TZMGs, as outlined in Algorithm 1. We present the following lemma concerning the dataset $\mathcal{D}_0$, which is proved in Appendix C.1.

**Lemma 1** *The dataset produced by the two-stage subsampling method is distributionally identical to $\mathcal{D}_0$ with probability at least $1 - 8\delta$, where $\{N_h(s, a, b)\}$ are independent of the sample transitions in $\mathcal{D}^0$ and obey: $\forall (h, s, a, b) \in [H] \times \mathcal{S} \times \mathcal{A} \times \mathcal{B}$,*

$$N_h(s, a, b) \geq \frac{K d_h^{\mathsf{n}}(s, a, b)}{8} - 5\sqrt{K d_h^{\mathsf{n}}(s, a, b) \log \frac{KH}{\delta}}. \tag{16}$$

By applying the two-fold sampling method, we can treat the dataset $\mathcal{D}_0$ as having independent samples, simplifying the analysis significantly as supported by Lemma 1.

## 3.2 AN OPTIMISTIC VARIANT OF ROBUST VI WITH LOWER CONFIDENCE BOUNDS.

We propose a model-based approach for solving RTZMGs using an approximate $\widehat{P}^0$ for $P^0$, which is the nominal transition kernel. Specifically, we introduce VI with lower confidence bounds (LCBs) for RTZMGs (RTZ-VI-LCB) to compute a robust NE for two players, as summarized in Algorithm 2.

Our algorithm begins at the final time step $h = H$ and proceeds backward through $h = H - 1, H - 2, \ldots, 1$. Drawing from the principle of pessimism in single-agent offline RL (Li et al., 2024a; Jin et al., 2021), we design an optimistic robust Q-value to estimate the robust Q-function at time step $h \in [H]$ as $\widehat{Q}_h^+$ and $\widehat{Q}_h^-$ for all $(h, s, a, b) \in [H] \times \mathcal{S} \times \mathcal{A} \times \mathcal{B}$, that is,

$$\widehat{Q}_h^+ (s, a, b) = \widehat{r}_h (s, a, b) + \inf_{P \in \mathcal{U}^{\sigma^+} \left( \widehat{P}_{h,s,a,b}^0 \right)} P \widehat{V}_{h+1}^+ + \beta_h \left( s, a, b, \widehat{V}_{h+1}^+ \right); \tag{17a}$$

$$\widehat{Q}_h^- (s, a, b) = \widehat{r}_h (s, a, b) + \sup_{P \in \mathcal{U}^{\sigma^-} \left( \widehat{P}_{h,s,a,b}^0 \right)} P \widehat{V}_{h+1}^- - \beta_h \left( s, a, b, \widehat{V}_{h+1}^- \right). \tag{17b}$$

**Dual problem.** Solving (17) directly is computationally intensive because it requires optimizing over an $S$-dimensional probability simplex, which becomes exponentially more difficult as the state space size $S$ increases. In fortunate, strong duality for TV distance allows us to tackle this problem by solving its dual (Iyengar, 2005):

$$\inf_{P \in \mathcal{U}^{\sigma^+} \left( \widehat{P}_{h,s,a,b}^0 \right)} P \widehat{V}_{h+1}^+ = \max_{\alpha \in [\min_s \widehat{V}_{h+1}^+, \max_s \widehat{V}_{h+1}^+]} \left\{ \widehat{P}_{h,s,a,b}^0 \left[ \widehat{V}_{h+1}^+ \right]_\alpha - \sigma^+ \left( \alpha - \min_{s'} \left[ \widehat{V}_{h+1}^+ \right]_\alpha (s') \right) \right\}. \tag{18}$$

where $\left[ \widehat{V}_{h+1}^+ \right]_\alpha$ denotes the clipped versions of $\widehat{V}_{h+1}^- \in \mathbb{R}^S$ and $\widehat{V}_{h+1}^+ \in \mathbb{R}^S$ based on some level $\alpha \geq 0$, as follows. $\sup_{P \in \mathcal{U}^{\sigma^-} \left( \widehat{P}_{h,s,a,b}^0 \right)} P \widehat{V}_{h+1}^-$ can be defined similarly. See Appendix A for details.

$$\left[ \widehat{V}_{h+1}^+ \right]_\alpha (s) := \begin{cases} \widehat{V}_{h+1}^+(s), & \text{if } \widehat{V}_{h+1}^+(s) > \alpha; \\ \alpha, & \text{otherwise}; \end{cases} \tag{19}$$

**Penalty term.** The optimistic robust $Q$-function estimate is refined by $\beta_h(s, a, b, \widehat{V})$, which is a data-driven penalty term and includes the uncertainty in value estimates. We adopt the Bernstein-style penalty to better capture the variance structure over time. In particular, for any $(s, a, b, h) \in \mathcal{S} \times \mathcal{A} \times \mathcal{B} \times [H]$ and $\delta \in (0, 1)$, the penalty term $\beta_h(s, a, b, \widehat{V})$ is defined as:

$$\beta_h \left( s, a, b, \widehat{V} \right) = \min \left\{ \max \left\{ \sqrt{\frac{C_{\mathsf{n}} \log \frac{KH}{\delta}}{N_h (s, a, b)} \mathsf{Var}_{\widehat{P}_{h,s,a,b}^0} (\widehat{V})}, \frac{2 C_{\mathsf{n}} H \log \frac{KH}{\delta}}{N_h (s, a, b)} \right\}, H \right\}, \tag{20}$$

where $C_{\mathsf{n}}$ is some universal constant, and

$$\mathsf{Var}_{\widehat{P}_{h,s,a,b}^0} \left( \widehat{V} \right) := \widehat{P}_{h,s,a,b}^0 \widehat{V}^2 - (\widehat{P}_{h,s,a,b}^0 \widehat{V})^2. \tag{21}$$

Note that we choose $\widehat{P}^0$, as opposed to $P^0$ (i.e., $\mathsf{Var}_{\widehat{P}_{h,s,a,b}^0} (\widehat{V})$) in the variance term, since we have no access to the true transition kernel $P^0$. This penalty term is distinct from those used in standard offline TZMGs (Cui et al., 2023; Li et al., 2024a), as it accounts for the unique structure of robust self-play MDPs. Specifically, it provides a tight upper bound on statistical uncertainty, considering the non-linear and implicit dependency introduced by the uncertainty set $\mathcal{U}(P^0)$, addressing challenges not present in standard MDP scenarios.

**Policy estimation.** We update the policies using the estimated $Q$-functions with uncertainty as line 6 in Algorithm 2. Specifically, for any matrix $\mathbf{N} \in \mathbb{R}^{A \times B}$, the function $\mathsf{ComputNash}(\mathbf{N})$ returns a solution $(\widehat{w}, \widehat{z})$ to the minimax problem $\max_{w \in \Delta(\mathcal{A})} \min_{z \in \Delta(\mathcal{B})} w^\top \mathbf{N} z$. In other words, for each $s \in \mathcal{S}$, we compute the NE policies $\left(\mu_h^+(s), \nu_h^+(s)\right)$ and $\left(\mu_h^-(s), \nu_h^-(s)\right) \in \Delta(\mathcal{A}) \times \Delta(\mathcal{B})$ for the zero-sum matrix games with payoff matrices $\widehat{Q}_h^+(s, \cdot, \cdot)$ and $\widehat{Q}_h^-(s, \cdot, \cdot)$, respectively. Solving these robust matrix games is generally PPAD-hard due to the potential for players to choose different worst-case transition kernels.

---

**Algorithm 2:** Value iteration with lower confidence bounds for RTZMGs (RTZ-VI-LCB).

---

**1** **Initialization**: Set uncertainty levels $\sigma^-$ and $\sigma^+$; set $\widehat{V}_h^-(s) = 0$ and $\widehat{V}_h^+(s) = H$ for all $(s, h) \in \mathcal{S} \times [H+1]$; set $\widehat{Q}_h^-(s, a, b) = 0$ and $\widehat{Q}_h^+(s, a, b) = H$ for all $(s, a, b, h) \in \mathcal{S} \times \mathcal{A} \times \mathcal{B} \times [H+1]$.

**2** **Compute** the empirical reward function $\widehat{r}$ using (13) and the empirical transition kernel $\widehat{P}_0$ using (12).

**3** **for** $h = H, H-1, \ldots, 1$ **do**

**4**      **Update** the robust Q-value estimate as

$$\widehat{Q}_h^+(s, a, b) = \min\left\{\widehat{r}_h(s, a, b) + \inf_{P \in \mathcal{U}^{\sigma^+}\left(\widehat{P}_{h,s,a,b}^0\right)} P\widehat{V}_{h+1}^+ + \beta_h\left(s, a, b, \widehat{V}_{h+1}^+\right), H\right\};$$

$$\widehat{Q}_h^-(s, a, b) = \max\left\{\widehat{r}_h(s, a, b) + \sup_{P \in \mathcal{U}^{\sigma^-}\left(\widehat{P}_{h,s,a,b}^0\right)} P\widehat{V}_{h+1}^- - \beta_h\left(s, a, b, \widehat{V}_{h+1}^-\right), 0\right\},$$

with $\beta_h(s, a, b, V) = \min\left\{\max\left\{\sqrt{\frac{C_\mathsf{n} \log \frac{KH}{\delta}}{N_h(s,a,b)} \mathsf{Var}_{\widehat{P}_{h,s,a,b}^0}(V)}, \frac{2C_\mathsf{n} H \log \frac{KH}{\delta}}{N_h(s,a,b)}\right\}, H\right\}.$

**5**      **Compute** Nash policy for each $s \in \mathcal{S}$ as

$$\left(\mu_h^+(s), \nu_h^+(s)\right) = \mathsf{ComputNash}\left(\widehat{Q}_h^+(s, \cdot, \cdot)\right);$$

$$\left(\mu_h^-(s), \nu_h^-(s)\right) = \mathsf{ComputNash}\left(\widehat{Q}_h^-(s, \cdot, \cdot)\right),$$

**6**      **Update** the robust value estimate for each $s \in \mathcal{S}$ as

**7**      $\widehat{V}_h^-(s) = \mathbb{E}_{a \sim \mu_h^-(s), b \sim \nu_h^-(s)}\left[\widehat{Q}_h^-(s, a, b)\right], \qquad \widehat{V}_h^+(s) = \mathbb{E}_{a \sim \mu_h^+(s), b \sim \nu_h^+(s)}\left[\widehat{Q}_h^+(s, a, b)\right].$

**8** **Output**: The policy pair $(\widehat{\mu}, \widehat{\nu})$, where $\widehat{\mu} = \{\mu_h^-\}_{h=1}^H$ and $\widehat{\nu} = \{\nu_h^+\}_{h=1}^H$.

---

## 4 PERFORMANCE GUARANTEES

**Robust unilateral clipped concentrability.** To assess the effectiveness of the historical dataset for achieving the desired goal, it is essential to measure the distributional discrepancy between the historical data and the target data. Drawing on the *single-policy clipped concentrability* assumption in the single-agent RL (Li et al., 2024a), we propose a novel assumption for RTZMGs as:

**Assumption 1 (Robust unilateral clipped concentrability)** *The behavior policies of the historical dataset $\mathcal{D}$ satisfies*

$$\max\left\{\sup_{(\mu,s,a,b,h,P) \in \Delta(\mathcal{A}) \times \mathcal{S} \times \mathcal{A} \times \mathcal{B} \times [H] \times \mathcal{U}^{\sigma^-}(P^0)} \frac{\min\left\{d_h^{\mu, \nu^\star, P}(s, a, b), \frac{1}{S(A+B)}\right\}}{d_h^{\mathsf{n}, P^0}(s, a, b)},\right.$$

$$\left. \sup_{(\nu,s,a,b,h,P) \in \Delta(\mathcal{B}) \times \mathcal{S} \times \mathcal{A} \times \mathcal{B} \times [H] \times \mathcal{U}^{\sigma^+}(P^0)} \frac{\min\left\{d_h^{\mu^\star, \nu, P}(s, a, b), \frac{1}{S(A+B)}\right\}}{d_h^{\mathsf{n}, P^0}(s, a, b)}\right\} \leq C_\mathsf{r}^\star \quad (22)$$

for some quantity $C_r^\star \in \left[\frac{1}{S(A+B)}, \infty\right]$. *We define $C_r^\star$ as the smallest value that satisfies (22), referring to it as the robust unilateral clipped concentrability coefficient. For consistency, we adopt the convention $0/0 = 0$.*

Notably, if $d_h^{\mu,\nu^\star,P}(s,a,b)$ or $d_h^{\mu^\star,\nu,P}(s,a,b)$ is larger than $\frac{1}{S(A+B)}$, the robust unilateral clipped concentrability assumption above do not require the data distribution $d_h^{n,P^0}(s,a,b)$ to scale with $d_h^{\mu,\nu^\star,P}(s,a,b)$ or $d_h^{\mu^\star,\nu,P}(s,a,b)$ proportionally. We here outline the principal theoretical findings concerning the sample complexity of learning robust NE in RTZMGs, including an upper bound for the RTZ-VI-LCB algorithm (Algorithm 2) and an information-theoretic lower bound. Initially, we present the finite-sample guarantee for RTZ-VI-LCB, with detailed proof provided in Appendix B.

**Theorem 1 (Upper bound for RTZ-VI-LCB)** *Under the TV uncertainty set $\mathcal{U}^{\sigma^+}(\cdot)$ and $\mathcal{U}^{\sigma^-}(\cdot)$ defined in (2) with $\sigma^+$, $\sigma^- \in (0,1]$. Define $d_m^n = \min_{h,s,a,b}\{d_h^n(s,a,b) : d_h^n(s,a,b) > 0\}$. Define $f(\sigma^+, \sigma^-) = \min\left\{\frac{(H\sigma^+ - 1 + (1-\sigma^+)^H)}{(\sigma^+)^2}, \frac{(H\sigma^- - 1 + (1-\sigma^-)^H)}{(\sigma^-)^2}, H\right\}$. Consider any $\delta \in (0,1)$ and any RTZMG $\mathcal{MG}_r = \left\{\mathcal{S}, \mathcal{A}, \mathcal{B}, \mathcal{U}^{\sigma^+}(P^0), \mathcal{U}^{\sigma^-}(P^0), r, H\right\}$. For sufficient large constants $c_0, c_1 > 0$, with probability at least $1 - \delta$, we can achieve*

$$\mathrm{Gap}(\widehat{\mu}, \widehat{\nu}) \leq c_1 \sqrt{\frac{C_r^\star H^3 S(A+B)\log\frac{KH}{\delta}}{K} f(\sigma^+, \sigma^-, H)}, \tag{23}$$

*with the total number of samples $T$ exceeding*

$$T = KH \geq c_0 \frac{H^2 S(A+B)}{d_m^n} \log\frac{KH}{\delta} f(\sigma^+, \sigma^-, H). \tag{24}$$

Now, we introduce a lower bound of sample complexity in RTZMGs, whose proof is in Appendix D.

**Theorem 2 (Lower bound for solving robust MGs)** *Consider any tuple $\mathcal{MG}_r = \left\{\mathcal{S}, \mathcal{A}, \mathcal{B}, \mathcal{U}^{\sigma^+}(P^0), \mathcal{U}^{\sigma^-}(P^0), r, H\right\}$ obeying $H > 16\log 2$ and $\sigma^+$, $\sigma^- \in (0, 1-c_0]$ with any small efficiently positive constant $0 < c_0 \leq \frac{1}{4}$. Let*

$$\varepsilon \leq \begin{cases} \frac{c_2}{H}, & \text{if } \max\{\sigma^+, \sigma^-\} \leq \frac{c_2}{2H}, \\ 1 & \text{otherwise} \end{cases} \tag{25}$$

*for any $c_2 \leq \frac{1}{4}$. With an initial state distribution $\varrho$, we can construct a set of RTZMGs $\left\{\mathcal{M}_f^\phi | f \in \mathcal{F} = \{0, 1, \cdots, SA-1\}, \phi = [\phi_h]_{1\leq h\leq H} \in \Phi \subseteq \{0,1\}^H\right\}$ such that for any dataset with $K$ independent samples trajectories and $H$ lengths per trajectories satisfying $C \leq C_r^\star \leq 2C$, such that*

$$\inf_{\widehat{\mu}, \widehat{\nu}} \max_{(f,\phi) \in \mathcal{F} \times \Phi} \left\{\mathbb{P}_\phi\big(\mathrm{Gap}(\widehat{\mu}, \widehat{\nu}) > \varepsilon\big)\right\} \geq \frac{1}{8}, \tag{26}$$

*provided that*

$$T = KH \leq \frac{c_2 C_r^\star H^3 S(A+B)\min\{\frac{1}{\min\{\sigma^+, \sigma^-\}}, H\}}{\varepsilon^2}. \tag{27}$$

*Here, $c_2$ denotes an efficiently small constant. The infimum is obtained over all estimators $(\widehat{\mu}, \widehat{\nu})$.*

Moreover, our algorithm can be extended to multi-player general-sum Markov games with $m$ players and $A_i$ actions and uncertainty size $\sigma_i$ per player with details provided in Appendix F, i.e., Multi-RTZ-VI-LCB. Specifically, we obtain the following theoretical guarantee of Multi-RTZ-VI-LCB:

**Theorem 3 (Upper bound for Multi-RTZ-VI-LCB)** *Consider any $\delta \in (0,1)$ and any robust multi-player general-sum MGs $\mathcal{MG}_r = \mathcal{M}(\mathcal{S}, \{\mathcal{A}_i\}_{i=1}^m, H, \{\mathcal{U}_\rho^{\sigma_i}(P^0)\}_{i=1}^m, \{r_i\}_{i=1}^m)$. Under the TV uncertainty set $\mathcal{U}^{\sigma_i}(\cdot)$ defined in (2) with $\sigma_i \in (0,1]$ for $i = 1, 2, \cdots, m$. Define $d_m^n =$*

$\min_{h,s,\boldsymbol{a}} \{d_h^n(s,\boldsymbol{a}) : d_h^n(s,\boldsymbol{a}) > 0\}$, and $f(\{\sigma_i\}_{i=1}^m, H) = \min \left\{ \left\{ \frac{(H\sigma_i - 1 + (1-\sigma_i)^H)}{(\sigma_i)^2} \right\}_{i=1}^m, H \right\}$.
*For sufficient large constants $c_0, c_1 > 0$, with probability of at least $1 - \delta$, we can achieve*

$$\text{Gap}(\widehat{\pi}) \leq c_1 \sqrt{\frac{C_r^\star H^3 S \sum_{i=1}^m A_i \log \frac{KH}{\delta}}{K} f(\{\sigma_i\}_{i=1}^m, H)}, \tag{28}$$

*with the total number of samples $T$ exceeding*

$$T = KH \geq c_0 \frac{H^2 S \sum_{i=1}^m A_i}{d_m^n} \log \frac{KH}{\delta} f(\{\sigma_i\}_{i=1}^m, H). \tag{29}$$

Here are the key implications of these theorems:

- Theorem 1 demonstrates that the proposed RTZ-VI-LCB algorithm can attain an $\varepsilon$-robust NE solution when the total sample size exceeds:

$$\widetilde{O}\left( \frac{C_r^\star H^4 S(A+B)}{\varepsilon^2} \min \left\{ \frac{(H\sigma^+ - 1 + (1-\sigma^+)^H)}{(\sigma^+)^2}, \frac{(H\sigma^- - 1 + (1-\sigma^-)^H)}{(\sigma^-)^2}, H \right\} \right),$$

  suggesting that the sample efficiency for robust offline TZMGs is strongly influenced by the dataset quality (quantified by $C_r^\star$) and the problem structure of RTZMGs (reflected in the occupancy distributions $d_m^n$). If $C_r^\star$ is as small as $\frac{1}{S(A+B)}$, the upper bound of the sample complexity exhibits a weaker dependency on actions $\{A, B\}$ and state $S$. Combining this upper bound with the lower bound in Theorem 2 shows that RTZ-VI-LCB's sample complexity is optimal w.r.t. key factors $S$, $A$, $B$ and $\varepsilon$. This is the first optimal sample complexity upper bound for offline RTZMGs, regarding state $S$ and actions $\{A, B\}$.

- Theorem 2 conveys two important points. When the uncertainty level is small (i.e., $\min\{\sigma^+, \sigma^-\} \lesssim \frac{1}{H}$), no algorithm can find an $\varepsilon$-optimal robust policy with fewer than $\Omega\left( \frac{C_r^\star SH^4(A+B)}{\varepsilon^2} \right)$ samples, matching the complexity requirement for non-robust offline TZMGs (Jin et al., 2022). This implies that robust TZMGs are at least as challenging as standard TZMGs for low uncertainty. When the uncertainty level satisfies $\min\{\sigma^+, \sigma^-\} \gtrsim \frac{1}{H}$, no algorithm can find an $\varepsilon$-optimal robust policy with the numbers of samples fewer than $\Omega\left( \frac{C_r^\star SH^3(A+B)}{\varepsilon^2 \min\{\sigma^+, \sigma^-\}} \right)$. Thus, RTZ-VI-LCB is the first provably near-optimal algorithm on $S$ and $\{A, B\}$ for RTZMGs without requiring full coverage assumptions.

- Theorem 3 demonstrates that the proposed Multi-RTZ-VI-LCB algorithm can attain an $\varepsilon$-robust NE solution when the total sample size exceeds:

$$\widetilde{O}\left( \frac{C_r^\star H^4 S \sum_{i=1}^m A_i}{\varepsilon^2} \min \left\{ \left\{ \frac{(H\sigma_i - 1 + (1-\sigma_i)^H)}{(\sigma_i)^2} \right\}_{i=1}^m, H \right\} \right),$$

  suggesting that the algorithm can break the curse of multiagency.

## 5 CONCLUSION

To balance model robustness with sample efficiency, we design an efficient robust model-based algorithm for offline RTZMGs, which is value iteration with lower confidence bounds for RTZMGs (RTZ-VI-LCB). Our algorithm integrates robust VI with the principle of pessimism. By imposing a tailored and mild assumption (robust unilateral clipped concentrability) on the historical dataset to account for the distribution shift, we do not require full state-action space coverage. We address robustness against the distribution shifts in the worse-case scenario of the shared environment, analyze the finite-sample complexity of the proposed RTZ-VI-LCB algorithm, and establish an information-theoretic lower bound to evaluate its optimality across various uncertainty levels.

To the best of our knowledge, this is the first provably optimal algorithm for offline RTZMGs that addresses the dependency on states $S$ and actions $\{A, B\}$, while accounting for model perturbations and partial coverage. Furthermore, we extend RTZ-VI-LCB to multi-agent general-sum MGs, demonstrating a breakthrough in breaking the curse of multiagency. Our algorithm opens up several intriguing questions, such as designing efficient model-free algorithms for robust offline TZMGs with partial coverage and exploring ways to adjust the size and metric of the uncertainty set to complete the algorithmic design.

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

## A PRELIMINARIES

**Dual equivalence of robust Bellman.** We can compute the robust Bellman operator by solving its dual formulation rather than the original form, as long as the predefined uncertainty set is in a benign form (e.g., utilizing TV distance as the divergence function) (Iyengar, 2005; Shi et al., 2024a). Taking TV distance as an example, we describe the equivalence under strong duality between the robust Bellman operator and its dual form as Lemma 2.

**Lemma 2** *Consider any TV uncertainty set $\mathcal{U}^{\sigma^+}(P)$ and $\mathcal{U}^{\sigma^-}(P)$ associated with fixed uncertainty levels $\sigma^+$ and $\sigma^- \in (0,1]$ and any probability vector $P \in \Delta(\mathcal{S})$, respectively. For any vector $V \in \mathbb{R}^S$ obeying $V \geq 0$, one has*

$$\inf_{P \in \mathcal{U}^{\sigma^+}(P)} PV = \max_{\alpha \in [\min_s V(s), \max_s V(s)]} \left\{ P[V]_\alpha - \sigma^+ \left( \alpha - \min_{s'} [V]_\alpha (s') \right) \right\};  \tag{30a}$$

$$\sup_{P \in \mathcal{U}^{\sigma^-}(P)} PV = \max_{\alpha \in [\min_s V(s), \max_s V(s)]} \left\{ P[V]_\alpha - \sigma^- \left( \alpha - \min_{s'} [V]_\alpha (s') \right) \right\},  \tag{30b}$$

*where $[V]_\alpha$ is defined in (19)*

The proof of Lemma 2 is similar to Iyengar (2005, Lemma 4.3). Therefore, comparing the standard Bellman operator, the lemma above guarantees that no more computation cost is required when applying the robust Bellman operator, ignoring some logarithmic factors (Iyengar, 2005).

**Facts of RTZMGs and empirical RTZMGs.** Recall the definition of any RTZMG $\mathcal{MG}_r = \{\mathcal{S}, \mathcal{A}, \mathcal{B}, \mathcal{U}_\rho^{\sigma^+}(P^0), \mathcal{U}_\rho^{\sigma^-}(P^0), r, H\}$. According to robust Bellman equations in (8), one has: for any product policy $(\mu, \nu)$ and any $(h, s, a, b) \in [H] \times \mathcal{S} \times \mathcal{A} \times \mathcal{B}$,

$$Q_h^{\mu,\nu,\sigma^+}(s,a,b) = r_h(s,a,b) + \inf_{P \in \mathcal{U}_\rho^{\sigma^+}(P_{h,s,a,b}^0)} PV_{h+1}^{\mu,\nu,\sigma^+};  \tag{31a}$$

$$Q_h^{\mu,\nu,\sigma^-}(s,a,b) = r_h(s,a,b) + \sup_{P \in \mathcal{U}_\rho^{\sigma^-}(P_{h,s,a,b}^0)} PV_{h+1}^{\mu,\nu,\sigma^-},  \tag{31b}$$

where

$$V_h^{\mu,\nu,\sigma^+}(s) = \mathbb{E}_{a \sim \mu_h(s), b \sim \nu_h(s)} \left[ Q_h^{\mu,\nu,\sigma^+}(s,a,b) \right];$$

$$V_h^{\mu,\nu,\sigma^-}(s) = \mathbb{E}_{a \sim \mu_h(s), b \sim \nu_h(s)} \left[ Q_h^{\mu,\nu,\sigma^-}(s,a,b) \right].$$

Considering the offline setting, we use $\widehat{\mathcal{MG}}_r = \{\mathcal{S}, \mathcal{A}, \mathcal{B}, \mathcal{U}_\rho^{\sigma^+}(\widehat{P}^0), \mathcal{U}_\rho^{\sigma^-}(\widehat{P}^0), r, H\}$ to represent the empirical RTZMG, which is establishing along with the estimated nominal distribution $\widehat{P}^0$ in (12). Therefore, for any product policy $(\mu, \nu)$, we define the empirical robust value function (resp. empirical robust Q-function) in $\widehat{\mathcal{MG}}_r$ as $\widehat{V}_h^{\mu,\nu,\sigma^+}$ and $\widehat{V}_h^{\mu,\nu,\sigma^-}$ (resp. $\widehat{Q}_h^{\mu,\nu,\sigma^+}$ and $\widehat{Q}_h^{\mu,\nu,\sigma^-}$), which are analogous to (4). Moreover, we can similarly define the optimal of the empirical robust value function for both player over $\widehat{\mathcal{MG}}_r$, which is: for $\forall s \in \mathcal{S}$,

$$\widehat{V}_h^{\star,\nu,\sigma^+}(s) = \widehat{V}_h^{\mu^\star,\nu,\sigma^+}(s) := \max_{\mu:\mathcal{S} \times [H] \to \Delta(\mathcal{A})} \widehat{V}_h^{\mu,\nu,\sigma^+}(s) = \max_{\mu:\mathcal{S} \times [H] \to \Delta(\mathcal{A})} \inf_{P \in \mathcal{U}^{\sigma^+}(\widehat{P}^0)} \widehat{V}_h^{\mu,\nu,P}(s);  \tag{32a}$$

$$\widehat{V}_h^{\mu,\star,\sigma^-}(s) = \widehat{V}_h^{\mu,\nu^\star,\sigma^-}(s) := \max_{\nu:\mathcal{S} \times [H] \to \Delta(\mathcal{B})} \widehat{V}_h^{\mu,\nu,\sigma^-}(s) = \max_{\nu:\mathcal{S} \times [H] \to \Delta(\mathcal{B})} \inf_{P \in \mathcal{U}^{\sigma^-}(\widehat{P}^0)} \widehat{V}_h^{\mu,\nu,P}(s).  \tag{32b}$$

Notably, for all $s \in \mathcal{S}$, there exists at least one *robust best-response* policy that can achieve $\widehat{V}_h^{\star,\nu,\sigma^+}(s)$ and $\widehat{V}_h^{\mu,\star,\sigma^-}(s)$, as proved by Blanchet et al. (2024).

Therefore, we can obtain the empirical robust Bellman equation similar to (8) as: for any product policy $(\mu, \nu)$,

$$\widehat{Q}_h^{\mu,\nu,\sigma^+}(s, a, b) = r_h(s, a, b) + \inf_{P \in \mathcal{U}_\rho^{\sigma^+}(\widehat{P}_{h,s,a,b}^0)} P\widehat{V}_{h+1}^{\mu,\nu,\sigma^+}; \tag{33a}$$

$$\widehat{Q}_h^{\mu,\nu,\sigma^-}(s, a, b) = r_h(s, a, b) + \sup_{P \in \mathcal{U}_\rho^{\sigma^-}(\widehat{P}_{h,s,a,b}^0)} P\widehat{V}_{h+1}^{\mu,\nu,\sigma^-}, \tag{33b}$$

where

$$\widehat{V}_h^{\mu,\nu,\sigma^+}(s) = \mathbb{E}_{a \sim \mu_h(s), b \sim \nu_h(s)}[\widehat{Q}_h^{\mu,\nu,\sigma^+}(s, a, b)];$$

$$\widehat{V}_h^{\mu,\nu,\sigma^-}(s) = \mathbb{E}_{a \sim \mu_h(s), b \sim \nu_h(s)}[\widehat{Q}_h^{\mu,\nu,\sigma^-}(s, a, b)].$$

# B    PROOF OF THEOREM 1

The proof of Theorem 1 can be separated into three steps, as outlined below.

## B.1    STEP 1: DECOUPLING STATISTICAL DEPENDENCY

Before bounding $\mathrm{Gap}(\widehat{\mu}, \widehat{\nu})$, we introduce an important lemma, quantifying the difference between $\widehat{P}$ and $P$ when projected in the direction of the value function.

**Lemma 3** *Instate the assumptions in Theorem 1. Consider any vector $V \in \mathbb{R}^S$ with $\|V\|_\infty \leq H$ for all $(h, s, a, b) \in [H] \times \mathcal{S} \times \mathcal{A} \times \mathcal{B}$ satisfying $N_h(s, a, b) > 0$. With probability at least $1 - \delta$, one has*

$$\left| \inf_{P \in \mathcal{U}^{\sigma^+}(\widehat{P}_{h,s,a,b}^0)} PV - \inf_{P \in \mathcal{U}^{\sigma^+}(P_{h,s,a,b}^0)} PV \right| \leq C_4 \sqrt{\frac{1}{N_h(s,a,b)} \mathsf{Var}_{\widehat{P}_{h,s,a,b}^0}(V) \log \frac{KH}{\delta}} + C_4 \frac{H \log \frac{KH}{\delta}}{N_h(s,a,b)} \tag{34}$$

*for some sufficiently large constant $C_4 > 0$, and*

$$\mathsf{Var}_{\widehat{P}_{h,s,a,b}^0}(V) \leq 2\mathsf{Var}_{P_{h,s,a,b}^0}(V) + O\left( \frac{H^2}{N_h(s,a,b)} \log \frac{KH}{\delta} \right). \tag{35}$$

Proof can be found in Appendix C.3.

In simple terms, (34) provides a Bernstein-type concentration bound, while (35) ensures that the empirical variance estimate (i.e., the plug-in estimate) closely matches the true variance. Notably, Lemma 3 does not require $V$ to be statistically independent of $\widehat{P}_{h,s,a,b}^0$, which is essential given the complex statistical dependencies in our iterative algorithm. Under the leave-one-out analysis (see, e.g., Agarwal et al. (2020); Chen et al. (2021); Li et al. (2024a;b)), we prove Lemma 3 to decouple statistical dependencies, as illustrated in Appendix C.3. With Lemma 3, we can now have

$$\left| \inf_{\mathcal{P} \in \mathcal{U}^{\sigma^+}(\widehat{P}_{h,s,a,b}^0)} PV - \inf_{\mathcal{P} \in \mathcal{U}^{\sigma^+}(P_{h,s,a,b}^0)} PV \right| \leq \beta_h(s, a, b, V) \tag{36}$$

for any $(h, s, a, b) \in [H] \times \mathcal{S} \times \mathcal{A} \times \mathcal{B}$ satisfying $N_h(s, a, b) \geq 1$.

Therefore, we conclude that $\widehat{Q}_h^+(s, a, b)$ is an optimistic estimation of $\widehat{Q}_h^{\mu,\nu,\sigma^+}(s, a, b)$, which is summarized below.

**Lemma 4** *With probability exceeding $1 - \delta$, it holds that*

$$\widehat{Q}_h^+(s, a, b) \geq Q_h^{\star,\widehat{\nu},\sigma^+}(s, a, b) \qquad \text{and} \qquad \widehat{V}_h^+(s) \geq V_h^{\star,\widehat{\nu},\sigma^+}(s); \tag{37}$$

See Appendix C.4 for detail proofs.

Besides, we introduce another key lemma highlighting the difference between RTZMGs and standard TZMGs from the same idea by Shi et al. (2024b, Lemma 3). The range of the robust value function narrows as the uncertainty level $\sigma^+$ of its uncertainty set increases, as shown below.

**Lemma 5** *Consider the uncertainty set $\mathcal{U}^{\sigma^+}(\cdot)$ with TV distance and any RTZMG $\mathcal{MG}_r = \left\{ \mathcal{S}, \mathcal{A}, \mathcal{B}, \mathcal{U}^{\sigma^+}(P), \mathcal{U}^{\sigma^-}(P), r, H \right\}$. The optimistic robust value function estimate $\widehat{V}_h^+$:*

$$\forall h \in [H]: \quad \max_{s \in \mathcal{S}} \widehat{V}_h^+ - \min_{s \in \mathcal{S}} \widehat{V}_h^+ \leq \min \left\{ \frac{(H+1)\left(1 - (1 - \sigma^+)^{H-h}\right)}{\sigma^+}, H \right\}.$$

See Appendix C.5 for detail proofs.

### B.2 STEP 2: DECOMPOSING THE ERROR $\mathrm{Gap}(\widehat{\mu}, \widehat{\nu})$

The goal of our algorithm is to output an $\varepsilon$-robust NE policy $(\widehat{\mu}, \widehat{\nu})$ satisfying $\mathrm{Gap}(\widehat{\mu}, \widehat{\nu})$ in (11), i.e.,

$$\mathrm{Gap}(\widehat{\mu}, \widehat{\nu}) := \max \left\{ V_1^{\star, \widehat{\nu}, \sigma^+}(\varrho) - V_1^{\star, \sigma^+}(\varrho), \; V_1^{\star, \sigma^-}(\varrho) - V_1^{\widehat{\mu}, \star, \sigma^-}(\varrho) \right\} \leq \varepsilon.$$

Due to the symmetry between max-player and min-player, we assume without loss of generality that $V_1^{\star, \widehat{\nu}, \sigma^+}(\varrho) - V_1^{\star, \sigma^+}(\varrho)$ is larger than $V_1^{\star, \sigma^-}(\varrho) - V_1^{\widehat{\mu}, \star, \sigma^-}(\varrho)$, leading to $\mathrm{Gap}(\widehat{\mu}, \widehat{\nu}) \leq \left\{ V_1^{\star, \widehat{\nu}, \sigma^+}(\varrho) - V_1^{\star, \sigma^+}(\varrho) \right\}$.

According to the relationship in Lemma 4, we obtain

$$V_h^{\star, \widehat{\nu}, \sigma^+}(s) \leq \widehat{V}_h^+(s) = \max_{\mu \in \Delta(\mathcal{A})} \min_{\nu \in \Delta(\mathcal{B})} \mathbb{E}_{(a,b) \sim (\mu(s), \nu(s))} \left[ \widehat{Q}_h^+(s, a, b) \right]$$

$$\leq \max_{\mu \in \Delta(\mathcal{A})} \mathbb{E}_{(a,b) \sim (\mu(s), \nu^\star(s))} \left[ Q_h^+(s, a, b) \right], \tag{38}$$

where the first equality comes from line 6 in Algorithm 2. Therefore, there exists a deterministic policy $\mu^{\mathsf{d}} : \mathcal{S} \leftarrow \Delta(\mathcal{A})$ satisfying that for any $s \in \mathcal{S}$

$$\mu^{\mathsf{d}}(s) := \arg \max_{\mu \in \Delta(\mathcal{A})} \mathbb{E}_{(a,b) \sim (\mu(s), \nu^\star(s))} \left[ Q_h^+(s, a, b) \right]. \tag{39}$$

Before starting, we introduce several useful notations:

- The state-action space covered by the behavior policy $(\mu^{\mathsf{n}}, \nu^{\mathsf{n}})$ in the nominal transition kernel $P^0$ is denoted as

$$\mathcal{C}^{\mathsf{n}} = \left\{ (h, s, a, b) : d_h^{\mathsf{n}}(s, a, b) > 0 \right\}. \tag{40}$$

- The set of potential state occupancy distributions w.r.t. the policy $(\mu^{\mathsf{d}}(s), \nu^\star(s))$ in a model within the uncertainty set $P \in \mathcal{U}^{\sigma^+}(P^0)$ for any time step $h \in [H]$ is denoted as

$$\mathcal{D}_h^{\mathsf{p}} := \left\{ \left[ d_h^{\mu^{\mathsf{d}}(s), \nu^\star(s), P}(s) \right]_{s \in \mathcal{S}} : P \in \mathcal{U}^{\sigma^+}(P^0) \right\}; \tag{41}$$

$$\mathcal{D}_h^{\mathsf{pa}} := \left\{ \left[ d_h^{\mu^{\mathsf{d}}(s), \nu^\star(s), P}(s, a, b) \right]_{(s,a,b) \in \mathcal{S} \times \mathcal{A} \times \mathcal{B}} : P \in \mathcal{U}^{\sigma^+}(P^0) \right\}. \tag{42}$$

- For convenience and without ambiguity, we introduce an additional notation for $h \in [H]$ as

$$\beta_h^{\mu^{\mathsf{d}}, \nu^\star}(s) = \mathbb{E}_{(a,b) \sim (\mu^{\mathsf{d}}(s), \nu^\star(s))} \beta_h \left( s, a, b, \widehat{V}_{h+1}^+ \right).$$

In particular, the vector $\beta_h^{\mu^{\mathsf{d}}, \nu^\star} \in \mathbb{R}^S$ is defined with its $s$-th item given by $\beta_h^{\mu^{\mathsf{d}}, \nu^\star}(s)$.

- Similarly, we can define the notation related to rewards for $h \in [H]$ as

$$\widehat{r}_h^{\mu^{\mathsf{d}}, \nu^\star}(s) = \mathbb{E}_{(a,b) \sim (\mu^{\mathsf{d}}(s), \nu^\star(s))} \widehat{r}_h(s, a, b).$$

According to the update rule in line 4 in Algorithm 2 and robust Bellman equality (31), we derive

$$V_h^{\star,\widehat{\nu},\sigma^+}(s) - V_h^{\star,\sigma^+}(s)$$

$$\leq \widehat{V}_h^+(s) - V_h^{\mu^{\mathsf{d}},\nu^\star,\sigma^+}(s)$$

$$\leq \mathbb{E}_{(a,b)\sim(\mu^{\mathsf{d}}(s),\nu^\star(s))} \inf_{P\in\mathcal{U}^{\sigma^+}\left(\widehat{P}_{h,s,a,b}^0\right)} P\widehat{V}_{h+1}^+ + \beta_h^{\mu^{\mathsf{d}},\nu^\star}(s)$$

$$- \mathbb{E}_{(a,b)\sim(\mu^{\mathsf{d}}(s),\nu^\star(s))} \inf_{P\in\mathcal{U}^{\sigma^+}\left(P_{h,s,a,b}^0\right)} PV_{h+1}^{\mu^{\mathsf{d}},\nu^\star,\sigma^+}$$

$$\leq \mathbb{E}_{(a,b)\sim(\mu^{\mathsf{d}}(s),\nu^\star(s))} \left[ \inf_{P\in\mathcal{U}^{\sigma^+}\left(P_{h,s,a,b}^0\right)} P\widehat{V}_{h+1}^+ - \inf_{P\in\mathcal{U}^{\sigma^+}\left(P_{h,s,a,b}^0\right)} PV_{h+1}^{\mu^{\mathsf{d}},\nu^\star,\sigma^+} \right.$$

$$\left. + \left| \inf_{P\in\mathcal{U}^{\sigma^+}\left(P_{h,s,a,b}^0\right)} P\widehat{V}_{h+1}^+ - \inf_{P\in\mathcal{U}^{\sigma^+}\left(\widehat{P}_{h,s,a,b}^0\right)} P\widehat{V}_{h+1}^+ \right| \right] + \beta_h^{\mu^{\mathsf{d}},\nu^\star}(s)$$

$$\overset{(i)}{\leq} \mathbb{E}_{(a,b)\sim(\mu^{\mathsf{d}}(s),\nu^\star(s))} \left[ \inf_{P\in\mathcal{U}^{\sigma^+}\left(P_{h,s,a,b}^0\right)} P\widehat{V}_{h+1}^+ - \inf_{P\in\mathcal{U}^{\sigma^+}\left(P_{h,s,a,b}^0\right)} PV_{h+1}^{\mu^{\mathsf{d}},\nu^\star,\sigma^+} \right] + 2\beta_h^{\mu^{\mathsf{d}},\nu^\star}(s)$$

$$\overset{(ii)}{\leq} \mathbb{E}_{(a,b)\sim(\mu^{\mathsf{d}}(s),\nu^\star(s))} \left[ P_{h,s,a,b}^{\mathrm{inf},V} \left( \widehat{V}_{h+1}^+ - V_{h+1}^{\mu^{\mathsf{d}},\nu^\star,\sigma^+} \right) \right] + 2\beta_h^{\mu^{\mathsf{d}},\nu^\star}(s). \tag{43}$$

Here, (ii) is valid under the notation

$$P_{h,s,a,b}^{\mathrm{inf},V} := \mathrm{argmin}_{P\in\mathcal{U}^{\sigma^+}\left(P_{h,s,a,b}^0\right)} PV_{h+1}^{\mu^{\mathsf{d}},\nu^\star,\sigma^+} \tag{44}$$

and consequently,

$$\inf_{P\in\mathcal{U}^{\sigma^+}\left(P_{h,s,a,b}^0\right)} PV_{h+1}^{\mu^{\mathsf{d}},\nu^\star,\sigma^+} = P_{h,s,a,b}^{\mathrm{inf},V} V_{h+1}^{\mu^{\mathsf{d}},\nu^\star,\sigma^+}, \text{ and } \inf_{P\in\mathcal{U}^{\sigma^+}\left(P_{h,s,a,b}^0\right)} P\widehat{V}_{h+1}^+ \leq P_{h,s,a,b}^{\mathrm{inf},V} \widehat{V}_{h+1}^+.$$

Besides, (i) in (43) exists due to (36) in Lemma 3 for $N_h(s,a,b) > 0$ and

$$\left| \inf_{P\in\mathcal{U}^{\sigma^+}\left(P_{h,s,a,b}^0\right)} P\widehat{V}_{h+1}^+ - \inf_{P\in\mathcal{U}^{\sigma^+}\left(\widehat{P}_{h,s,a,b}^0\right)} P\widehat{V}_{h+1}^+ \right| \leq H = \beta_h^{\mu^{\mathsf{d}},\nu^\star}(s) \tag{45}$$

for $N_h(s,a,b) = 0$.

For ease of proof, we introduce a notation as $\widetilde{P}_{h,s}^{\mathrm{inf},V} := \mathbb{E}_{(a,b)\sim(\mu^{\mathsf{d}}(s),\nu^\star(s))} P_{h,s,a,b}^{\mathrm{inf},V}$. Furthermore, we define a sequence of matrices $\widetilde{P}_h^{\mathrm{inf},V} \in \mathbb{R}^{S\times S}$. We can utilizing (43) recursively over the time steps $h, h+1, \cdots, H$ and derive

$$V_h^{\star,\widehat{\nu},\sigma^+}(s) - V_h^{\star,\sigma^+}(s) \leq \widehat{V}_h^+(s) - V_h^{\mu^{\mathsf{d}},\nu^\star,\sigma^+}(s)$$

$$\leq \widetilde{P}_h^{\mathrm{inf},V} \left( \widehat{V}_{h+1}^+ - V_{h+1}^{\mu^{\mathsf{d}},\nu^\star,\sigma^+} \right) + 2\beta_h^{\mu^{\mathsf{d}},\nu^\star}(s)$$

$$\leq \widetilde{P}_h^{\mathrm{inf},V} \widetilde{P}_{h+1}^{\mathrm{inf},V} \left( \widehat{V}_{h+2}^+ - V_{h+2}^{\mu^{\mathsf{d}},\nu^\star,\sigma^+} \right) + 2\widetilde{P}_h^{\mathrm{inf},V} \beta_{h+1}^{\mu^{\mathsf{d}},\nu^\star} + 2\beta_h^{\mu^{\mathsf{d}},\nu^\star}(s)$$

$$\leq \cdots \leq 2\sum_{i=h}^{H} \left( \prod_{j=h}^{i-1} \widetilde{P}_j^{\mathrm{inf},V} \right) \beta_i^{\mu^{\mathsf{d}},\nu^\star}, \tag{46}$$

where we define $\left( \prod_{j=h}^{i-1} \widetilde{P}_j^{\mathrm{inf},V} \right) = I$ for convenience.

For any $d_h^{\mu^{\mathsf{d}},\nu^\star} \in \mathcal{D}_h^{\mathsf{p}}$ (cf. (41)), taking inner product with (46) yields

$$\left\langle d_h^{\mu^{\mathsf{d}},\nu^\star}, V_h^{\star,\widehat{\nu},\sigma^+}(s) - V_h^{\star,\sigma^+}(s) \right\rangle \le \left\langle d_h^{\mu^{\mathsf{d}},\nu^\star}, 2\sum_{i=h}^{H}\left(\prod_{j=h}^{i-1}\widetilde{P}_j^{\mathrm{inf},V}\right)\beta_i^{\mu^{\mathsf{d}},\nu^\star}\right\rangle$$

$$= 2\sum_{i=h}^{H}\left\langle d_i^{\mathsf{p},\mu^{\mathsf{d}},\nu^\star}, \beta_i^{\mu^{\mathsf{d}},\nu^\star}\right\rangle, \tag{47}$$

where

$$d_i^{\mathsf{p},\mu^{\mathsf{d}},\nu^\star} := \left[\left(d_h^{\mu^{\mathsf{d}},\nu^\star}\right)^\top\left(\prod_{j=h}^{i-1}\widetilde{P}_j^{\mathrm{inf},V}\right)\right]^\top \in \mathcal{D}_i^{\mathsf{p}} \tag{48}$$

by the definition of $\mathcal{D}_i^{\mathsf{p}}$ (cf. (41)) for all $i = h+1, \cdots, H$.

Next, we control $\langle d_i^{\mathsf{p},\mu^{\mathsf{d}},\nu^\star}, \beta_i^{\mu^{\mathsf{d}},\nu^\star}\rangle$ utilizing concentrability. First of all, according to (20) in Lemma 3, we demonstrate that the pessimistic penalty satisfies

$$\beta_i(s,a,b,\hat{V}) \le \max\left\{\sqrt{\frac{C_{\mathsf{n}}\log\frac{KH}{\delta}}{N_i(s,a,b)}\mathsf{Var}_{\widehat{P}_{i,s,a,b}^0}(\widehat{V})}, \frac{2C_{\mathsf{n}}H\log\frac{KH}{\delta}}{N_i(s,a,b)}\right\}$$

$$\le \sqrt{\frac{C_{\mathsf{n}}\log\frac{KH}{\delta}}{N_i(s,a,b)}\mathsf{Var}_{\widehat{P}_{i,s,a,b}^0}(\widehat{V})} + \frac{2C_{\mathsf{n}}H\log\frac{KH}{\delta}}{N_i(s,a,b)}$$

$$\overset{(i)}{\le} \sqrt{\frac{C_{\mathsf{n}}\log\frac{KH}{\delta}}{N_i(s,a,b)}\left(2\mathsf{Var}_{P_{i,s,a,b}^0}(\widehat{V}) + \frac{C_0 H^2}{N_i(s,a,b)}\log\frac{KH}{\delta}\right)} + \frac{2C_{\mathsf{n}}H\log\frac{KH}{\delta}}{N_i(s,a,b)}$$

$$\overset{(ii)}{\le} \sqrt{\frac{2C_{\mathsf{n}}\log\frac{KH}{\delta}}{N_i(s,a,b)}\mathsf{Var}_{P_{i,s,a,b}^0}(\widehat{V})} + \frac{\left(2C_{\mathsf{n}} + \sqrt{C_{\mathsf{n}}C_0}\right)H\log\frac{KH}{\delta}}{N_i(s,a,b)} \tag{49}$$

where (i) holds by applying (35) for some sufficiently large $C_0$ and (ii) exists follows from the Cauchy-Schwarz inequality. Therefore, combining the definition of $\beta_i^{\mu^{\mathsf{d}},\nu^\star}(s)$, we obtain

$$\langle d_i^{\mathsf{p},\mu^{\mathsf{d}},\nu^\star}, \beta_i^{\mu^{\mathsf{d}},\nu^\star}\rangle = \sum_{s\in\mathcal{S}} d_i^{\mathsf{p},\mu^{\mathsf{d}},\nu^\star}(s)\beta_i^{\mu^{\mathsf{d}},\nu^\star}(s)$$

$$= \sum_{s\in\mathcal{S}} d_i^{\mathsf{p},\mu^{\mathsf{d}},\nu^\star}(s)\mathbb{E}_{(a,b)\sim(\mu^{\mathsf{d}}(s),\nu^\star(s))}\beta_i(s,a,b,\hat{V})$$

$$= \sum_{(s,a,b)\in\mathcal{S}\times\mathcal{A}\times\mathcal{B}} d_i^{\mathsf{p},\mu^{\mathsf{d}},\nu^\star}(s)\mathbb{1}\{a=\mu^{\mathsf{d}}(s)\}\nu^\star(b|s)\beta_i(s,a,b,\hat{V})$$

$$= \sum_{(s,b)\in\mathcal{S}\times\mathcal{B}} d_i^{\mathsf{p},\mu^{\mathsf{d}},\nu^\star}(s,\mu^{\mathsf{d}}(s),b)\beta_i(s,\mu^{\mathsf{d}}(s),b,\hat{V}), \tag{50}$$

where the last equation holds due to the definition in (7b). Then, we observe $d_h^{\mathsf{p},\mu^{\mathsf{d}},\nu^\star}(s,a,b) \in \mathcal{D}_h^{\mathsf{pa}}$ (cf. (42)). Thereafter, we divide the bound (50) into two cases.

**For the first case**, i.e., $s \in S$ where $\max_{P\in\mathcal{U}^{\sigma^+}(P^0)} d_i^{\mu^{\mathsf{d}},\nu^\star,P}(s,\mu^{\mathsf{d}}(s),b) = 0$, it follows from the definition (cf. (41)) that for any $d_i^{\mathsf{p},\mu^{\mathsf{d}},\nu^\star}(s,\mu^{\mathsf{d}}(s),b) \in \mathcal{D}_i^{\mathsf{pa}}$, it satisfies that

$$d_i^{\mathsf{p},\mu^{\mathsf{d}},\nu^\star}(s,\mu^{\mathsf{d}}(s),b) = 0. \tag{51}$$

**For the second case**, i.e., $s \in S$ where $\max_{P\in\mathcal{U}^{\sigma^+}(P^0)} d_i^{\mu^{\mathsf{d}},\nu^\star,P}(s,\mu^{\mathsf{d}}(s),b) > 0$, by the assumption in (22)

$$\max_{P\in\mathcal{U}^{\sigma^+}(P^0)} \frac{\min\left\{d_i^{\mu^{\mathsf{d}},\nu^\star,P}(s,\mu^{\mathsf{d}}(s),b), \frac{1}{S(A+B)}\right\}}{d_i^{\mathsf{n}}(s,\mu^{\mathsf{d}}(s),b)} \le C_{\mathsf{r}}^\star < \infty.$$

It implies that

$$d_i^{\mathsf{n}}\big(s,\mu^{\mathsf{d}}(s),b\big) > 0 \quad \text{and} \quad \big(i,s,\mu^{\mathsf{d}}(s),b\big) \in \mathcal{C}^{\mathsf{n}}. \tag{52}$$

Lemma 1 tells that with probability at least $1 - 8\delta$,

$$
\begin{aligned}
N_i\big(s,\mu^{\mathsf{d}}(s),b\big) &\geq \frac{Kd_i^{\mathsf{n}}\big(s,\mu^{\mathsf{d}}(s),b\big)}{8} - 5\sqrt{Kd_i^{\mathsf{n}}\big(s,\mu^{\mathsf{d}}(s),b\big)\log\frac{KH}{\delta}} \\
&\overset{(i)}{\geq} \frac{Kd_i^{\mathsf{n}}\big(s,\mu^{\mathsf{d}}(s),b\big)}{16} \\
&\overset{(ii)}{\geq} \frac{K\max_{P\in\mathcal{U}^{\sigma}(P^0)}\min\left\{d_i^{\mu^{\mathsf{d}},\nu^{\star},P}\big(s,\mu^{\mathsf{d}}(s),b\big), \frac{1}{S(A+B)}\right\}}{16C_{\mathsf{r}}^{\star}} \\
&\geq \frac{K\min\left\{d_i^{\mathsf{p},\mu^{\mathsf{d}},\nu^{\star}}\big(s,\mu^{\mathsf{d}}(s),b\big), \frac{1}{S(A+B)}\right\}}{16C_{\mathsf{r}}^{\star}},
\end{aligned} \tag{53}
$$

where (ii) comes from Assumption 1 and (i) holds due to

$$
\begin{aligned}
Kd_i^{\mathsf{n}}\big(s,\mu^{\mathsf{d}}(s),b\big) &\geq c_0\frac{HS(A+B)}{d_{\mathsf{m}}^{\mathsf{n}}}\log\frac{KH}{\delta}f(\sigma^+,\sigma^-,H)d_i^{\mathsf{n}}\big(s,\mu^{\mathsf{d}}(s),b\big) \\
&\geq c_0HS(A+B)\log\frac{KH}{\delta}f(\sigma^+,\sigma^-,H) \geq 1600\log\frac{KH}{\delta},
\end{aligned} \tag{54}
$$

where $f(\sigma^+,\sigma^-,H) = \min\left\{\frac{H\sigma^++1-(1-\sigma^+)^H}{(\sigma^+)^2}, \frac{H\sigma^-+1-(1-\sigma^-)^H}{(\sigma^-)^2}, H\right\}$, the first inequality follows from condition (24), and the second inequality follows from

$$d_{\mathsf{m}}^{\mathsf{n}} = \min_{h,s,\mu^{\mathsf{d}}(s),b}\left\{d_h^{\mathsf{n}}(s,\mu^{\mathsf{d}}(s),b) : d_h^{\mathsf{n}}(s,\mu^{\mathsf{d}}(s),b) > 0\right\} \leq d_i^{\mathsf{n}}\big(s,\mu^{\mathsf{d}}(s),b\big). \tag{55}$$

Combining the results in (49) and (50), we arrive at

$$
\begin{aligned}
&\langle d_i^{\mathsf{p},\mu^{\mathsf{d}},\nu^{\star}}, \beta_i^{\mu^{\mathsf{d}},\nu^{\star}}\rangle \\
&= \sum_{(s,b)\in\mathcal{S}\times\mathcal{B}} d_i^{\mathsf{p},\mu^{\mathsf{d}},\nu^{\star}}(s,\mu^{\mathsf{d}}(s),b)\beta_i(s,\mu^{\mathsf{d}}(s),b,\hat{V}) \\
&\leq \sum_{(s,b)\in\mathcal{S}\times\mathcal{B}} d_i^{\mathsf{p},\mu^{\mathsf{d}},\nu^{\star}}(s,\mu^{\mathsf{d}}(s),b)\sqrt{\frac{2C_{\mathsf{n}}\log\frac{KH}{\delta}}{N_i\big(s,\mu^{\mathsf{d}}(s),b\big)}\mathsf{Var}_{P_{i,s,\mu^{\mathsf{d}}(s),b}^0}\big(\widehat{V}\big)} \\
&\quad + \sum_{(s,b)\in\mathcal{S}\times\mathcal{B}} d_i^{\mathsf{p},\mu^{\mathsf{d}},\nu^{\star}}(s,\mu^{\mathsf{d}}(s),b)\frac{\big(2C_{\mathsf{n}}+\sqrt{C_{\mathsf{n}}C_0}\big)H\log\frac{KH}{\delta}}{N_i\big(s,\mu^{\mathsf{d}}(s),b\big)} \\
&\overset{(i)}{\leq} \sum_{(s,b)\in\mathcal{S}\times\mathcal{B}} d_i^{\mathsf{p},\mu^{\mathsf{d}},\nu^{\star}}(s,\mu^{\mathsf{d}}(s),b)\underbrace{\sqrt{\frac{32C_{\mathsf{r}}^{\star}C_{\mathsf{n}}\log\frac{KH}{\delta}}{K\min\left\{d_i^{\mathsf{p},\mu^{\mathsf{d}},\nu^{\star}}(s,\mu^{\mathsf{d}}(s),b), \frac{1}{S(A+B)}\right\}}\mathsf{Var}_{P_{i,s,\mu^{\mathsf{d}}(s),b}^0}\big(\widehat{V}\big)}}_{B_1} \\
&\quad + \sum_{(s,b)\in\mathcal{S}\times\mathcal{B}} d_i^{\mathsf{p},\mu^{\mathsf{d}},\nu^{\star}}(s,\mu^{\mathsf{d}}(s),b)\underbrace{\frac{16C_{\mathsf{r}}^{\star}\big(2C_{\mathsf{n}}+\sqrt{C_{\mathsf{n}}C_0}\big)H\log\frac{KH}{\delta}}{K\min\left\{d_i^{\mathsf{p},\mu^{\mathsf{d}},\nu^{\star}}(s,\mu^{\mathsf{d}}(s),b), \frac{1}{S(A+B)}\right\}}}_{B_2}.
\end{aligned} \tag{56}
$$

Therefore, according to (47), we just need to bound $\sum_{i=1}^{H}\sum_{(s,b)\in\mathcal{S}\times\mathcal{B}} d_i^{\mathsf{p},\mu^{\mathsf{d}},\nu^{\star}}(s,\mu^{\mathsf{d}}(s),b)B_1$ and $\sum_{i=1}^{H}\sum_{(s,b)\in\mathcal{S}\times\mathcal{B}} d_i^{\mathsf{p},\mu^{\mathsf{d}},\nu^{\star}}(s,\mu^{\mathsf{d}}(s),b)B_2$, which is introduced as follows.

**Part 1: Bounding** $\sum_{i=1}^{H} \sum_{(s,b)\in\mathcal{S}\times\mathcal{B}} d_i^{\mathsf{p},\mu^{\mathsf{d}},\nu^{\star}}(s,\mu^{\mathsf{d}}(s),b)B_1$ Combining the result in (54) with $\sum_{i=1}^{H} \sum_{(s,b)\in\mathcal{S}\times\mathcal{B}} d_i^{\mathsf{p},\mu^{\mathsf{d}},\nu^{\star}}(s,\mu^{\mathsf{d}}(s),b)B_1$ yields

$$
\begin{aligned}
&\sum_{i=1}^{H} \sum_{(s,b)\in\mathcal{S}\times\mathcal{B}} d_i^{\mathsf{p},\mu^{\mathsf{d}},\nu^{\star}}(s,\mu^{\mathsf{d}}(s),b)B_1 \\
&= \sum_{i=1}^{H} \sum_{(s,b)\in\mathcal{S}\times\mathcal{B}} d_i^{\mathsf{p},\mu^{\mathsf{d}},\nu^{\star}}(s,\mu^{\mathsf{d}}(s),b) \sqrt{\frac{32 C_{\mathsf{r}}^{\star} C_{\mathsf{n}} \log \frac{KH}{\delta}}{K \min\left\{ d_i^{\mathsf{p},\mu^{\mathsf{d}},\nu^{\star}}(s,\mu^{\mathsf{d}}(s),b), \frac{1}{S(A+B)} \right\}} \mathsf{Var}_{P^0_{i,s,\mu^{\mathsf{d}}(s),b}}\big(\widehat{V}\big)} \\
&\leq \sum_{i=1}^{H} \sum_{(s,b)\in\mathcal{S}\times\mathcal{B}} d_i^{\mathsf{p},\mu^{\mathsf{d}},\nu^{\star}}(s,\mu^{\mathsf{d}}(s),b) \times \\
&\qquad \max\left\{ \sqrt{\frac{32 C_{\mathsf{r}}^{\star} C_{\mathsf{n}} \log \frac{KH}{\delta}}{K d_i^{\mathsf{p},\mu^{\mathsf{d}},\nu^{\star}}(s,\mu^{\mathsf{d}}(s),b)} \mathsf{Var}_{P^0_{i,s,\mu^{\mathsf{d}}(s),b}}\big(\widehat{V}\big)}, \sqrt{\frac{32 C_{\mathsf{r}}^{\star} C_{\mathsf{n}} S(A+B) \log \frac{KH}{\delta}}{K} \mathsf{Var}_{P^0_{i,s,\mu^{\mathsf{d}}(s),b}}\big(\widehat{V}\big)} \right\} \\
&\leq \sum_{i=1}^{H} \sum_{(s,b)\in\mathcal{S}\times\mathcal{B}} \sqrt{\frac{32 C_{\mathsf{r}}^{\star} C_{\mathsf{n}} \log \frac{KH}{\delta}}{K} d_i^{\mathsf{p},\mu^{\mathsf{d}},\nu^{\star}}(s,\mu^{\mathsf{d}}(s),b)\mathsf{Var}_{P^0_{i,s,\mu^{\mathsf{d}}(s),b}}\big(\widehat{V}\big)} \\
&\quad + \sum_{i=1}^{H} \sum_{(s,b)\in\mathcal{S}\times\mathcal{B}} d_i^{\mathsf{p},\mu^{\mathsf{d}},\nu^{\star}}(s,\mu^{\mathsf{d}}(s),b) \sqrt{\frac{32 C_{\mathsf{r}}^{\star} C_{\mathsf{n}} S(A+B) \log \frac{KH}{\delta}}{K} \mathsf{Var}_{P^0_{i,s,\mu^{\mathsf{d}}(s),b}}\big(\widehat{V}\big)} \\
&\leq \sqrt{\frac{32 C_{\mathsf{r}}^{\star} C_{\mathsf{n}} S(A+B) \log \frac{KH}{\delta}}{K}} \left( \sqrt{H \sum_{i=1}^{H} \sum_{(s,b)\in\mathcal{S}\times\mathcal{B}} d_i^{\mathsf{p},\mu^{\mathsf{d}},\nu^{\star}}(s,\mu^{\mathsf{d}}(s),b)\mathsf{Var}_{P^0_{i,s,\mu^{\mathsf{d}}(s),b}}\big(\widehat{V}\big)} \right. \\
&\quad \left. + \sqrt{\sum_{i=1}^{H} \sum_{(s,b)\in\mathcal{S}\times\mathcal{B}} d_i^{\mathsf{p},\mu^{\mathsf{d}},\nu^{\star}}(s,\mu^{\mathsf{d}}(s),b)\mathsf{Var}_{P^0_{i,s,\mu^{\mathsf{d}}(s),b}}\big(\widehat{V}\big)} \times \sqrt{\sum_{i=1}^{H} \sum_{(s,b)\in\mathcal{S}\times\mathcal{B}} d_i^{\mathsf{p},\mu^{\mathsf{d}},\nu^{\star}}(s,\mu^{\mathsf{d}}(s),b)} \right) \\
&= \sqrt{\frac{128 C_{\mathsf{r}}^{\star} C_{\mathsf{n}} HS(A+B) \log \frac{KH}{\delta}}{K} \sum_{i=1}^{H} \sum_{(s,b)\in\mathcal{S}\times\mathcal{B}} d_i^{\mathsf{p},\mu^{\mathsf{d}},\nu^{\star}}(s,\mu^{\mathsf{d}}(s),b)\mathsf{Var}_{P^0_{i,s,\mu^{\mathsf{d}}(s),b}}\big(\widehat{V}\big)}, \qquad (57)
\end{aligned}
$$

where the last inequality follows from the Cauchy-Schwarz inequality. Then, we introduce the following lemma about $\sum_{i=1}^{H} \sum_{(s,b)\in\mathcal{S}\times\mathcal{B}} d_i^{\mathsf{p},\mu^{\mathsf{d}},\nu^{\star}}(s,\mu^{\mathsf{d}}(s),b)\mathsf{Var}_{P_{i,s,\mu^{\mathsf{d}}(s),b}}\big(\widehat{V}\big)$, whose proof is postponed to Appendix C.6.

**Lemma 6** *Considering $\forall \delta \in (0,1)$, with probability at least $1-\delta$, one has: for any product policy $(\widehat{\mu},\widehat{\nu})$,*

$$
\begin{aligned}
&\sum_{i=1}^{H} \sum_{(s,b)\in\mathcal{S}\times\mathcal{B}} d_i^{\mathsf{p},\mu^{\mathsf{d}},\nu^{\star}}(s,\mu^{\mathsf{d}}(s),b)\mathsf{Var}_{P^0_{i,s,a,b}}\big(\widehat{V}_{i+1}\big) \leq H \min\left\{ \frac{2(H\sigma^+ - 1 + (1-\sigma^+)^H)}{(\sigma^+)^2}, H \right\} \\
&\qquad \times \left( 4 \sum_{i=1}^{H} \sum_{(s,b)\in\mathcal{S}\times\mathcal{B}} d_i^{\mathsf{p},\mu^{\mathsf{d}},\nu^{\star}}(s,\mu^{\mathsf{d}}(s),b)\beta_i(s,\mu^{\mathsf{d}}(s),b,\widehat{V}) + (H+3) \right). \qquad (58)
\end{aligned}
$$

Armed with Lemma 6, (57) can be further bounded as

$$
\sum_{i=1}^{H} \sum_{(s,b)\in\mathcal{S}\times\mathcal{B}} d_i^{\mathsf{p},\mu^{\mathsf{d}},\nu^{\star}}(s,\mu^{\mathsf{d}}(s),b) B_1
$$

$$
\leq \sqrt{\frac{128 C_{\mathsf{r}}^{\star} C_{\mathsf{n}} H S (A+B) \log \frac{KH}{\delta}}{K}} \sqrt{H \min\left\{\frac{2(H\sigma^+ - 1 + (1-\sigma^+)^H)}{(\sigma^+)^2}, H\right\}}
$$

$$
\times \sqrt{\left(4 \sum_{i=1}^{H} \sum_{(s,b)\in\mathcal{S}\times\mathcal{B}} d_i^{\mathsf{p},\mu^{\mathsf{d}},\nu^{\star}}(s,\mu^{\mathsf{d}}(s),b) \beta_i(s,\mu^{\mathsf{d}}(s),b,\widehat{V}) + (H+3)\right)}. \tag{59}
$$

**Part 2: Bounding** $\sum_{i=1}^{H} \sum_{(s,b)\in\mathcal{S}\times\mathcal{B}} d_i^{\mathsf{p},\mu^{\mathsf{d}},\nu^{\star}}(s,\mu^{\mathsf{d}}(s),b) B_2$   Combining the result in (53) with $\sum_{i=1}^{H} \sum_{(s,b)\in\mathcal{S}\times\mathcal{B}} d_i^{\mathsf{p},\mu^{\mathsf{d}},\nu^{\star}}(s,\mu^{\mathsf{d}}(s),b) B_2$ yields

$$
\sum_{i=1}^{H} \sum_{(s,b)\in\mathcal{S}\times\mathcal{B}} d_i^{\mathsf{p},\mu^{\mathsf{d}},\nu^{\star}}(s,\mu^{\mathsf{d}}(s),b) B_2
$$

$$
= \sum_{i=1}^{H} \sum_{(s,b)\in\mathcal{S}\times\mathcal{B}} d_i^{\mathsf{p},\mu^{\mathsf{d}},\nu^{\star}}(s,\mu^{\mathsf{d}}(s),b) \frac{16 C_{\mathsf{r}}^{\star} \left(2C_{\mathsf{n}} + \sqrt{C_{\mathsf{n}}C_3}\right) H \log \frac{KH}{\delta}}{K \min\left\{d_i^{\mathsf{p},\mu^{\mathsf{d}},\nu^{\star}}(s,\mu^{\mathsf{d}}(s),b), \frac{1}{S(A+B)}\right\}}
$$

$$
\overset{(\mathrm{i})}{\leq} \frac{32 C_{\mathsf{r}}^{\star} \left(2C_{\mathsf{n}} + \sqrt{C_{\mathsf{n}}C_3}\right) H^2 S(A+B) \log \frac{KH}{\delta}}{K}, \tag{60}
$$

where the inequality holds by the trivial fact

$$
\sum_{(s,b)\in\mathcal{S}\times\mathcal{B}} \frac{d_i^{\mathsf{p},\mu^{\mathsf{d}},\nu^{\star}}(s,\mu^{\mathsf{d}}(s),b)}{\min\left\{d_i^{\mathsf{p},\mu^{\mathsf{d}},\nu^{\star}}(s,\mu^{\mathsf{d}}(s),b), \frac{1}{S(A+B)}\right\}}
$$

$$
\leq \sum_{(s,b)\in\mathcal{S}\times\mathcal{B}} d_i^{\mathsf{p},\mu^{\mathsf{d}},\nu^{\star}}(s,\mu^{\mathsf{d}}(s),b) \left(\frac{1}{d_i^{\mathsf{p},\mu^{\mathsf{d}},\nu^{\star}}(s,\mu^{\mathsf{d}}(s),b)} + \frac{1}{1/S(A+B)}\right)
$$

$$
= \sum_{(s,b)\in\mathcal{S}\times\mathcal{B}} 1 + S(A+B) \sum_{(s,b)\in\mathcal{S}\times\mathcal{B}} d_i^{\mathsf{p},\mu^{\mathsf{d}},\nu^{\star}}(s,\mu^{\mathsf{d}}(s),b) \leq 2S(A+B). \tag{61}
$$

**Putting all together**   Combining the results (59) and (60) in Part 1 and Part 2, we obtain

$$
\sum_{i=1}^{H} \sum_{(s,b)\in\mathcal{S}\times\mathcal{B}} d_i^{\mathsf{p},\mu^{\mathsf{d}},\nu^{\star}}(s,\mu^{\mathsf{d}}(s),b) \beta_i(s,\mu^{\mathsf{d}}(s),b,\widehat{V})
$$

$$
\leq \sqrt{\frac{128 C_{\mathsf{r}}^{\star} C_{\mathsf{n}} H^2 S (A+B) \log \frac{KH}{\delta}}{K} \min\left\{\frac{2(H\sigma^+ - 1 + (1-\sigma^+)^H)}{(\sigma^+)^2}, H\right\}}
$$

$$
\times \sqrt{\left(4 \sum_{i=1}^{H} \sum_{(s,b)\in\mathcal{S}\times\mathcal{B}} d_i^{\mathsf{p},\mu^{\mathsf{d}},\nu^{\star}}(s,\mu^{\mathsf{d}}(s),b) \beta_i(s,\mu^{\mathsf{d}}(s),b,\widehat{V}) + (H+3)\right)}
$$

$$
+ \frac{32 C_{\mathsf{r}}^{\star} \left(2C_{\mathsf{n}} + \sqrt{C_{\mathsf{n}}C_3}\right) H^2 S(A+B) \log \frac{KH}{\delta}}{K}, \tag{62}
$$

which can further bound as

$$\sum_{i=1}^{H} \sum_{(s,b)\in\mathcal{S}\times\mathcal{B}} d_i^{\mathsf{p},\mu^{\mathsf{d}},\nu^{\star}}(s,\mu^{\mathsf{d}}(s),b)\beta_i(s,\mu^{\mathsf{d}}(s),b,\widehat{V})$$

$$\leq \sqrt{\frac{128C_{\mathsf{r}}^{\star}C_{\mathsf{n}}H^2(H+3)S(A+B)\log\frac{KH}{\delta}}{K}\min\left\{\frac{2(H\sigma^+ - 1 + (1-\sigma^+)^H)}{(\sigma^+)^2}, H\right\}}$$

$$+ \frac{32C_{\mathsf{r}}^{\star}\left(2C_{\mathsf{n}}+\sqrt{C_{\mathsf{n}}C_3}\right)H^2 S(A+B)\log\frac{KH}{\delta}}{K} + \sqrt{\frac{512C_{\mathsf{r}}^{\star}C_{\mathsf{n}}H^2 S(A+B)\log\frac{KH}{\delta}}{K}}$$

$$\times \sqrt{\min\left\{\frac{2(H\sigma^+ - 1 + (1-\sigma^+)^H)}{(\sigma^+)^2}, H\right\} \sum_{i=1}^{H}\sum_{(s,b)\in\mathcal{S}\times\mathcal{B}} d_i^{\mathsf{p},\mu^{\mathsf{d}},\nu^{\star}}(s,\mu^{\mathsf{d}}(s),b)\beta_i(s,\mu^{\mathsf{d}}(s),b,\widehat{V})}$$

$$\leq \sqrt{\frac{128C_{\mathsf{r}}^{\star}C_{\mathsf{n}}H^2(H+3)S(A+B)\log\frac{KH}{\delta}}{K}\min\left\{\frac{2(H\sigma^+ - 1 + (1-\sigma^+)^H)}{(\sigma^+)^2}, H\right\}}$$

$$+ \frac{32C_{\mathsf{r}}^{\star}\left(2C_{\mathsf{n}}+\sqrt{C_{\mathsf{n}}C_3}\right)H^2 S(A+B)\log\frac{KH}{\delta}}{K}$$

$$+ \frac{256C_{\mathsf{r}}^{\star}C_{\mathsf{n}}H^2 S(A+B)\log\frac{KH}{\delta}}{K}\min\left\{\frac{2(H\sigma^+ - 1 + (1-\sigma^+)^H)}{(\sigma^+)^2}, H\right\}$$

$$+ \frac{1}{2}\sum_{i=1}^{H}\sum_{(s,b)\in\mathcal{S}\times\mathcal{B}} d_i^{\mathsf{p},\mu^{\mathsf{d}},\nu^{\star}}(s,\mu^{\mathsf{d}}(s),b)\beta_i(s,\mu^{\mathsf{d}}(s),b,\widehat{V}), \tag{63}$$

where the last relation follows from the AM-GM inequality. Rearranging terms, it follows that

$$\sum_{i=1}^{H}\sum_{(s,b)\in\mathcal{S}\times\mathcal{B}} d_i^{\mathsf{p},\mu^{\mathsf{d}},\nu^{\star}}(s,\mu^{\mathsf{d}}(s),b)\beta_i(s,\mu^{\mathsf{d}}(s),b,\widehat{V})$$

$$\leq \sqrt{\frac{512C_{\mathsf{r}}^{\star}C_{\mathsf{n}}H^2(H+3)S(A+B)\log\frac{KH}{\delta}}{K}\min\left\{\frac{2(H\sigma^+ - 1 + (1-\sigma^+)^H)}{(\sigma^+)^2}, H\right\}}$$

$$+ \frac{64C_{\mathsf{r}}^{\star}\left(2C_{\mathsf{n}}+\sqrt{C_{\mathsf{n}}C_3}\right)H^2 S(A+B)\log\frac{KH}{\delta}}{K}$$

$$+ \frac{512C_{\mathsf{r}}^{\star}C_{\mathsf{n}}H^2 S(A+B)\log\frac{KH}{\delta}}{K}\min\left\{\frac{2(H\sigma^+ - 1 + (1-\sigma^+)^H)}{(\sigma^+)^2}, H\right\}$$

$$\leq \sqrt{\frac{512C_{\mathsf{r}}^{\star}C_{\mathsf{n}}H^2(H+3)S(A+B)\log\frac{KH}{\delta}}{K}\min\left\{\frac{2(H\sigma^+ - 1 + (1-\sigma^+)^H)}{(\sigma^+)^2}, H\right\}}$$

$$+ \frac{C_{\mathsf{r}}^{\star}C_2 H^2 S(A+B)\log\frac{KH}{\delta}}{K}\min\left\{\frac{2(H\sigma^+ - 1 + (1-\sigma^+)^H)}{(\sigma^+)^2}, H\right\}. \tag{64}$$

Along with the above result, we are ready to bound $V_1^{\star,\sigma^+}(\varrho) - V_1^{\widehat{\mu},\star,\sigma^+}(\varrho)$. There exists some sufficiently large constants $C_1, C_2, C_3 > 0$, and

$$V_1^{\star,\widehat{\nu},\sigma^+}(\varrho) - V_1^{\star,\sigma^+}(\varrho) \leq \sqrt{\frac{C_{\mathsf{r}}^{\star}C_1 H^3 S(A+B)\log\frac{KH}{\delta}}{K}\min\left\{\frac{2(H\sigma^+ - 1 + (1-\sigma^+)^H)}{(\sigma^+)^2}, H\right\}}$$

$$+ \frac{C_{\mathsf{r}}^{\star}C_2 H^2 S(A+B)\log\frac{KH}{\delta}}{K}\min\left\{\frac{2(H\sigma^+ - 1 + (1-\sigma^+)^H)}{(\sigma^+)^2}, H\right\}$$

$$\leq \sqrt{\frac{C_{\mathsf{r}}^{\star}C_3 H^3 S(A+B)\log\frac{KH}{\delta}}{K}\min\left\{\frac{2(H\sigma^+ - 1 + (1-\sigma^+)^H)}{(\sigma^+)^2}, H\right\}}, \tag{65}$$

where the last inequality follows from condition (24).

### B.3 Step 3: Summing up the results

Consequently, we obtain the upper bound of $V_1^{\star,\widehat{\nu},\sigma^+}(\varrho) - V_1^{\widehat{\mu},\widehat{\nu},\sigma^+}(\varrho)$ in (65). Similarly,

$$V_1^{\star,\sigma^-}(\varrho) - V_1^{\widehat{\mu},\star,\sigma^-}(\varrho)$$

$$\leq \sqrt{\frac{C_r^\star C_3 H^2 S(A+B)\log\frac{KH}{\delta}}{K}\min\left\{\frac{(H+1)(H\sigma^- - 1 + (1-\sigma^-)^H)}{(\sigma^-)^2}, H\right\}}, \qquad (66)$$

which directly leads to

$$\mathrm{Gap}(\widehat{\mu},\widehat{\nu}) \leq c_1 \sqrt{\frac{C_r^\star H^2 S(A+B)\log\frac{KH}{\delta}}{K}}$$

$$\times \sqrt{\min\left\{\frac{2(H\sigma^+ - 1 + (1-\sigma^+)^H)}{(\sigma^+)^2}, \frac{2(H\sigma^- - 1 + (1-\sigma^-)^H)}{(\sigma^-)^2}, H\right\}}, \qquad (67)$$

for some sufficiently large $c_1$ and

$$K \geq HS(A+B)\log\frac{KH}{\delta}\min\left\{\frac{2(H\sigma^+ - 1 + (1-\sigma^+)^H)}{(\sigma^+)^2}, \frac{2(H\sigma^- - 1 + (1-\sigma^-)^H)}{(\sigma^-)^2}, H\right\}.$$

**Discussion of (67).** For the term $T = \min\left(f(\sigma^+, \sigma^-), H\right)$, considering the symmetry between $\sigma^+$ and $\sigma^-$, we define $g(\sigma^+, H) = H\sigma^+ - H(1-\sigma^+)^H - (\sigma^+)^2 H$. For $H \geq 2$, we derive the first derivative as $\frac{\partial g(\sigma^+, H)}{\partial \sigma^+} = H + H^2(1-\sigma^+)^{H-1} - 2H\sigma^+$. Further, the second derivative is given by $\frac{\partial^2 g(\sigma^+, H)}{\partial(\sigma^+)^2} = -H^2(H-1)(1-\sigma^+)^{H-2} - 2H < 0$, indicating that $g(\sigma^+, H)$ is concave. By evaluating the first derivative at the boundaries, we find $\frac{\partial g(\sigma^+, H)}{\partial \sigma^+}\big|_{\sigma^+ \to 0} \to H^2 + H > 0$ and $\frac{\partial g(\sigma^+, H)}{\partial \sigma^+}\big|_{\sigma^+ = 1} = -H < 0$, which shows that $g(\sigma^+, H)$ first increases monotonically, reaches a maximum at some point $\sigma^\star$, and then decreases monotonically. Furthermore, since $g(\sigma^+ \to 0, H) \to -H < 0$ and $g(\sigma^+ = 1, H) = 0$, there exists $0 < \sigma^0 < 1$ such that $g(\sigma^0, H) = 0$. Thus, when $\sigma^0 \lesssim \min\{\sigma^+, \sigma^-\} \lesssim 1$, we have $T = H$. Otherwise, $T = \min\left\{\frac{(H\sigma^+ - 1 + (1-\sigma^+)^H)}{(\sigma^+)^2}, \frac{(H\sigma^- - 1 + (1-\sigma^-)^H)}{(\sigma^-)^2}\right\}$.

## C  Auxiliary lemmas for Theorem 1

### C.1  Proof of Lemma 1

In this part, we prove Lemma 1 produced in Algorithm 1.

Before next proof, we clarify the independent property. Let us examine two distinct data-generation mechanisms, where a sample transition quadruple $(s, a, b, h, s')$ represents a transition from state $s$ with actions $(a, b)$ to state $s'$ at step $h$.

**Step 1: Augmenting $\mathcal{D}^{\mathrm{t}}$ to Create $\mathcal{D}^{\mathrm{t,a}}$.** To construct the augmented dataset $\mathcal{D}^{\mathrm{t,a}}$, for each $(s, h) \in \mathcal{S} \times [H]$, we proceed as follows: (i). Include in $\mathcal{D}^{\mathrm{t,a}}$ all $N_h^{\mathrm{t}}(s)$ sample transitions in $\mathcal{D}^{\mathrm{t}}$ originating from state $s$ at step $h$. (ii). If $N_h^{\mathrm{t}}(s) > N_h^{\mathrm{m}}(s)$, supplement $\mathcal{D}^{\mathrm{t,a}}$ with an additional $N_h^{\mathrm{t}}(s) - N_h^{\mathrm{m}}(s)$ independent sample transitions $\left\{\left(s, a_{h,s}^{(i)}, b_{h,s}^{(i)}, h, s_{h,s}'^{(i)}\right)\right\}$, generated as follows:

$$a_{h,s}^{(i)} \overset{\mathrm{i.i.d.}}{\sim} \mu_h^{\mathrm{b}}(\cdot|s), \quad b_{h,s}^{(i)} \overset{\mathrm{i.i.d.}}{\sim} \nu_h^{\mathrm{b}}(\cdot|s), \quad s_{h,s}'^{(i)} \overset{\mathrm{i.i.d.}}{\sim} P_h\left(\cdot|s, a_{h,s}^{(i)}, b_{h,s}^{(i)}\right), \quad N_h^{\mathrm{m}}(s) < i \leq N_h^{\mathrm{t}}(s).$$

**Step 2: Constructing $\mathcal{D}^{\mathrm{iid}}$.** For each $(s, h) \in \mathcal{S} \times [H]$, generate $N_h^{\mathrm{t}}(s)$ independent sample transitions $\left\{\left(s, a_{h,s}^{(i)}, b_{h,s}^{(i)}, h, s_{h,s}'^{(i)}\right)\right\}$ as follows:

$$a_{h,s}^{(i)} \overset{\mathrm{i.i.d.}}{\sim} \mu_h^{\mathrm{b}}(\cdot|s), \quad b_{h,s}^{(i)} \overset{\mathrm{i.i.d.}}{\sim} \nu_h^{\mathrm{b}}(\cdot|s), \quad s_{h,s}'^{(i)} \overset{\mathrm{i.i.d.}}{\sim} P_h\left(\cdot|s, a, b\right), \quad 1 \leq i \leq N_h^{\mathrm{t}}(s).$$

The resulting dataset is defined as:

$$\mathcal{D}^{\mathrm{iid}} := \left\{\left(s, a_{h,s}^{(i)}, b_{h,s}^{(i)}, h, s_{h,s}'^{(i)}\right) \mid s \in \mathcal{S}, 1 \leq h \leq H, 1 \leq i \leq N_h^{\mathrm{t}}(s)\right\}.$$

**Establishing independent property.** The dataset $\mathcal{D}^{\mathrm{t,a}}$ deviates from $\mathcal{D}^{\mathrm{t}}$ only when $N_h^{\mathrm{t}}(s) > N_h^{\mathrm{m}}(s)$ holds. This augmentation ensures that $\mathcal{D}^{\mathrm{t,a}}$ contains precisely $N_h^{\mathrm{t}}(s)$ sample transitions from state $s$ at step $h$. Both $\mathcal{D}^{\mathrm{t,a}}$ and $\mathcal{D}^{\mathrm{iid}}$ comprise exactly $N_h^{\mathrm{t}}(s)$ sample transitions from state $s$ at step $h$, with $\{N_h^{\mathrm{t}}(s)\}$ being statistically independent of the randomness in sample generation. Consequently, given $\{N_h^{\mathrm{t}}(s)\}$, the sample transitions in $\mathcal{D}^{\mathrm{t,a}}$ across different steps are statistically independent. As a result, both $\mathcal{D}^{\mathrm{t}}$ and $\mathcal{D}^{\mathrm{iid}}$ can be regarded as collections of independent samples.

Next, we begin to prove $N_h^{\mathrm{t}}(s) \leq N_h^{\mathrm{m}}(s)$. Since $\mathcal{D}^{\mathrm{a}}$ is generated by half of the sample trajectories in line 2 in Algorithm 1, there is

$$N_h^{\mathrm{a}}(s) = \sum_{k=K/2+1}^{K} \mathbb{1}\{s_h^k = s\}$$

for each $s \in \mathcal{S}$ and $1 \leq h \leq H$. Thus, we can view $N_h^{\mathrm{a}}(s)$ as the sum of $K/2$ independent Bernoulli random variables with mean $d_h^{\mu^{\mathrm{n}},\nu^{\mathrm{n}}}(s)$. According to the Bernstein inequality and the union bound, we derive

$$\mathbb{P}\left\{\exists (s,h) \in \mathcal{S} \times [H] : \left| N_h^{\mathrm{a}}(s) - \frac{K}{2} d_h^{\mu^{\mathrm{n}},\nu^{\mathrm{n}}}(s) \right| \geq N_0 \right\}$$

$$\leq \sum_{s \in \mathcal{S}, h \in [H]} \mathbb{P}\left\{ \left| N_h^{\mathrm{a}}(s) - \frac{K}{2} d_h^{\mu^{\mathrm{n}},\nu^{\mathrm{n}}}(s) \right| \geq N_0 \right\}$$

$$\leq 2HS \exp\left( -\frac{N_0^2/2}{N_{h,s} + N_0/3} \right), \quad \forall N_0 \geq 0,$$

where

$$N_{h,s} := \frac{K}{2} \mathsf{Var}\big(\mathbb{1}\{s_h^t = s\}\big) = \frac{K d_h^{\mu^{\mathrm{n}},\nu^{\mathrm{n}}}(s)\big(1 - d_h^{\mu^{\mathrm{n}},\nu^{\mathrm{n}}}(s)\big)}{2} \leq \frac{K d_h^{\mu^{\mathrm{n}},\nu^{\mathrm{n}}}(s)}{2}.$$

Therefore, with probability at least $1 - 2\delta$, we yield that: $\forall s \in \mathcal{S}$ and $\forall 1 \leq h \leq H$,

$$\left| N_h^{\mathrm{a}}(s) - \frac{K}{2} d_h^{\mu^{\mathrm{n}},\nu^{\mathrm{n}}}(s) \right| \leq \sqrt{4 N_{h,s} \log \frac{HS}{\delta}} + \frac{2}{3}\log\frac{HS}{\delta} \leq \sqrt{2K d_h^{\mu^{\mathrm{n}},\nu^{\mathrm{n}}}(s) \log\frac{HS}{\delta}} + \log\frac{HS}{\delta}. \tag{68}$$

As generated by the same way between $\mathcal{D}^{\mathrm{m}}$ and $\mathcal{D}^{\mathrm{a}}$, we similarly obtain that with probability exceeding $1 - 2\delta$, $\forall s \in \mathcal{S}$ and $\forall 1 \leq h \leq H$,

$$\left| N_h^{\mathrm{m}}(s) - \frac{K}{2} d_h^{\mu^{\mathrm{n}},\nu^{\mathrm{n}}}(s) \right| \leq \sqrt{2K d_h^{\mu^{\mathrm{n}},\nu^{\mathrm{n}}}(s) \log\frac{HS}{\delta}} + \log\frac{HS}{\delta}. \tag{69}$$

Combining (68) and (69), there is

$$|N_h^{\mathrm{m}}(s) - N_h^{\mathrm{a}}(s)| \leq 2\sqrt{2K d_h^{\mu^{\mathrm{n}},\nu^{\mathrm{n}}}(s) \log\frac{HS}{\delta}} + 2\log\frac{HS}{\delta} \tag{70}$$

for all $s \in \mathcal{S}$ and $1 \leq h \leq H$.

Now, we complete the proof of $N_h^{\mathrm{t}}(s) \leq N_h^{\mathrm{m}}(s)$, which can be divided into two cases.

*The first case is $N_h^{\mathrm{a}}(s) \leq 100\log\frac{HS}{\delta}$.* According to the definition in (15), we obtain

$$N_h^{\mathrm{t}}(s) = \max\left\{ N_h^{\mathrm{a}}(s) - 10\sqrt{N_h^{\mathrm{a}}(s) \log\frac{HS}{\delta}}, \, 0 \right\} = 0 \leq N_h^{\mathrm{m}}(s). \tag{71}$$

*The second case is $N_h^{\mathrm{a}}(s) > 100\log\frac{HS}{\delta}$.* Followed by (68), we obtain

$$\frac{K}{2} d_h^{\mu^{\mathrm{n}},\nu^{\mathrm{n}}}(s) + \sqrt{2K d_h^{\mu^{\mathrm{n}},\nu^{\mathrm{n}}}(s) \log\frac{HS}{\delta}} + \log\frac{HS}{\delta} \geq N_h^{\mathrm{a}}(s),$$

leading to

$$K d_h^{\mu^{\mathrm{n}},\nu^{\mathrm{n}}}(s) \geq (9\sqrt{2})^2 \log\frac{HS}{\delta} \geq 100\log\frac{HS}{\delta}. \tag{72}$$

Thus, we take (72) back to (68) and derive

$$
N_h^{\mathsf{a}}(s) \geq \frac{K}{2} d_h^{\mu^{\mathsf{n}}, \nu^{\mathsf{n}}}(s) - \sqrt{2K d_h^{\mu^{\mathsf{n}}, \nu^{\mathsf{n}}}(s) \log \frac{HS}{\delta}} - \log \frac{HS}{\delta} \geq \frac{K}{4} d_h^{\mu^{\mathsf{n}}, \nu^{\mathsf{n}}}(s). \tag{73}
$$

Consequently, in the case of $N_h^{\mathsf{a}}(s) > 100 \log \frac{HS}{\delta}$, we have

$$
\begin{aligned}
N_h^{\mathsf{t}}(s) &= \max \left\{ N_h^{\mathsf{a}}(s) - 10 \sqrt{N_h^{\mathsf{a}}(s) \log \frac{HS}{\delta}}, \, 0 \right\} \\
&= N_h^{\mathsf{a}}(s) - 10 \sqrt{N_h^{\mathsf{a}}(s) \log \frac{HS}{\delta}} \\
&\overset{(i)}{\leq} N_h^{\mathsf{a}}(s) - 5 \sqrt{K d_h^{\mu^{\mathsf{n}}, \nu^{\mathsf{n}}}(s) \log \frac{HS}{\delta}} \\
&\overset{(ii)}{\leq} N_h^{\mathsf{a}}(s) - \left\{ 2 \sqrt{2K d_h^{\mu^{\mathsf{n}}, \nu^{\mathsf{n}}}(s) \log \frac{HS}{\delta}} + 2 \log \frac{HS}{\delta} \right\} \overset{(iii)}{\leq} N_h^{\mathsf{m}}(s),
\end{aligned} \tag{74}
$$

where (i) holds under condition (73), (ii) exists under the condition (72), and (iii) comes from the inequality (70) with probability at least $1 - 2\delta$.

Combining the results in (71) and (74) together, we establish $N_h^{\mathsf{t}}(s) \leq N_h^{\mathsf{m}}(s)$.

Now, we claim the following bound with proof in Appendix C.2: $\forall (s, a, b, h) \in \mathcal{S} \times \mathcal{A} \times \mathcal{B} \times [H]$, with probability exceeding $1 - 2\delta$,

$$
N_h^{\mathsf{t}}(s, a, b) \geq N_h^{\mathsf{t}}(s) \mu_h^{\mathsf{n}}(a \mid s) \nu_h^{\mathsf{n}}(b \mid s) - \sqrt{4 N_h^{\mathsf{t}}(s) \mu_h^{\mathsf{n}}(a \mid s) \nu_h^{\mathsf{n}}(b \mid s) \log \frac{KH}{\delta}} - \log \frac{KH}{\delta}. \tag{75}
$$

Armed with the fact $N_h^{\mathsf{t}}(s) \leq N_h^{\mathsf{m}}(s)$ and claim (75), we start to prove (16). In the following, we discuss two cases, i.e., $K d_h^{\mu^{\mathsf{n}}, \nu^{\mathsf{n}}}(s, a, b) \leq 1600 \log \frac{KH}{\delta}$ and $K d_h^{\mu^{\mathsf{n}}, \nu^{\mathsf{n}}}(s, a, b) > 1600 \log \frac{KH}{\delta}$.

For the first case of $K d_h^{\mu^{\mathsf{n}}, \nu^{\mathsf{n}}}(s, a) \leq 1600 \log \frac{KH}{\delta}$, we can easily classified that

$$
\frac{K}{8} d_h^{\mu^{\mathsf{n}}, \nu^{\mathsf{n}}}(s, a) - 5 \sqrt{K d_h^{\mu^{\mathsf{n}}, \nu^{\mathsf{n}}}(s, a) \log \frac{KH}{\delta}} \leq 0 \leq N_h^{\mathsf{t}}(s, a). \tag{76}
$$

For the second case of $K d_h^{\mu^{\mathsf{n}}, \nu^{\mathsf{n}}}(s, a, b) = K d_h^{\mu^{\mathsf{n}}, \nu^{\mathsf{n}}}(s) \mu_h^{\mathsf{n}}(a \mid s) \nu_h^{\mathsf{n}}(b \mid s) > 1600 \log \frac{KH}{\delta}$, we obtain

$$
N_h^{\mathsf{a}}(s) \geq \frac{K}{4} d_h^{\mu^{\mathsf{n}}, \nu^{\mathsf{n}}}(s) \geq 400 \log \frac{KH}{\delta}, \tag{77}
$$

which is derived by the same line of (73) with slight modification. The property in (77) and the definition of $N_h^{\mathsf{t}}(s)$ together yield

$$
\begin{aligned}
N_h^{\mathsf{t}}(s) &\geq N_h^{\mathsf{a}}(s) - 10 \sqrt{N_h^{\mathsf{a}}(s) \log \frac{KH}{\delta}} \\
&\geq \frac{K}{4} d_h^{\mu^{\mathsf{n}}, \nu^{\mathsf{n}}}(s) - 10 \sqrt{\frac{K}{4} d_h^{\mu^{\mathsf{n}}, \nu^{\mathsf{n}}}(s) \log \frac{KH}{\delta}} \geq \frac{K}{8} d_h^{\mu^{\mathsf{n}}, \nu^{\mathsf{n}}}(s).
\end{aligned}
$$

As a consequent,

$$
N_h^{\mathsf{t}}(s) \mu_h^{\mathsf{n}}(a \mid s) \nu_h^{\mathsf{n}}(b \mid s) \geq \frac{K}{8} d_h^{\mu^{\mathsf{n}}, \nu^{\mathsf{n}}}(s) \mu_h^{\mathsf{n}}(a \mid s) \nu_h^{\mathsf{n}}(b \mid s) \tag{78}
$$

$$
= \frac{K}{8} d_h^{\mu^{\mathsf{n}}, \nu^{\mathsf{n}}}(s, a, b) \geq 200 \log \frac{KH}{\delta}, \tag{79}
$$

where the last inequality holds due to the assumption of the second case. Taking the lower bound (78) with (75) together, there is

$$
\begin{aligned}
N_h^{\mathsf{t}}(s, a, b) &\geq \frac{K}{8} d_h^{\mu^{\mathsf{n}}, \nu^{\mathsf{n}}}(s, a, b) - \sqrt{\frac{K}{2} d_h^{\mu^{\mathsf{n}}, \nu^{\mathsf{n}}}(s, a, b) \log \frac{KH}{\delta}} - \log \frac{KH}{\delta} \\
&\geq \frac{K}{8} d_h^{\mu^{\mathsf{n}}, \nu^{\mathsf{n}}}(s, a, b) - 2 \sqrt{K d_h^{\mu^{\mathsf{n}}, \nu^{\mathsf{n}}}(s, a, b) \log \frac{KH}{\delta}}.
\end{aligned}
$$

Putting the result above and (76) together, according to the claim (75), we can finally complete the proof of Lemma 1.

## C.2 Proof of claim (75).

To prove claim (75), we analyze two cases, i.e., $N_h^{\mathsf{t}}(s)\mu_h^{\mathsf{n}}(a\,|\,s)\nu_h^{\mathsf{n}}(b\,|\,s) \leq 4\log\frac{KH}{\delta}$ and $N_h^{\mathsf{t}}(s)\mu_h^{\mathsf{n}}(a\,|\,s)\nu_h^{\mathsf{n}}(b\,|\,s) > 4\log\frac{KH}{\delta}$.

For the first case of $N_h^{\mathsf{t}}(s)\mu_h^{\mathsf{n}}(a\,|\,s)\nu_h^{\mathsf{n}}(b\,|\,s) \leq 4\log\frac{KH}{\delta}$, we conclude the right-hand side of (75) is negative, leading to the claim (75).

For the second case of $N_h^{\mathsf{t}}(s)\mu_h^{\mathsf{n}}(a\,|\,s)\nu_h^{\mathsf{n}}(b\,|\,s) > 4\log\frac{KH}{\delta}$, we compose a special set $\mathcal{D}^{\mathsf{l}}$ as

$$\mathcal{D}^{\mathsf{l}} := \left\{(s,a,b,h) \in \mathcal{S}\times\mathcal{A}\times\mathcal{B}\times[H] \;\Big|\; N_h^{\mathsf{t}}(s)\mu_h^{\mathsf{n}}(a\,|\,s)\nu_h^{\mathsf{n}}(b\,|\,s) > 4\log\frac{KH}{\delta}\right\}. \tag{80}$$

With the fact of

$$\sum_{(s,a,b,h)\in\mathcal{S}\times\mathcal{A}\times\mathcal{B}\times[H]} N_h^{\mathsf{t}}(s)\mu_h^{\mathsf{n}}(a\,|\,s)\nu_h^{\mathsf{n}}(b\,|\,s) = \sum_{(s,h)\in\mathcal{S}\times[H]} N_h^{\mathsf{t}}(s) \sum_{(a,b)\in\mathcal{A}\times\mathcal{B}} \mu_h^{\mathsf{n}}(a\,|\,s)\nu_h^{\mathsf{n}}(b\,|\,s)$$

$$= \sum_{(s,h)\in\mathcal{S}\times[H]} N_h^{\mathsf{t}}(s) \leq \sum_{(s,h)\in\mathcal{S}\times[H]} N_h^{\mathsf{a}}(s) = \frac{KH}{2},$$

the cardinality of $\mathcal{D}^{\mathsf{l}}$ can be bounded as:

$$\left|\mathcal{D}^{\mathsf{l}}\right| < \frac{\sum_{(s,a,b,h)} N_h^{\mathsf{t}}(s)\mu_h^{\mathsf{n}}(a\,|\,s)\nu_h^{\mathsf{n}}(b\,|\,s)}{4\log\frac{KH}{\delta}} \leq KH/2. \tag{81}$$

Besides, we can view $N_h^{\mathsf{t}}(s,a)$ as the sum of $N_h^{\mathsf{t}}(s)$ independent Bernoulli random variables with mean $\mu_h^{\mathsf{n}}(a\,|\,s)\nu_h^{\mathsf{n}}(b\,|\,s)$, which holds due to $N_h^{\mathsf{t}}(s) \leq N_h^{\mathsf{m}}(s)$ with high probability and condition on $N_h^{\mathsf{t}}(s), N_h^{\mathsf{m}}(s)$. Analogous to (68) based on the condition $N_h^{\mathsf{t}}(s) \leq N_h^{\mathsf{m}}(s)$, we can repeat the Bernstein-type argument and obtain that for any fixed triple $(s,a,b,h)$, with probability at least $1 - 2\delta/(KH)$,

$$N_h^{\mathsf{t}}(s,a,b) \geq N_h^{\mathsf{t}}(s)\mu_h^{\mathsf{n}}(a\,|\,s)\nu_h^{\mathsf{n}}(b\,|\,s)$$

$$- \sqrt{4N_h^{\mathsf{t}}(s)\mu_h^{\mathsf{n}}(a\,|\,s)\nu_h^{\mathsf{n}}(b\,|\,s)\log\frac{KH}{\delta}} - \log\frac{KH}{\delta}. \tag{82}$$

Therefore, with probability exceeding $1 - \delta$, (82) holds for all $(s,a,b,h) \in \mathcal{D}^{\mathsf{l}}$ by utilizing the union bound of (81) over all $(s,a,b,h) \in \mathcal{D}^{\mathsf{l}}$.

Consequently, combining the results above under two cases, we derive that the property (75) holds for all $(s,a,b,h) \in \mathcal{S}\times\mathcal{A}\times\mathcal{B}\times[H]$ with probability at least $1 - \delta$.

## C.3 Proof of Lemma 3

We prove Lemma 3 similar to the proof of claim 1 by Yan et al. (2024), which is separated into two parts as follows.

**Part 1: proof of inequality (34).** According to the definition in (18), for any fixed value vector $V$ independent from $\widehat{P}^0_{h,s,a,b}$, we have

$$\left| \inf_{P \in \mathcal{U}^{\sigma^+}(\widehat{P}^0_{h,s,a,b})} PV - \inf_{P \in \mathcal{U}^{\sigma^+}(P^0_{h,s,a,b})} PV \right|$$

$$= \left| \max_{\alpha \in [\min_s V(s), \max_s V(s)]} \left\{ \widehat{P}^0_{h,s,a,b} [V]_\alpha - \sigma^+ \left( \alpha - \min_{s'} [V]_\alpha (s') \right) \right\} \right.$$

$$\left. - \max_{\alpha \in [\min_s V(s), \max_s V(s)]} \left\{ P^0_{h,s,a,b} [V]_\alpha - \sigma^+ \left( \alpha - \min_{s'} [V]_\alpha (s') \right) \right\} \right|$$

$$\leq \max_{\alpha \in [\min_s V(s), \max_s V(s)]} \left| \widehat{P}^0_{h,s,a,b} [V]_\alpha - P^0_{h,s,a,b} [V]_\alpha \right|$$

$$\leq \max_{\alpha \in [0,H]} \left| \widehat{P}^0_{h,s,a,b} [V]_\alpha - P^0_{h,s,a,b} [V]_\alpha \right|, \tag{83}$$

where the last inequality exists due to the fact that the maximum operator is 1-Lipschitz.

According to the definition of empirical transition kernel $\widehat{P}^0_{h,s,a,b}$, we get

$$\left( \widehat{P}^0_{h,s,a,b} - P^0_{h,s,a,b} \right) [V]_\alpha$$

$$= \sum_{s' \in \mathcal{S}} [V(s')]_\alpha \underbrace{\left[ \frac{\sum_{i=1}^N \mathbb{1}\{h_i = h, s_i = s, a_i = a, b_i = b, s_i' = s'\}}{N_h(s,a,b)} - P^0_h(s' \mid s,a,b) \right]}_{=:X_{s'}}$$

as a sum of independent random variables. Based on the relationship between $P^0_{h,s,a,b}$ and $\widehat{P}^0_{h,s,a,b}$, we verify $\mathbb{E}[X_{s'}] = 0$ and $|X_{s'}| \leq H$ for all $s' \in \mathcal{S}$. Therefore, with probability exceeding $1 - \delta$ and for some universal constant $C_4 > 0$, under the Bernstein inequality (Vershynin, 2018, Theorem 2.8.4), we have

$$\left( \widehat{P}^0_{h,s,a,b} - P^0_{h,s,a,b} \right) [V]_\alpha \leq C_4 \sqrt{\frac{1}{N_h(s,a,b)} \mathsf{Var}_{P^0_{h,s,a,b}} ([V]_\alpha) \log \frac{KH}{\delta}} + \frac{C_4 H \log \frac{KH}{\delta}}{N_h(s,a,b)}$$

$$\leq C_4 \sqrt{\frac{1}{N_h(s,a,b)} \mathsf{Var}_{P^0_{h,s,a,b}} (V) \log \frac{KH}{\delta}} + \frac{C_4 H \log \frac{KH}{\delta}}{N_h(s,a,b)}, \tag{84}$$

where the last inequality comes from the definition of $[V]_\alpha$ in (19).

Let $\overline{V} := V - \left( P^0_{h,s,a,b} V \right) \mathbf{1}$, we have

$$\mathsf{Var}_{P^0_{h,s,a,b}} (V) = P^0_{h,s,a,b} \left( \overline{V} \circ \overline{V} \right)$$

$$= \widehat{P}^0_{h,s,a,b} \left( \overline{V} \circ \overline{V} \right) + \left( P^0_{h,s,a,b} - \widehat{P}^0_{h,s,a,b} \right) \left( \overline{V} \circ \overline{V} \right)$$

$$= \mathsf{Var}_{\widehat{P}^0_{h,s,a,b}} (V) + \left[ \left( P^0_{h,s,a,b} - \widehat{P}^0_{h,s,a,b} \right) V \right]^2 + \left( P^0_{h,s,a,b} - \widehat{P}^0_{h,s,a,b} \right) \left( \overline{V} \circ \overline{V} \right), \tag{85}$$

where the last equation holds since

$$\widehat{P}^0_{h,s,a,b} \left( \overline{V} \circ \overline{V} \right) = \widehat{P}^0_{h,s,a,b} \left( \left[ V - \left( P^0_{h,s,a,b} V \right) \mathbf{1} \right] \circ \left[ V - \left( P^0_{h,s,a,b} V \right) \mathbf{1} \right] \right)$$

$$= \widehat{P}^0_{h,s,a,b} (V \circ V) - 2 \left( P^0_{h,s,a,b} V \right) \left( \widehat{P}^0_{h,s,a,b} V \right) + \left( P^0_{h,s,a,b} V \right)^2$$

$$= \widehat{P}^0_{h,s,a,b} \left( \left[ V - \left( \widehat{P}^0_{h,s,a,b} V \right) \mathbf{1} \right] \circ \left[ V - \left( \widehat{P}^0_{h,s,a,b} V \right) \mathbf{1} \right] \right) + \left( \widehat{P}^0_{h,s,a,b} V \right)^2$$

$$- 2 \left( P^0_{h,s,a,b} V \right) \left( \widehat{P}^0_{h,s,a,b} V \right) + \left( P^0_{h,s,a,b} V \right)^2$$

$$= \mathsf{Var}_{\widehat{P}^0_{h,s,a,b}} (V) + \left[ \left( P^0_{h,s,a,b} - \widehat{P}^0_{h,s,a,b} \right) V \right]^2.$$

Analogous to (84), with probability exceeding $1 - \delta$, there is

$$\left|\left(\widehat{P}^0_{h,s,a,b} - P^0_{h,s,a,b}\right)\left(\overline{V} \circ \overline{V}\right)\right| \leq C_4 \sqrt{\frac{1}{N_h\left(s,a,b\right)} \mathsf{Var}_{P^0_{h,s,a,b}}\left(\overline{V} \circ \overline{V}\right) \log \frac{KH}{\delta}} + \frac{C_4 H^2 \log \frac{KH}{\delta}}{N_h\left(s,a,b\right)}$$

$$\leq C_4 \sqrt{\frac{H^2}{N_h\left(s,a,b\right)} \mathsf{Var}_{P^0_{h,s,a,b}}\left(V\right) \log \frac{KH}{\delta}} + \frac{C_4 H^2 \log \frac{KH}{\delta}}{N_h\left(s,a,b\right)}, \tag{86}$$

where the last inequation comes from the fact that

$$\mathsf{Var}_{P^0_{h,s,a,b}}\left(\overline{V} \circ \overline{V}\right) \leq P^0_{h,s,a,b}\left(\overline{V} \circ \overline{V} \circ \overline{V} \circ \overline{V}\right) \leq H^2 P^0_{h,s,a,b}\left(\overline{V} \circ \overline{V}\right) = H^2 \mathsf{Var}_{P^0_{h,s,a,b}}\left(V\right).$$

Under the result in (86), we bound (85) further as:

$$\mathsf{Var}_{P^0_{h,s,a,b}}\left(V\right) \leq \mathsf{Var}_{\widehat{P}^0_{h,s,a,b}}\left(V\right) + \left[\left(P^0_{h,s,a,b} - \widehat{P}^0_{h,s,a,b}\right)V\right]^2$$

$$+ C_4 \sqrt{\frac{H^2 \log \frac{KH}{\delta}}{N_h\left(s,a,b\right)} \mathsf{Var}_{P^0_{h,s,a,b}}\left(V\right)} + \frac{C_4 H^2 \log \frac{KH}{\delta}}{N_h\left(s,a,b\right)}$$

$$\leq \mathsf{Var}_{\widehat{P}^0_{h,s,a,b}}\left(V\right) + \left[\left(P^0_{h,s,a,b} - \widehat{P}^0_{h,s,a,b}\right)V\right]^2 + \frac{C_4 H^2 \log \frac{KH}{\delta}}{N_h\left(s,a,b\right)}$$

$$+ \frac{1}{2}\mathsf{Var}_{P^0_{h,s,a,b}}\left(V\right) + \frac{C_4^2 H^2 \log \frac{KH}{\delta}}{2 N_h\left(s,a,b\right)},$$

where the last relation holds due to the AM-GM inequality. Therefore, we obtain

$$\mathsf{Var}_{P^0_{h,s,a,b}}\left(V\right) \leq 2\mathsf{Var}_{\widehat{P}^0_{h,s,a,b}}\left(V\right) + 2\left[\left(P^0_{h,s,a,b} - \widehat{P}^0_{h,s,a,b}\right)V\right]^2 + \frac{\left(C_4^2 + 2C_4\right) H^2 \log \frac{KH}{\delta}}{N_h\left(s,a,b\right)}. \tag{87}$$

Combining (87) and (84), we derive

$$\left|\left(\widehat{P}^0_{h,s,a,b} - P^0_{h,s,a,b}\right)V\right| \leq \sqrt{\frac{2C_4^2}{N_h\left(s,a,b\right)} \mathsf{Var}_{\widehat{P}^0_{h,s,a,b}}\left(V\right) \log \frac{KH}{\delta}} + \frac{\sqrt{C_4^2\left(C_4^2 + 2C_4\right) H \log \frac{KH}{\delta}}}{N_h\left(s,a,b\right)}$$

$$+ \sqrt{\frac{2C_4^2}{N_h\left(s,a,b\right)} \log \frac{KH}{\delta}} \left|\left(\widehat{P}^0_{h,s,a,b} - P^0_{h,s,a,b}\right)V\right| + \frac{C_4 H \log \frac{KH}{\delta}}{N_h\left(s,a,b\right)}. \tag{88}$$

In the following, we consider two cases, i.e., $N_h\left(s,a,b\right) \leq \frac{1}{8C_4^2} \log \frac{KH}{\delta}$ and $N_h\left(s,a,b\right) > \frac{1}{8C_4^2} \log \frac{KH}{\delta}$.

For the first case of $N_h\left(s,a,b\right) \leq \frac{1}{8C_4^2} \log \frac{KH}{\delta}$, (34) is valid since

$$\left|\inf_{P \in \mathcal{U}^{\sigma^+}(\widehat{P}^0_{h,s,a,b})} PV - \inf_{P \in \mathcal{U}^{\sigma^+}(P^0_{h,s,a,b})} PV\right| \leq \max_{\alpha \in [\min_s V(s), \max_s V(s)]} \left|\left(\widehat{P}^0_{h,s,a,b} - P^0_{h,s,a,b}\right)V\right|$$

$$\leq 2H = O\left(\frac{H \log \frac{KH}{\delta}}{N_h\left(s,a,b\right)}\right). \tag{89}$$

For the second case of $N_h\left(s,a,b\right) > \frac{1}{8C_4^2} \log \frac{KH}{\delta}$, we observe from (88) that

$$\left|\left(\widehat{P}^0_{h,s,a,b} - P^0_{h,s,a,b}\right)V\right| \leq \frac{1}{2}\left|\left(\widehat{P}^0_{h,s,a,b} - P^0_{h,s,a,b}\right)V\right| + \sqrt{\frac{2C_4^2}{N_h\left(s,a,b\right)} \mathsf{Var}_{\widehat{P}^0_{h,s,a,b}}\left(V\right) \log \frac{KH}{\delta}}$$

$$+ \frac{C_4 + \sqrt{C_4^2\left(C_4^2 + 2C_4\right)}}{N_h\left(s,a,b\right)} H \log \frac{KH}{\delta}.$$

Rearrange terms above and yield

$$\left|\left(\widehat{P}_{h,s,a,b}^0 - P_{h,s,a,b}^0\right)V\right| \leq \sqrt{\frac{8C_4^2}{N_h(s,a,b)}\mathsf{Var}_{\widehat{P}_{h,s,a,b}^0}(V)\log\frac{KH}{\delta}}$$

$$+ 2H\frac{C_4 + \sqrt{C_4^2\left(C_4^2 + 2C_4\right)}}{N_h(s,a,b)}\log\frac{KH}{\delta}. \tag{90}$$

Putting (83) and (90) together, we get

$$\left|\inf_{P\in\mathcal{U}^{\sigma^+}(\widehat{P}_{h,s,a,b}^0)}PV - \inf_{P\in\mathcal{U}^{\sigma^+}(P_{h,s,a,b}^0)}PV\right| \leq \sqrt{\frac{8C_4^2}{N_h(s,a,b)}\mathsf{Var}_{\widehat{P}_{h,s,a,b}^0}(V)\log\frac{KH}{\delta}}$$

$$+ 2H\frac{C_4 + \sqrt{C_4^2\left(C_4^2 + 2C_4\right)}}{N_h(s,a,b)}\log\frac{KH}{\delta}. \tag{91}$$

Putting the above bounds for two cases together, we conclude the proof of (34).

**Part 2: proof of inequality (35).** In the process of proving inequality (35), we just divide the problem into two cases, i.e., $N_h(s,a,b) < 16C_4^2\log\frac{KH}{\delta}$ and $N_h(s,a,b) \geq 16C_4^2\log\frac{KH}{\delta}$.

For the first case of $N_h(s,a,b) < 16C_4^2\log\frac{KH}{\delta}$, the result (35) is valid since

$$\mathsf{Var}_{\widehat{P}_{h,s,a,b}^0}(V) \leq H^2 = O\left(\frac{H^2\log\frac{KH}{\delta}}{N_h(s,a,b)}\right).$$

For the second case of $N_h(s,a,b) \geq 16C_4^2\log\frac{KH}{\delta}$, there is

$$\mathsf{Var}_{\widehat{P}_{h,s,a,b}^0}(V) \overset{\text{(i)}}{=} \mathsf{Var}_{P_{h,s,a,b}^0}(V) - \left[\left(P_{h,s,a,b}^0 - \widehat{P}_{h,s,a,b}^0\right)V\right]^2 - \left(P_{h,s,a,b}^0 - \widehat{P}_{h,s,a,b}^0\right)\left(\overline{V}\circ\overline{V}\right)$$

$$\overset{\text{(ii)}}{\leq} \mathsf{Var}_{P_{h,s,a,b}^0}(V) + C_4\sqrt{\frac{H^2}{N_h(s,a,b)}\mathsf{Var}_{P_{h,s,a,b}^0}(V)\log\frac{KH}{\delta}} + \frac{C_4 H^2\log\frac{KH}{\delta}}{N_h(s,a,b)}$$

$$\overset{\text{(iii)}}{\leq} 2\mathsf{Var}_{P_{h,s,a,b}^0}(V) + \frac{\left(C_4^2/4 + C_4\right)H^2\log\frac{KH}{\delta}}{N_h(s,a,b)}$$

$$= 2\mathsf{Var}_{P_{h,s,a,b}^0}(V) + O\left(\frac{H^2\log\frac{KH}{\delta}}{N_h(s,a,b)}\right),$$

where (i) comes from (85), (ii) holds due to (86), and (iii) exists under the AM-GM inequality.

Putting the two cases together, we complete the proof of (35). Thus, Lemma 3 is finally proven.

## C.4 PROOF OF LEMMA 4

Assuming that $\widehat{Q}_h^+(s,a,b) \geq Q_h^{\star,\widehat{\nu},\sigma^+}(s,a,b)$ holds, then we can easily obtain $\widehat{V}_h^+(s) \geq V_h^{\star,\nu,\sigma^+}(s)$, since

$$\widehat{V}_h^+(s) = \mathbb{E}_{a\sim\mu_h^+(s),b\sim\nu_h^+(s)}\left[\widehat{Q}_h^+(s,a,b)\right]$$

$$\overset{\text{(i)}}{\geq} \mathbb{E}_{a\sim\mu^\star(s),b\sim\widehat{\nu}(s)}\left[\widehat{Q}_h^+(s,a,b)\right] \geq \mathbb{E}_{a\sim\mu^\star(s),b\sim\widehat{\nu}(s)}\left[Q_h^{\star,\widehat{\nu},\sigma^+}(s,a,b)\right] = V_h^{\star,\widehat{\nu},\sigma^+}(s),$$

where (i) holds due to the fact that $\widehat{\nu} = \nu_h^+$ and $(\mu_h^+, \nu_h^+)$ is the Nash equilibrium of $\widehat{Q}_h^+(s,a,b)$.

Consequently, we just need to verify

$$\widehat{Q}_h^+(s,a,b) \geq Q_h^{\star,\widehat{\nu},\sigma^+}(s,a,b), \tag{92}$$

which is obtained by induction.

It can be easily verified that (92) holds at the base case when $h = H + 1$ under the trivial fact $\widehat{Q}_{H+1}^+(s, a, b) = Q_{H+1}^{\star, \sigma^+}(s, a, b) = 0$.

Suppose that (92) holds for all $(s, a, b) \in \mathcal{S} \times \mathcal{A} \times \mathcal{B}$ at some time step $h \in [H]$ next.

According to the update rule in line 4 in Algorithm 2, (92) exists if $\widehat{Q}_h^+(s, a, b) = H$ because $\widehat{Q}_h^+(s, a, b) = H \geq Q_h^{\star, \widehat{\nu}, \sigma^+}(s, a, b)$.

Besides, in the case of $N_h(s, a, b) = 0$, we have $\beta_h\left(s, a, b, \widehat{V}_{h+1}^+\right) = H$, leading to $\widehat{Q}_h^+(s, a, b) = H \geq Q_h^{\star, \widehat{\nu}, \sigma^+}(s, a, b)$. Otherwise, for $N_h(s, a, b) > 0$, $\widehat{Q}_h^+(s, a, b)$ is updated as

$$
\widehat{Q}_h^+(s, a, b) = \widehat{r}(s, a, b) + \inf_{P \in \mathcal{U}^{\sigma^+}\left(\widehat{P}_{h,s,a,b}^0\right)} P\widehat{V}_{h+1}^+ + \beta_h\left(s, a, b, \widehat{V}_{h+1}^+\right)
$$

$$
\geq \widehat{r}(s, a, b) + \inf_{P \in \mathcal{U}^{\sigma^+}\left(P_{h,s,a,b}^0\right)} P\widehat{V}_{h+1}^+ + \beta_h\left(s, a, b, \widehat{V}_{h+1}^+\right)
$$

$$
\quad - \left| \inf_{P \in \mathcal{U}^{\sigma^+}\left(\widehat{P}_{h,s,a,b}^0\right)} P\widehat{V}_{h+1}^+ - \inf_{P \in \mathcal{U}^{\sigma^+}\left(P_{h,s,a,b}^0\right)} P\widehat{V}_{h+1}^+ \right|
$$

$$
\geq \widehat{r}(s, a, b) + \inf_{P \in \mathcal{U}^{\sigma^+}\left(P_{h,s,a,b}^0\right)} P\widehat{V}_{h+1}^+ + 0
$$

$$
\geq \widehat{r}(s, a, b) + \inf_{P \in \mathcal{U}^{\sigma^+}\left(P_{h,s,a,b}^0\right)} P\widehat{V}_{h+1}^{\star, \widehat{\nu}, \sigma^+} + 0 = Q_h^{\star, \widehat{\nu}, \sigma^+}(s, a, b), \tag{93}
$$

where the second inequality holds due to (36) in Lemma 3 and the last equality comes from the empirical robust Bellman equation (33).

Armed with the case of $h = H + 1$, we complete prove Lemma 4 by induction.

## C.5 PROOF OF LEMMA 5

Following the proof by Shi et al. (2024b, Lemma 3), we bound $\min_{s \in \mathcal{S}} \widehat{V}_h^+(s)$ and $\max_{s \in \mathcal{S}} \widehat{V}_h^+(s)$, respectively. Specifically, we have

$$
\min_{s \in \mathcal{S}} \widehat{V}_h^+(s) = \min_{s \in \mathcal{S}} \mathbb{E}_{(a,b) \sim \mu_h^+ \times \nu_h^+}\left[\widehat{Q}_h^+(s, a, b)\right]
$$

$$
= \min_{s \in \mathcal{S}} \mathbb{E}_{(a,b) \sim \mu_h^+ \times \nu_h^+}\left[\widehat{r}_h(s, a, b) + \inf_{P \in \mathcal{U}^{\sigma^+}\left(\widehat{P}_{h,s,a,b}^0\right)} P\widehat{V}_{h+1}^+ + \beta_h\left(s, a, b, \widehat{V}_{h+1}\right)\right]
$$

$$
\geq 0 + \min_{s \in \mathcal{S}} \widehat{V}_{h+1}^+(s) + 0, \tag{94}
$$

where the middle equality is valid due to the update rule in line 4 in Algorithm 2. Similarly, there is

$$
\max_{s \in \mathcal{S}} \widehat{V}_h^+ = \max_{s \in \mathcal{S}} \mathbb{E}_{(a,b) \sim \mu_h^+ \times \nu_h^+}\left[\widehat{Q}_h^+(s, a, b)\right]
$$

$$
= \max_{s \in \mathcal{S}} \mathbb{E}_{(a,b) \sim \mu_h^+ \times \nu_h^+}\left[\widehat{r}_h(s, a, b) + \inf_{P \in \mathcal{U}^{\sigma^+}\left(\widehat{P}_{h,s,a,b}^0\right)} P\widehat{V}_{h+1}^+ + \beta_h\left(s, a, b, \widehat{V}_{h+1}\right)\right]
$$

$$
\leq 1 + \max_{(s,a,b) \in \mathcal{S} \times \mathcal{A} \times \mathcal{B}} \inf_{P \in \mathcal{U}^{\sigma^+}\left(\widehat{P}_{h,s,a,b}\right)} P\widehat{V}_{h+1}^+ + H. \tag{95}
$$

In order to prove Lemma 5, we here introduce several useful notations. For any $h \in [H]$, there exists at least one state $s_h^\star$ that satisfies $\widehat{V}_h^+(s_h^\star) = \min_{s \in \mathcal{S}} \widehat{V}_h^+(s)$.

Furthermore, for any accessible uncertainty set $\sigma^+ > 0$ and $(s, a, b) \in \mathcal{S} \times \mathcal{A} \times \mathcal{B}$, we define an auxiliary vector $\widehat{P}_{h,s,a,b}' \in \mathbb{R}^S$ by reducing the values of several elements of $\widehat{P}_{h,s,a,b}^0$ strictly, namely,

$$
0 \leq \widehat{P}_{h,s,a,b}' \leq \widehat{P}_{h,s,a,b}^0 \quad \text{and} \quad \sum_{s' \in \mathcal{S}} \widehat{P}_{h,s,a,b}^0(s') - \widehat{P}_{h,s,a,b}'(s') = \left\|\widehat{P}_{h,s,a,b}' - \widehat{P}_{h,s,a,b}^0\right\|_1 = \sigma^+.
$$

$$
\tag{96}
$$

We use $l_{s_h^\star}$ to represent a $S$-dimensional standard basis under $s_h^\star$, we can derive that

$$\frac{1}{2}\left\|\widehat{P}'_{h,s,a,b} + \sigma^+\left[l_{s_h^\star}\right]^\top - \widehat{P}^0_{h,s,a,b}\right\|_1 \leq \frac{1}{2}\left\|\widehat{P}'_{h,s,a,b} - \widehat{P}^0_{h,s,a,b}\right\|_1 + \frac{1}{2}\left\|\sigma^+\left[l_{s_h^\star}\right]^\top\right\|_1 \leq \sigma^+, \quad (97)$$

where the first inequality is valid since the 'distance' function (e.g., TV distance) satisfies the triangle inequality.

Therefore, we can conclude that $\widehat{P}'_{h,s,a,b} + \sigma^+\left[l_{s_h^\star}\right]^\top \in \mathcal{U}^{\sigma^+}(\widehat{P}^0_{h,s,a,b})$ and $\widehat{P}'_{h,s,a,b} + \sigma^+\left[l_{s_h^\star}\right]^\top$ is a distribution vector based on (97), leading to

$$\inf_{P \in \mathcal{U}^{\sigma^+}(\widehat{P}^0_{h,s,a,b})} P\widehat{V}^+_{h+1} \leq \left(\widehat{P}'_{h,s,a,b} + \sigma^+\left[l_{s_{i,h}^\star}\right]^\top\right)\widehat{V}^+_{h+1}$$

$$\leq \left\|\widehat{P}'_{h,s,a,b}\right\|_1\left\|\widehat{V}^+_{h+1}\right\|_\infty + \sigma^+\widehat{V}^+_{h+1}(s_{h+1}^\star)$$

$$\leq \left(1 - \sigma^+\right)\max_{s\in\mathcal{S}}\widehat{V}^+_{h+1}(s) + \sigma^+\min_{s\in\mathcal{S}}\widehat{V}^+_{h+1}(s), \quad (98)$$

where the last inequality holds since

$$\left\|P'_{h,s,a,b}\right\|_1 = \sum_{s'} P'_{h,s,a,b}(s') = -\sum_{s'}\left(P^0_{h,s,a,b}(s') - P'_{h,s,a,b}(s')\right) + \sum_{s'} P^0_{h,s,a,b}(s') = 1 - \sigma^+.$$
$$(99)$$

Putting (98) and (95) together shows

$$\max_{s\in\mathcal{S}}\widehat{V}^+_h(s) \leq 1 + \max_{(s,a,b)\in\mathcal{S}\times\mathcal{A}\times\mathcal{B}}\inf_{P\in\mathcal{U}^{\sigma^+}(P^0_{h,s,a,b})} P\widehat{V}^+_{h+1} + H$$

$$\leq H + 1 + \left(1 - \sigma^+\right)\max_{s\in\mathcal{S}}\widehat{V}^+_{h+1}(s) + \sigma^+\min_{s\in\mathcal{S}}\widehat{V}^+_{h+1}(s). \quad (100)$$

Taking the result (100) with (94), we obtain

$$\max_{s\in\mathcal{S}}\widehat{V}^+_h - \min_{s\in\mathcal{S}}\widehat{V}^+_h$$

$$\leq H + 1 + \left(1 - \sigma^+\right)\max_{s\in\mathcal{S}}\widehat{V}^+_{h+1}(s) + \sigma^+\min_{s\in\mathcal{S}}\widehat{V}^+_{h+1}(s) - \min_{s\in\mathcal{S}}V^+_{h+1}(s)$$

$$= H + 1 + (1 - \sigma^+)\left(\max_{s\in\mathcal{S}}\widehat{V}^+_{h+1}(s) - \min_{s\in\mathcal{S}}\widehat{V}^+_{h+1}(s)\right)$$

$$\leq H + 1 + (1 - \sigma^+)\left[H + 1 + (1 - \sigma^+)\left(\max_{s\in\mathcal{S}}\widehat{V}^+_{h+2}(s) - \min_{s\in\mathcal{S}}\widehat{V}^+_{h+2}(s)\right)\right]$$

$$\leq \cdots \leq \frac{(H+1)\left(1 - (1-\sigma^+)^{H-h}\right)}{\sigma^+}. \quad (101)$$

Combining this result with $\max_{s\in\mathcal{S}}\widehat{V}^+_h(s) - \min_{s\in\mathcal{S}}\widehat{V}^+_h(s) \leq H$, we complete the proof.

### C.6 PROOF OF LEMMA 6

First of all, we introduce some auxiliary values and reward functions to control $\sum_{i=1}^H \sum_{(s,b)\in\mathcal{S}\times\mathcal{B}} d_i^{\mathsf{p},\mu^\mathsf{d},\nu^\star}(s, \mu^\mathsf{d}(s), b)\mathsf{Var}_{P^0_{i,s,\mu^\mathsf{d}(s),b}}(\widehat{V})$ as below: for any time step $i$

- $\widehat{V}_i^\mathsf{m} := \min_{s\in\mathcal{S}}\widehat{V}_i^+(s)$: the minimum value of all the entries in vector $\widehat{V}_i^+$.

- $\widehat{V}_i' := \widehat{V}_i^+ - \widehat{V}_i^\mathsf{m}\mathbf{1}$: truncated value function.

- $\widehat{r}_i^{\mu^\mathsf{d},\nu^\star}(s) = \mathbb{E}_{(a,b)\sim(\mu^\mathsf{d}(s),\nu^\star(s))}\widehat{r}_i(s,a,b)$: average reward function.

- $\widehat{r}_i^\mathsf{m} = r_i^{\mu^\mathsf{d},\nu^\star} + \left(\widehat{V}_{i+1}^\mathsf{m} - \widehat{V}_i^\mathsf{m}\right)\mathbf{1}$: truncated reward function.

Then applying the robust Bellman's consistency equation in (33) gives

$$
\begin{aligned}
\widehat{V}_i' &= \widehat{V}_i^+ - \widehat{V}_i^{\mathsf{m}} 1 \\
&\overset{(i)}{\le} \widehat{r}_i^{\mu^{\mathsf{d}}, \nu^\star} + \widetilde{P}_i^{\inf, \widehat{V}} \widehat{V}_{i+1}^+ + 2\beta_i^{\mu^{\mathsf{d}}, \nu^\star} - \widehat{V}_i^{\mathsf{m}} 1 \\
&= \widehat{r}_i^{\mu^{\mathsf{d}}, \nu^\star} + \widetilde{P}_i^{\inf, \widehat{V}} \widehat{V}_{i+1}^+ + \left( \widehat{V}_{i+1}^{\mathsf{m}} 1 - \widehat{V}_i^{\mathsf{m}} 1 \right) - \widehat{V}_{i+1}^{\mathsf{m}} 1 + 2\beta_i^{\mu^{\mathsf{d}}, \nu^\star} \\
&= \widehat{r}_i^{\mathsf{m}} + \widetilde{P}_i^{\inf, \widehat{V}} \widehat{V}_{i+1}^+ - \widehat{V}_{i+1}^{\mathsf{m}} 1 + 2\beta_i^{\mu^{\mathsf{d}}, \nu^\star} \\
&= \widehat{r}_i^{\mathsf{m}} + \widetilde{P}_i^{\inf, \widehat{V}} \widehat{V}_{i+1}' + 2\beta_i^{\mu^{\mathsf{d}}, \nu^\star},
\end{aligned}
\tag{102}
$$

where (i) follows from the fact that

$$
\begin{aligned}
\widehat{V}_i^+(s) &\le \widehat{r}_i^{\mu^{\mathsf{d}}, \nu^\star}(s) + \mathbb{E}_{(a,b) \sim (\mu^{\mathsf{d}}(s), \nu^\star(s))} \inf_{P \in \mathcal{U}^{\sigma+}\left(\widehat{P}_{i,s,a,b}^0\right)} P\widehat{V}_{i+1}^+ + \beta_i^{\mu^{\mathsf{d}}, \nu^\star}(s) \\
&\overset{(i)}{\le} \widehat{r}_i^{\mu^{\mathsf{d}}, \nu^\star}(s) + \mathbb{E}_{(a,b) \sim (\mu^{\mathsf{d}}(s), \nu^\star(s))} \Bigg[ \inf_{P \in \mathcal{U}^{\sigma+}\left(P_{i,s,a,b}^0\right)} P\widehat{V}_{i+1}^+ \\
&\qquad + \left| \inf_{P \in \mathcal{U}^{\sigma+}\left(\widehat{P}_{i,s,a,b}^0\right)} P\widehat{V}_{i+1}^+ - \inf_{P \in \mathcal{U}^{\sigma+}\left(P_{i,s,a,b}^0\right)} P\widehat{V}_{i+1}^+ \right| \Bigg] + \beta_i^{\mu^{\mathsf{d}}, \nu^\star}(s) \\
&\overset{(ii)}{\le} \widehat{r}_i^{\mu^{\mathsf{d}}, \nu^\star}(s) + \mathbb{E}_{(a,b) \sim (\mu^{\mathsf{d}}(s), \nu^\star(s))} \left[ P_{i,s,a,b}^{\inf, \widehat{V}} \widehat{V}_{i+1}^+ \right] + 2\beta_i^{\mu^{\mathsf{d}}, \nu^\star}(s) \\
&\overset{(iii)}{=} \widehat{r}_i^{\mu^{\mathsf{d}}, \nu^\star}(s) + \widetilde{P}_{i,s}^{\inf, \widehat{V}} \widehat{V}_{i+1}^+ + 2\beta_i^{\mu^{\mathsf{d}}, \nu^\star}(s).
\end{aligned}
\tag{103}
$$

Here, (ii) is valid under the notation

$$
P_{i,s,a,b}^{\inf, \widehat{V}} := \operatorname{argmin}_{P \in \mathcal{U}^{\sigma+}\left(P_{i,s,a,b}^0\right)} P\widehat{V}_{i+1}^+
\tag{104}
$$

and (iii) holds under the notation as $\widetilde{P}_{i,s}^{\inf, \widehat{V}} := \mathbb{E}_{(a,b) \sim (\mu^{\mathsf{d}}(s), \nu^\star(s))} P_{i,s,a,b}^{\inf, \widehat{V}}$ and the sequence as $\widetilde{P}_i^{\inf, V} \in \mathbb{R}^{S \times S}$ Besides, (i) in (103) exists due to (36) in Lemma 3 for $N_i(s,a,b) > 0$ and

$$
\left| \inf_{P \in \mathcal{U}^{\sigma+}\left(P_{i,s,a,b}^0\right)} P\widehat{V}_{i+1}^+ - \inf_{P \in \mathcal{U}^{\sigma+}\left(\widehat{P}_{i,s,a,b}^0\right)} P\widehat{V}_{i+1}^+ \right| \le H = \beta_i^{\mu^{\mathsf{d}}, \nu^\star}(s)
\tag{105}
$$

for $N_i(s,a,b) = 0$.

The above fact leads to

$$
\begin{aligned}
&\mathbb{E}_{(a,b) \sim (\mu^{\mathsf{d}}(s), \nu^\star(s))} \mathsf{Var}_{P_{i,s,a,b}^{\inf, V}}\left( \widehat{V}_{i+1}^+ \right) \\
&\overset{(i)}{=} \mathbb{E}_{(a,b) \sim (\mu^{\mathsf{d}}(s), \nu^\star(s))} \mathsf{Var}_{P_{i,s,a,b}^{\inf, V}}\left( \widehat{V}_{i+1}' \right) \\
&= \mathbb{E}_{(a,b) \sim (\mu^{\mathsf{d}}(s), \nu^\star(s))} \left[ P_{i,s,a,b}^{\inf, V}\left( \widehat{V}_{i+1}' \circ \widehat{V}_{i+1}' \right) - \left( P_{i,s,a,b}^{\inf, V} \widehat{V}_{i+1}' \right) \circ \left( P_{i,s,a,b}^{\inf, V} \widehat{V}_{i+1}' \right) \right] \\
&\le \mathbb{E}_{(a,b) \sim (\mu^{\mathsf{d}}(s), \nu^\star(s))} \left[ P_{i,s,a,b}^{\inf, V}\left( \widehat{V}_{i+1}' \circ \widehat{V}_{i+1}' \right) - \left( P_{i,s,a,b}^{\inf, \widehat{V}} \widehat{V}_{i+1}' \right) \circ \left( P_{i,s,a,b}^{\inf, \widehat{V}} \widehat{V}_{i+1}' \right) \right] \\
&\overset{(ii)}{\le} \widetilde{P}_{i,s}^{\inf, V}\left( \widehat{V}_{i+1}' \circ \widehat{V}_{i+1}' \right) - \left( \widetilde{P}_{i,s}^{\inf, \widehat{V}} \widehat{V}_{i+1}' \right) \circ \left( \widetilde{P}_{i,s}^{\inf, \widehat{V}} \widehat{V}_{i+1}' \right) \\
&= \widetilde{P}_{i,s}^{\inf, V}\left( \widehat{V}_{i+1}' \circ \widehat{V}_{i+1}' \right) - \widehat{V}_i'(s) \circ \widehat{V}_i'(s) + \widehat{V}_i'(s) \circ \widehat{V}_i'(s) - \left( \widetilde{P}_{i,s}^{\inf, \widehat{V}} \widehat{V}_{i+1}' \right) \circ \left( \widetilde{P}_{i,s}^{\inf, \widehat{V}} \widehat{V}_{i+1}' \right) \\
&= \widetilde{P}_{i,s}^{\inf, V}\left( \widehat{V}_{i+1}' \circ \widehat{V}_{i+1}' \right) - \widehat{V}_i'(s) \circ \widehat{V}_i'(s) + \left( \widehat{V}_i'(s) - \left( \widetilde{P}_{i,s}^{\inf, \widehat{V}} \widehat{V}_{i+1}' \right) \right) \circ \left( \widehat{V}_i'(s) + \left( \widetilde{P}_{i,s}^{\inf, \widehat{V}} \widehat{V}_{i+1}' \right) \right) \\
&\overset{(iii)}{\le} \widetilde{P}_{i,s}^{\inf, V}\left( \widehat{V}_{i+1}' \circ \widehat{V}_{i+1}' \right) - \widehat{V}_i'(s) \circ \widehat{V}_i'(s) + \left( \widehat{r}_i^{\mathsf{m}}(s) + 2\beta_h^{\mu^{\mathsf{d}}, \nu^\star}(s) \right) \circ \left( \widehat{V}_i'(s) + \left( \widetilde{P}_{i,s}^{\inf, \widehat{V}} \widehat{V}_{i+1}' \right) \right) \\
&\overset{(iv)}{\le} \widetilde{P}_{i,s}^{\inf, V}\left( \widehat{V}_{i+1}' \circ \widehat{V}_{i+1}' \right) - \widehat{V}_i'(s) \circ \widehat{V}_i'(s) + \left( \left\| \widehat{V}_i' \right\|_\infty + \left\| \widehat{V}_{i+1}' \right\|_\infty \right) \left( 2\beta_i^{\mu^{\mathsf{d}}, \nu^\star}(s) + 1 \right),
\end{aligned}
\tag{106}
$$

where (i) follows from the fact that $\mathrm{Var}_{P_{i,s}^{\mathrm{inf},V}}(V - b\mathbf{1}) = \mathrm{Var}_{P_{i,s}^{\mathrm{inf},V}}(V)$ for any value vector $V \in \mathbb{R}^S$ and scalar $b$, (ii) holds with the fact

$$
\mathbb{E}_{(a,b)\sim(\mu^{\mathsf{d}}(s),\nu^{\star}(s))}\left[\left(P_{i,s,a,b}^{\mathrm{inf},\widehat{V}}\widehat{V}'_{i+1}\right)\circ\left(P_{i,s,a,b}^{\mathrm{inf},\widehat{V}}\widehat{V}'_{i+1}\right)\right]
$$
$$
\geq\mathbb{E}_{(a,b)\sim(\mu^{\mathsf{d}}(s),\nu^{\star}(s))}\left[\left(P_{i,s,a,b}^{\mathrm{inf},\widehat{V}}\widehat{V}'_{i+1}\right)\right]\circ\mathbb{E}_{(a,b)\sim(\mu^{\mathsf{d}}(s),\nu^{\star}(s))}\left[\left(P_{i,s,a,b}^{\mathrm{inf},\widehat{V}}\widehat{V}'_{i+1}\right)\right],
$$

(iv) arises from $\widehat{r}_i^{\mathsf{m}} \leq r_i \leq 1$ due to $\widehat{V}_{i+1}^{\mathsf{m}} - \widehat{V}_i^{\mathsf{m}} \leq 0$ by definition, and (iii) comes from (102). Consequently, combining (48), we arrive at

$$
\sum_{(s,b)\in\mathcal{S}\times\mathcal{B}} d_i^{\mathsf{p},\mu^{\mathsf{d}},\nu^{\star}}(s,\mu^{\mathsf{d}}(s),b)\mathsf{Var}_{P_{i,s,a,b}^{\mathrm{inf},V}}\left(\widehat{V}_{i+1}^+\right)
$$
$$
=\sum_{s\in\mathcal{S}} d_i^{\mathsf{p},\mu^{\mathsf{d}},\nu^{\star}}(s)\mathbb{E}_{(a,b)\sim(\mu^{\mathsf{d}}(s),\nu^{\star}(s))}\mathsf{Var}_{P_{i,s,a,b}^{\mathrm{inf},V}}\left(\widehat{V}_{i+1}^+\right)
$$
$$
\leq\sum_{s\in\mathcal{S}} d_i^{\mathsf{p},\mu^{\mathsf{d}},\nu^{\star}}(s)\left(\widetilde{P}_{i,s}^{\mathrm{inf},V}\left(\widehat{V}'_{i+1}\circ\widehat{V}'_{i+1}\right)-\widehat{V}'_i(s)\circ\widehat{V}'_i(s)+\left(\left\|\widehat{V}'_i\right\|_{\infty}+\left\|\widehat{V}'_{i+1}\right\|_{\infty}\right)\left(2\beta_i^{\mu^{\mathsf{d}},\nu^{\star}}(s)+1\right)\right)
$$
$$
\leq\sum_{s\in\mathcal{S}} d_i^{\mathsf{p},\mu^{\mathsf{d}},\nu^{\star}}(s)\left(\widetilde{P}_{i,s}^{\mathrm{inf},V}\left(\widehat{V}'_{i+1}\circ\widehat{V}'_{i+1}\right)-\widehat{V}'_i(s)\circ\widehat{V}'_i(s)\right)+\left(\left\|\widehat{V}'_i\right\|_{\infty}+\left\|\widehat{V}'_{i+1}\right\|_{\infty}\right)
$$
$$
+2\left(\left\|\widehat{V}'_i\right\|_{\infty}+\left\|\widehat{V}'_{i+1}\right\|_{\infty}\right)\sum_{s\in\mathcal{S}} d_i^{\mathsf{p},\mu^{\mathsf{d}},\nu^{\star}}(s)\beta_i^{\mu^{\mathsf{d}},\nu^{\star}}(s)
$$
$$
=\sum_{s\in\mathcal{S}}\left(d_{i+1}^{\mathsf{p},\mu^{\mathsf{d}},\nu^{\star}}(s)\left(\widehat{V}'_{i+1}(s)\circ\widehat{V}'_{i+1}(s)\right)-d_i^{\mathsf{p},\mu^{\mathsf{d}},\nu^{\star}}(s)\widehat{V}'_i(s)\circ\widehat{V}'_i(s)\right)+\left(\left\|\widehat{V}'_i\right\|_{\infty}+\left\|\widehat{V}'_{i+1}\right\|_{\infty}\right)
$$
$$
+2\left(\left\|\widehat{V}'_i\right\|_{\infty}+\left\|\widehat{V}'_{i+1}\right\|_{\infty}\right)\sum_{s\in\mathcal{S}} d_i^{\mathsf{p},\mu^{\mathsf{d}},\nu^{\star}}(s)\beta_i^{\mu^{\mathsf{d}},\nu^{\star}}(s)
$$
$$
=\sum_{s\in\mathcal{S}}\left(d_{i+1}^{\mathsf{p},\mu^{\mathsf{d}},\nu^{\star}}(s)\left(\widehat{V}'_{i+1}(s)\circ\widehat{V}'_{i+1}(s)\right)-d_i^{\mathsf{p},\mu^{\mathsf{d}},\nu^{\star}}(s)\widehat{V}'_i(s)\circ\widehat{V}'_i(s)\right)+\left(\left\|\widehat{V}'_i\right\|_{\infty}+\left\|\widehat{V}'_{i+1}\right\|_{\infty}\right)
$$
$$
+2\left(\left\|\widehat{V}'_i\right\|_{\infty}+\left\|\widehat{V}'_{i+1}\right\|_{\infty}\right)\sum_{(s,b)\in\mathcal{S}\times\mathcal{B}} d_i^{\mathsf{p},\mu^{\mathsf{d}},\nu^{\star}}(s,\mu^{\mathsf{d}}(s),b)\beta_i(s,\mu^{\mathsf{d}}(s),b,\widehat{V}). \tag{107}
$$

Besides, under TV distance, we have

$$
\left|\mathsf{Var}_{P_{i,s,a,b}^0}\left(\widehat{V}_{i+1}^+\right)-\mathsf{Var}_{P_{i,s,a,b}^{\mathrm{inf},V}}\left(\widehat{V}_{i+1}^+\right)\right|=\left|\mathsf{Var}_{P_{i,s,a,b}^0}\left(\widehat{V}'_{i+1}\right)-\mathsf{Var}_{P_{i,s,a,b}^{\mathrm{inf},V}}\left(\widehat{V}'_{i+1}\right)\right|
$$
$$
\leq\left\|P_{i,s,a,b}^0-P_{i,s,a,b}^{\mathrm{inf},V}\right\|_1\left\|\widehat{V}'_{i+1}\right\|_{\infty}^2
$$
$$
\leq\sigma^+\left\|\widehat{V}'_{i+1}\right\|_{\infty}^2\leq(H+1)\left\|\widehat{V}'_{i+1}\right\|_{\infty}, \tag{108}
$$

where the last inequality comes from Lemma 5.

Therefore, we derive

$$\sum_{i=1}^{H} \sum_{(s,b)\in\mathcal{S}\times\mathcal{B}} d_i^{\mathsf{p},\mu^{\mathsf{d}},\nu^{\star}}(s,\mu^{\mathsf{d}}(s),b)\mathsf{Var}_{P_{i,s,a,b}^0}\big(\widehat{V}_{i+1}^+\big)$$

$$\leq \sum_{i=1}^{H} \sum_{(s,b)\in\mathcal{S}\times\mathcal{B}} d_i^{\mathsf{p},\mu^{\mathsf{d}},\nu^{\star}}(s,\mu^{\mathsf{d}}(s),b)\mathsf{Var}_{P_{i,s,a,b}^{\mathrm{inf},V}}\big(\widehat{V}_{i+1}^+\big)$$

$$+ \sum_{i=1}^{H} \sum_{(s,b)\in\mathcal{S}\times\mathcal{B}} d_i^{\mathsf{p},\mu^{\mathsf{d}},\nu^{\star}}(s,\mu^{\mathsf{d}}(s),b)\left|\mathsf{Var}_{P_{i,s,a,b}^0}\big(\widehat{V}_{i+1}^+\big) - \mathsf{Var}_{P_{i,s,a,b}^{\mathrm{inf},V}}\big(\widehat{V}_{i+1}^+\big)\right|$$

$$\leq \sum_{i=1}^{H} 2\left(\big\|\widehat{V}_i'\big\|_{\infty} + \big\|\widehat{V}_{i+1}'\big\|_{\infty}\right)\sum_{s\in\mathcal{S}} d_i^{\mathsf{p},\mu^{\mathsf{d}},\nu^{\star}}(s)\beta_i^{\mu^{\mathsf{d}},\nu^{\star}}(s) + \sum_{i=1}^{H}\left(\big\|\widehat{V}_i'\big\|_{\infty} + (H+2)\big\|\widehat{V}_{i+1}'\big\|_{\infty}\right)$$

$$+ \sum_{s\in\mathcal{S}} d_{H+1}^{\mathsf{p},\mu^{\mathsf{d}},\nu^{\star}}(s)\widehat{V}_{H+1}'(s)\circ\widehat{V}_{H+1}'(s)$$

$$\leq 4\sum_{i=1}^{H} \min\left\{\frac{(H+1)\left(1-(1-\sigma^+)^{H-i}\right)}{\sigma^+},H\right\}\sum_{(s,b)\in\mathcal{S}\times\mathcal{B}} d_i^{\mathsf{p},\mu^{\mathsf{d}},\nu^{\star}}(s,\mu^{\mathsf{d}}(s),b)\beta_i(s,\mu^{\mathsf{d}}(s),b,\widehat{V})$$

$$+ (H+3)\sum_{i=1}^{H} \min\left\{\frac{(H+1)\left(1-(1-\sigma^+)^{H-i}\right)}{\sigma^+},H\right\}$$

$$\overset{(i)}{\leq} 4\sum_{i=1}^{H} \min\left\{\frac{(H+1)\left(1-(1-\sigma^+)^{H-i}\right)}{\sigma^+},H\right\}\sum_{i=1}^{H}\sum_{(s,b)\in\mathcal{S}\times\mathcal{B}} d_i^{\mathsf{p},\mu^{\mathsf{d}},\nu^{\star}}(s,\mu^{\mathsf{d}}(s),b)\beta_i(s,\mu^{\mathsf{d}}(s),b,\widehat{V})$$

$$+ (H+3)\sum_{i=1}^{H} \min\left\{\frac{(H+1)\left(1-(1-\sigma^+)^{H-i}\right)}{\sigma^+},H\right\}$$

$$\overset{(ii)}{\leq} 4H\min\left\{\frac{2(H\sigma^+ - 1 + (1-\sigma^+)^H)}{(\sigma^+)^2},H\right\}\sum_{i=1}^{H}\sum_{(s,b)\in\mathcal{S}\times\mathcal{B}} d_i^{\mathsf{p},\mu^{\mathsf{d}},\nu^{\star}}(s,\mu^{\mathsf{d}}(s),b)\beta_i(s,\mu^{\mathsf{d}}(s),b,\widehat{V})$$

$$+ (H+3)H\min\left\{\frac{2(H\sigma^+ - 1 + (1-\sigma^+)^H)}{(\sigma^+)^2},H\right\}, \tag{109}$$

where (i) comes from Cauchy-Schwarz inequality and the (ii) holds since

$$\sum_{i=1}^{H} \frac{(H+1)\left(1-(1-\sigma^+)^{H-i}\right)}{\sigma^+} = \frac{H(H+1)}{\sigma^+} - \sum_{i=0}^{H-1}\frac{(H+1)(1-\sigma^+)^i}{\sigma^+}$$

$$= \frac{H(H+1)}{\sigma^+} - \frac{(H+1)(1-(1-\sigma^+)^H)}{(\sigma^+)^2}$$

$$= \frac{(H+1)(H\sigma^+ - 1 + (1-\sigma^+)^H)}{(\sigma^+)^2}$$

$$\leq \frac{2H(H\sigma^+ - 1 + (1-\sigma^+)^H)}{(\sigma^+)^2}.$$

## D  PROOF OF THEOREM 2

In this section, we focus on a simpler class of RTZMGs: robust Markov decision processes (MDPs), which are single-agent versions of RTZMGs.

Before proceeding, we briefly define a Robust MDP (RMDP) in the finite-horizon episodic setting. Recall that an RTZMG with an uncertainty set is represented as $\mathcal{MG} = \{\mathcal{S},\mathcal{A},\mathcal{B},\mathcal{U}^{\sigma^+}(P^0),\mathcal{U}^{\sigma^-}(P^0),r,H\}$. For simplicity, we assume $\mathcal{A}\geq\mathcal{B}$, and set $|\mathcal{B}|=1$, meaning the min-player's actions do not affect transitions or rewards. Thus, finding a robust NE

in such RTZMGs reduces to finding the max-player's optimal policy in a corresponding RMDP $\mathcal{M}_r = \{\mathcal{S}, \mathcal{A}, \mathcal{U}^{\sigma^+}(P^0), r, H\}$.

Thus, in this section, we construct the lower bound for finding the optimal policy in RTZMGs, which also implies a lower bound for finding robust NE in RTZMGs. We first highlight a useful property about KL divergence from Tsybakov (2008, Lemma 2.7), which can be helpful in this section.

**Lemma 7** *For any $p, q \in (0, 1)$, it holds that*

$$\mathsf{KL}(p \parallel q) \leq \frac{(p - q)^2}{q(1 - q)}. \tag{110}$$

### D.1 STEP 1: CONSTRUCTING A FAMILY OF HARD MARKOV GAME INSTANCES

The hard instances developed here differ from standard MDP since we need to consider that the transition kernel can be perturbed in robust MDPs.

**Constructing hard robust MDP instances.** To begin with, we first introduce an auxiliary collection $\Phi \subseteq \{0, 1\}^H$, consisting of $H$-dimensional vectors. In addition, resorting to the Gilbert-Varshamov lemma (Gilbert, 1952), we notice that there exists a set $\Phi \subseteq \{0, 1\}^H$ such that:

$$\text{for any } \phi, \widetilde{\phi} \in \Phi \text{ obeying } \phi \neq \widetilde{\phi}: \quad \|\phi - \widetilde{\phi}\|_1 \geq \frac{H}{8} \quad \text{and} \quad |\Phi| \geq e^{H/8}. \tag{111}$$

With this in mind, we construct a set of RMDPs as below:

$$\mathcal{M}(\mathcal{F}, \Phi) := \left\{ \mathcal{M}_f^\phi = \left( \mathcal{S}, \mathcal{A}, \mathcal{U}^{\sigma^+}(P^{f,\phi}), r, H \right) \mid f \in \mathcal{F} = \{0, 1, \cdots, SA - 1\}, \right.$$

$$\left. \phi = [\phi_h]_{1 \leq h \leq H} \in \Phi \right\}, \tag{112}$$

where

$$\mathcal{S} = \{0, 1, \ldots, S - 1\}, \quad \text{and} \quad \mathcal{A} = \{0, 1, \cdots, A - 1\},$$

and $\sigma^+$ will be introduced momentarily.

In simple terms, the collection $\mathcal{M}(\mathcal{F}, \Phi)$ consists of $SA$ subsets, each containing $|\Phi|$ different RMDPs associated with some $f \in \mathcal{F}$. The state space for each RMDP $\mathcal{M}_f^\phi \in \mathcal{M}(\mathcal{F}, \Phi)$, denoted as $\mathcal{S}_{\mathsf{one}}$, includes two types of states: $\mathcal{M} = \{m_i \mid i \in \mathcal{F}\}$ and $\mathcal{N} = \{n_i \mid i \in \mathcal{F}\}$. Each state in $\mathcal{M}$ and $\mathcal{N}$ has two possible actions, $\mathcal{A}_{\mathsf{one}} = \{0, 1\}$. Thus, there are a total of $2SA$ states and $4SA$ state-action pairs.

With these notations, we define the transition kernels for $\mathcal{M}(\mathcal{F}, \Phi)$. For any RMDP $\mathcal{M}_f^\phi \in \mathcal{M}(\mathcal{F}, \Phi)$, the transition kernel $P^{f,\phi} = \{P_h^{f,\phi}\}_{h=1}^H$ is defined as follows, for any $(s, a, s', h) \in \mathcal{S}_{\mathsf{one}} \times \mathcal{A}_{\mathsf{one}} \times \mathcal{S}_{\mathsf{one}} \times [H]$,

$$P_h^{f,\phi}(s' \mid s, a) = \begin{cases} p\mathbb{1}(s' = n_f) + (1 - p)\mathbb{1}(s' = s) & \text{if} \quad s = m_f, a = \phi_h \\ q\mathbb{1}(s' = n_f) + (1 - q)\mathbb{1}(s' = s) & \text{if} \quad s = m_f, a = 1 - \phi_h \\ \mathbb{1}(s' = s) & \text{otherwise} \end{cases} \tag{113}$$

where $p$ and $q$ follow $p > q \geq \frac{1}{2}$.

In addition, the reward function is defined as

$$\forall (h, s, a) \in [H] \times \mathcal{S}_{\mathsf{one}} \times \mathcal{A}_{\mathsf{one}} : \quad r_h(s, a) = \begin{cases} 1 & \text{if } s \in \mathcal{N} \\ 0 & \text{otherwise.} \end{cases} \tag{114}$$

**Uncertainty set of the transition kernels.** Denote the transition kernel vector as

$$\forall (h, s, a) \in [H] \times \mathcal{S}_{\mathsf{one}} \times \mathcal{A}_{\mathsf{one}} : \quad P_{h,s,a}^{f,\phi} := P_h^{f,\phi}(\cdot \mid s, a) \in \Delta(\mathcal{S}). \tag{115}$$

Recalling the uncertainty set defined in (1), we know that $\mathcal{U}^{\sigma^+}(P^{f,\phi})$ represents:

$$\mathcal{U}^{\sigma^+}(P^{f,\phi}) := \otimes \, \mathcal{U}^{\sigma^+}(P^{f,\phi}_{h,s,a}), \quad \mathcal{U}^{\sigma^+}(P^{f,\phi}_{h,s,a}) := \left\{ \widetilde{P}^{f,\phi}_{h,s,a} \in \Delta(\mathcal{S}) : \rho\left(\widetilde{P}^{f,\phi}_{h,s,a} - P^{f,\phi}_{h,s,a}\right) \leq \sigma^+ \right\},$$

where $\otimes$ represents the Cartesian product over $(h, s, a) \in [H] \times \mathcal{S}_{\mathsf{one}} \times \mathcal{A}_{\mathsf{one}}$.

For the convenience of the subsequent proof, we analyze the TV distance as an uncertainty set for example, which means

$$\mathcal{U}^{\sigma^+}(P^{f,\phi}_{h,s,a}) := \left\{ \widetilde{P}^{f,\phi}_{h,s,a} \in \Delta(\mathcal{S}) : \frac{1}{2}\left\| \widetilde{P}^{f,\phi}_{h,s,a} - P^{f,\phi}_{h,s,a} \right\| \leq \sigma^+ \right\}. \tag{116}$$

Next, we introduce useful notations and facts for this section. For any RMDP $\mathcal{M}^\phi_f \in \mathcal{M}(\mathcal{F}, \Phi)$ and any $(h, s, a, s') \in [H] \times \mathcal{S}_{\mathsf{one}} \times \mathcal{A}_{\mathsf{one}} \times \mathcal{S}_{\mathsf{one}}$, we define the minimum transition probability from $(s, a)$ to $s'$, determined by any perturbed transition kernel $P_{h,s,a} \in \mathcal{U}^{\sigma^+}(P^{f,\phi}_{h,s,a})$, as:

$$P^{\mathrm{inf},f,\phi}_h(s' \,|\, s,a) := \inf_{P_{h,s,a} \in \mathcal{U}^{\sigma^+}(P^{f,\phi}_{h,s,a})} P_h(s' \,|\, s,a) = \max\{P_h(s' \,|\, s,a) - \sigma^+, 0\}, \tag{117}$$

where the last equation follows directly from the definition of $\mathcal{U}^{\sigma^+}(\cdot)$ in (116), with the remaining probability distributed to other states.

For convenience, we also define the transition from each $s \in \mathcal{M}$ to the corresponding state $s^{m \to n} \in \mathcal{N}$ for any $\mathcal{M}^\phi_f$, which is crucial in our analysis: for all $h \in [H]$,

$$\text{for } m_f: \quad p^{\mathrm{inf}}_h := P^{\mathrm{inf},f,\phi}_h(n_f \,|\, m_f, \phi_h) = p - \sigma^+,$$
$$q^{\mathrm{inf}}_h := P^{\mathrm{inf},f,\phi}_h(n_f \,|\, m_f, 1 - \phi_h) = q - \sigma^+. \tag{118}$$

Then it is obvious that

$$p^{\mathrm{inf}}_1 = p^{\mathrm{inf}}_2 = \cdots p^{\mathrm{inf}}_H, \quad q^{\mathrm{inf}}_1 = q^{\mathrm{inf}}_2 = \cdots q^{\mathrm{inf}}_H, \tag{119}$$

which motivates us to abbreviate them consistently as $p^{\mathrm{inf}} := p^{\mathrm{inf}}_1$ and $q^{\mathrm{inf}} := q^{\mathrm{inf}}_1$ later.

**Robust value functions and optimal policies.** We now define the robust value functions and identify the optimal policies for RMDP instances. For any RMDP $\mathcal{M}^\phi_f \in \mathcal{M}(\mathcal{F}, \Phi)$, let $\widetilde{\mu}^{\star,f,\phi} = \{\mu^{\star,f,\phi}_h\}^H_{h=1}$ represent the optimal policy, given that $\nu$ is deterministic. At each step $h$, we use $V^{\widetilde{\mu},\sigma^+,f,\phi}_h$ and $V^{\star,\sigma^+,f,\phi}_h$ to denote the robust value function of any policy $\widetilde{\mu}$ and the optimal policy $\widetilde{\mu}^{\star,f,\phi}$, respectively, under uncertainty level $\sigma^+$. The following lemma highlights key properties of robust value functions and optimal policies; the proof is deferred to Appendix E.1.

**Lemma 8** *Consider any $\mathcal{M}^\phi_f \in \mathcal{M}(\mathcal{F}, \Phi)$ and any policy $\widetilde{\mu}$. Defining*

$$m^{\widetilde{\mu},f,\phi}_h = p^{\mathrm{inf}}\widetilde{\mu}_h(\phi_h \,|\, m_f) + q^{\mathrm{inf}}\widetilde{\mu}_h(1 - \phi_h \,|\, m_f), \tag{120}$$

*it holds that*

$$\forall h \in [H]: \quad V^{\widetilde{\mu},\sigma^+,f,\phi}_h(m_f) = m^{\widetilde{\mu},f,\phi}_h V^{\widetilde{\mu},\sigma^+,f,\phi}_{h+1}(n_f) + (1 - m^{\widetilde{\mu},f,\phi}_h)V^{\widetilde{\mu},\sigma^+,f,\phi}_{h+1}(m_f), \tag{121a}$$

$$\forall (s, h) \in \mathcal{N} \times [H]: \quad V^{\widetilde{\mu},\sigma^+,f,\phi}_h(s) = 1 + (1 - \sigma^+)V^{\widetilde{\mu},\sigma^+,f,\phi}_{h+1}(s) + \sigma^+ V^{\widetilde{\mu},\sigma^+,f,\phi}_{h+1}(m_f). \tag{121b}$$

*In addition, for all $h \in [H]$, the optimal policy and the optimal value function obey*

$$\widetilde{\mu}^{\star,f,\phi}_h(\phi_h \,|\, m_f) = \widetilde{\mu}^{\star,f,\phi}_h(\phi_h \,|\, n_f) = 1, \tag{122}$$

$$V^{\star,\sigma^+,f,\phi}_h(m_f) = p^{\mathrm{inf}}V^{\widetilde{\mu},\sigma^+,f,\phi}_{h+1}(n_f) + (1 - p^{\mathrm{inf}})V^{\widetilde{\mu},\sigma^+,f,\phi}_{h+1}(m_f). \tag{123}$$

**Construction of the history/batch dataset.**   In the nominal environment $\mathcal{M}_f^{\phi,\mathsf{n}}$, a batch dataset is generated with $K$ independent sample trajectories, each of length $H$, according to (5) and based on the initial state distribution $\varrho^{\mathsf{n}}$ and behavior policy $\widetilde{\mu}^{\mathsf{n}} = \{\mu_h^{\mathsf{n}}\}_{h=1}^{H}$ satisfying

$$\varrho^{\mathsf{n}}(s) = \varrho(s) \quad \text{and} \quad \widetilde{\mu}_h^{\mathsf{n}}(a \,|\, s) = \frac{1}{2}, \qquad \forall (s, a, h) \in \mathcal{S}_{\mathsf{one}} \times \mathcal{A}_{\mathsf{one}} \times [H]. \tag{124}$$

We define the nominal transition kernels for $\mathcal{M}_f^{\phi,\mathsf{n}}$, where any state $m_i \in \mathcal{M}$ transitions only to the corresponding $n_i \in \mathcal{N}$ or remains at itself. For simplicity, for any $s = m_i \in \mathcal{M}$, we denote the corresponding state $n_i \in \mathcal{N}$ as $s^{m \to n}$. The basic nominal transition kernel is defined as follows for all $(h, s, a) \in [H] \times \mathcal{S}_{\mathsf{one}} \times \mathcal{A}_{\mathsf{one}}$:

$$P_h^{\star}(s' \,|\, s, a) = \begin{cases} (p + \Delta)\mathbb{1}(s' = s^{m \to n}) + (1 - p - \Delta)\mathbb{1}(s' = s) & \text{if} \quad s \in \mathcal{M}, a = \phi_h \\ p\mathbb{1}(s' = s^{m \to n}) + (1 - p)\mathbb{1}(s' = s) & \text{if} \quad s \in \mathcal{M}, a = 1 - \phi_h \\ \mathbb{1}(s' = s) & \text{if} \quad s \in \mathcal{N}. \end{cases} \tag{125}$$

In words, the transition kernel of each $\mathcal{M}_f^{\phi} \in \mathcal{M}(\mathcal{F}, \Phi)$ only differs slightly from the basic nominal transition kernel $\mathcal{M}_f^{\phi,\mathsf{n}}$ when $s = m_f$, which makes all the components within $\mathcal{M}(\mathcal{F}, \Phi)$ close to each other.

Specifically, $p$ and $q$ are set according to

$$0 \leq p \leq p + \Delta \leq 1 \quad \text{and} \quad 0 \leq q = p - \Delta \tag{126}$$

for some $p$ and $\Delta > 0$. Without loss of generality, let the uncertainty level be $\sigma^+ \in (0, 1 - c_0]$ for some $0 < c_0 < 1$. Then taking $c_2 \leq \frac{1}{4}$ and $c_1 := \frac{c_0}{2} \leq \frac{1}{4}$, $p$ and $\Delta$ are set as

$$p = \begin{cases} \frac{c_2}{H}, & \text{if } \sigma^+ \leq \frac{c_2}{2H} \\ \left(1 + \frac{c_1}{H}\right)\sigma^+ & \text{otherwise} \end{cases} \quad \text{and} \quad \Delta \leq \begin{cases} \frac{c_2}{2H}, & \text{if } \sigma^+ \leq \frac{c_2}{2H} \\ \frac{c_1}{H}\sigma^+ & \text{otherwise} \end{cases} \tag{127}$$

which establishes the fact that

$$p + \Delta \geq p \geq q = p - \Delta \geq \max\left\{\frac{c_2}{2H}, \sigma^+\right\}. \tag{128}$$

Combined with $H \geq 2$, it is easily verified that $0 \leq p + \Delta \leq 1$ as follows:

$$\text{when } \sigma^+ > \frac{c_2}{2H} : \quad \left(1 + \frac{c_1}{H}\right)\sigma^+ + \frac{c_1}{H}\sigma^+ \leq 1 - c_0 + \frac{2c_1}{H}\sigma^+ \leq 1 - \frac{c_0(H - 1)}{H} < 1,$$

$$\text{when } \sigma^+ \leq \frac{c_2}{2H} : \quad \frac{3c_2}{2H} \leq 1. \tag{129}$$

In addition, let $\overline{\varrho}(s)$ represents a state distribution supported on the state subset $(m_f, n_f) \in \mathcal{M} \times \mathcal{N}$:

$$\overline{\varrho}(s) = \frac{1}{CSA}\mathbb{1}(s = m_f) + \left(1 - \frac{1}{CSA}\right)\mathbb{1}(s = n_f), \tag{130}$$

where $\mathbb{1}(\cdot)$ is the indicator function, and $C > 0$ is some constant that will determine the concentrability coefficient $C_{\mathsf{r}}^{\star}$ (as we shall detail momentarily) and obeys

$$\frac{1}{CSA} \leq \frac{1}{4}. \tag{131}$$

As it turns out, for any MDP $\mathcal{M}_\phi^f$, the occupancy distributions of the above batch dataset are the same (due to symmetry) and admit the following simple characterization:

$$\forall (s, a) \in \mathcal{S}_{\mathsf{one}} \times \mathcal{A}_{\mathsf{one}}, \qquad d_1^{\mathsf{n}, P^{\phi,f}}(s, a) = \frac{1}{2}\overline{\varrho}(s), \tag{132a}$$

$$\forall (s, a, h) \in \mathcal{S}_{\mathsf{one}} \times \mathcal{A}_{\mathsf{one}} \times [H], \qquad \frac{\overline{\varrho}(s)}{2} \leq d_h^{\mathsf{n}, P^{\phi,f}}(s) \leq 2\overline{\varrho}(s), \quad \frac{\overline{\varrho}(s)}{4} \leq d_h^{\mathsf{n}, P^{\phi,f}}(s, a) \leq \overline{\varrho}(s). \tag{132b}$$

In addition, we choose the following initial state distribution

$$
\varrho(s) = \begin{cases} \frac{1}{CSA}, & \text{if } s \in \mathcal{M} \\ 0, & \text{if } s \in \mathcal{N}. \end{cases}
\tag{133}
$$

With this choice of $\varrho$, the single-policy clipped concentrability coefficient $C_{\mathrm{r}}^\star$ and the quantity $C$ are intimately connected as follows:

$$
C \leq C_{\mathrm{r}}^\star \leq 2C.
\tag{134}
$$

The proof of the claim (132) and (134) are postponed to Appendix E.2.

### D.2 STEP 2: ESTABLISHING THE MINIMAX LOWER BOUND

Recall our goal: for any policy estimator $\widetilde{\mu}$ computed based on the empirical dataset, we plan to control the quantity

$$
\max_{(f,\phi) \in \mathcal{F} \times \Phi} \left\{ V_1^{\star,\sigma^+,f,\phi}(\varrho) - V_1^{\widetilde{\mu},\sigma^+,f,\phi}(\varrho) \right\}
\tag{135}
$$

with initial state distribution defined in (133).

**Step 1: converting the goal to estimate** $(f,\phi)$. Towards this, we make the following essential claim which shall be verified in Appendix E.3: letting

$$
\varepsilon \leq \begin{cases} \frac{c_2}{H}, & \text{if } \sigma^+ \leq \frac{c_2}{2H} \\ 1 & \text{otherwise} \end{cases}
\tag{136}
$$

and

$$
\Delta = c_5 \begin{cases} \frac{\varepsilon}{H^2}, & \text{if } \sigma^+ \leq \frac{c_2}{2H} \\ \frac{\sigma^+ \varepsilon}{H} & \text{otherwise} \end{cases}
\tag{137}
$$

which satisfies (127), it leads to that for any policy $\widetilde{\mu}$ obeying

$$
\sum_{h=1}^{H} \left\| \widetilde{\mu}_h(\cdot \mid m_f) - \widetilde{\mu}_h^{\star,f,\phi}(\cdot \mid m_f) \right\|_1 \geq \frac{H}{8},
\tag{138}
$$

one has

$$
V_1^{\star,\sigma^+,f,\phi}(m_f) - V_1^{\widetilde{\mu},\sigma^+,f,\phi}(m_f) > \varepsilon.
\tag{139}
$$

We are now ready to convert the task of estimating an optimal policy to estimating $(f,\phi)$. For this, let $\mathbb{P}_{f,\phi}$ represent the probability distribution when the RMDP is $\mathcal{M}_f^\phi$ for any $(f,\phi) \in \mathcal{F} \times \Phi$. Then, for any $(f,\phi) \in \mathcal{F} \times \Phi$, suppose that there exists a policy $\widetilde{\mu}$ achieving

$$
\mathbb{P}_{f,\phi} \left\{ V_1^{\star,\sigma^+,f,\phi}(m_f) - V_1^{\widetilde{\mu},\sigma^+,f,\phi}(m_f) \leq \varepsilon \right\} \geq \frac{3}{4},
\tag{140}
$$

which in view of (139) indicates that we necessarily have

$$
\mathbb{P}_{f,\phi} \left\{ \sum_{h=1}^{H} \left\| \widetilde{\mu}_h(\cdot \mid m_f) - \widetilde{\mu}_h^{\star,f,\phi}(\cdot \mid m_f) \right\|_1 < \frac{H}{8} \right\} \geq \frac{3}{4}.
\tag{141}
$$

Consequently, taking $\widetilde{\phi} = \arg\min_{\phi \in \Phi} \sum_{h=1}^{H} \left\| \widetilde{\mu}_h(\cdot \mid m_f) - \widetilde{\mu}_h^{\star,f,\phi}(\cdot \mid m_f) \right\|_1$, we are motivated to construct the estimate of $\phi$ as $\widehat{\phi} = \widetilde{\phi}$. Namely, if $\sum_{h=1}^{H} \left\| \widetilde{\mu}_h(\cdot \mid m_f) - \widetilde{\mu}_h^{\star,f,\phi}(\cdot \mid m_f) \right\|_1 < \frac{H}{8}$ holds for some $\phi \in \Phi$, then for any $\phi' \in \Phi$ obeying $\phi' \neq \phi$, one has

$$
\sum_{h=1}^{H} \left\| \widetilde{\mu}_h(\cdot \mid m_f) - \widetilde{\mu}_h^{\star,f,\phi'}(\cdot \mid m_f) \right\|_1
$$

$$
\geq \sum_{h=1}^{H} \left\| \widetilde{\mu}_h^{\star,f,\phi}(\cdot \mid m_f) - \widetilde{\mu}_h^{\star,f,\phi'}(\cdot \mid m_f) \right\|_1 - \sum_{h=1}^{H} \left\| \widetilde{\mu}_h(\cdot \mid m_f) - \widetilde{\mu}_h^{\star,f,\phi}(\cdot \mid m_f) \right\|_1
$$

$$
> \frac{H}{4} - \frac{H}{8} = \frac{H}{8},
\tag{142}
$$

where the first inequality holds by the triangle inequality, and the last inequality follows from the assumption $\sum_{h=1}^{H} \left\| \widetilde{\mu}_h(\cdot \mid m_f) - \widetilde{\mu}_h^{\star,f,\phi}(\cdot \mid m_f) \right\|_1 < \frac{H}{8}$ and the separation property of $\phi \in \Phi$ (see (111)). Similarly, it shows that we have $\widehat{\phi} = \phi$ if

$$\sum_{h=1}^{H} \left\| \widetilde{\mu}_h(\cdot \mid m_f) - \widetilde{\mu}_h^{\star,f,\phi}(\cdot \mid m_f) \right\|_1 < \frac{H}{8} < \sum_{h=1}^{H} \left\| \widetilde{\mu}_h(\cdot \mid m_f) - \widetilde{\mu}_h^{\star,f,\phi'}(\cdot \mid m_f) \right\|_1 \qquad (143)$$

holds for all $\phi' \in \Phi$ that $\phi' \neq \phi$. It is clear that the above equation can be directly achieved when $\sum_{h=1}^{H} \left\| \widetilde{\mu}_h(\cdot \mid m_f) - \widetilde{\mu}_h^{\star,f,\phi}(\cdot \mid m_f) \right\|_1 < \frac{H}{8}$, which gives

$$\mathbb{P}_{f,\phi}\left[\widehat{\phi} = \phi\right] \geq \mathbb{P}_{f,\phi}\left\{ \sum_{h=1}^{H} \left\| \widetilde{\mu}_h(\cdot \mid m_f) - \widetilde{\mu}_h^{\star,f,\phi}(\cdot \mid m_f) \right\|_1 < \frac{H}{8} \right\} \geq \frac{3}{4}. \qquad (144)$$

**Step 2: developing the probability of error in testing multiple hypotheses.** Next, we address the hypothesis testing problem over $\phi \in \Phi$ and derive the information-theoretic lower bound for the probability of error. Specifically, we define the minimax probability of error as:

$$p_{\mathrm{e}} := \inf_{(\widehat{f},\widehat{\phi})} \max_{(f,\phi)\in\mathcal{F}\times\Phi} \mathbb{P}_{f,\phi}\left(\widehat{\phi} \neq \phi\right),$$

where the infimum is taken over all possible tests $\widehat{\phi}$ constructed from the available batch dataset.

Given the dataset $\mathcal{D}_0$ with $K$ independent trajectories, let $\varrho^{\mathsf{n},\phi}$ (and $\varrho_h^{\mathsf{n},\phi}(s,a)$) represent the distribution vector (and distribution) of each sample tuple $(s_h, a_h, s_h')$ at time step $h$ under the nominal transition kernel $P^\star$ for $\mathcal{M}_f^{\phi,\mathsf{n}}$. Using this, along with Fano's inequality (Tsybakov, 2008, Theorem 2.2) and the additivity of KL divergence (Tsybakov, 2008, Page 85), we derive the following result:

$$\begin{aligned}
p_{\mathrm{e}} &\geq 1 - K \frac{\max_{(\phi,\widetilde{\phi})\in\Phi,\phi\neq\widetilde{\phi}} \mathsf{KL}\left(\varrho^{\mathsf{n},\phi} \mid \varrho^{\mathsf{n},\widetilde{\phi}}\right) + \log 2}{\log|\Phi|} \\
&\overset{(i)}{\geq} 1 - \frac{8K}{H} \max_{(\phi,\widetilde{\phi})\in\Phi,\phi\neq\widetilde{\phi}} \mathsf{KL}\left(\varrho^{\mathsf{n},\phi} \mid \varrho^{\mathsf{n},\widetilde{\phi}}\right) - \frac{8\log 2}{H} \\
&\overset{(ii)}{\geq} \frac{1}{2} - \frac{8K}{H} \max_{(\phi,\widetilde{\phi})\in\Phi,\phi\neq\widetilde{\phi}} \mathsf{KL}\left(\varrho^{\mathsf{n},\phi} \mid \varrho^{\mathsf{n},\widetilde{\phi}}\right),
\end{aligned} \qquad (145)$$

where (i) holds by $|\Phi| \geq e^{H/8}$ and (ii) follows from $H \geq 16\log 2$.

Since the occupancy state distribution $d_h^{\mathsf{n}}$ is the same for any MDP $\mathcal{M}_f^\phi$ for $\phi \in \Phi$, we apply the chain rule of KL divergence (Duchi, 2018, Lemma 5.2.8) and the Markov property of the independent sample trajectories to obtain:

$$\begin{aligned}
\mathsf{KL}\left(\varrho^{\mathsf{n},\phi} \mid \varrho^{\mathsf{n},\widetilde{\phi}}\right) &= \sum_{h=1}^{H} \mathbb{E}_{s\sim d_h^{\mathsf{n}}(s)} \left[ \mathsf{KL}\left(P_h^{\star,\phi}(\cdot \mid s,a) \parallel P_h^{\star,\widetilde{\phi}}(\cdot \mid s,a)\right) \right] \\
&\overset{(i)}{=} \frac{1}{2}\overline{\varrho}(m_f) \sum_{h=1}^{H} \sum_{a\in\{0,1\}} \left[ \mathsf{KL}\left(P_h^\phi(\cdot \mid m_f,a) \parallel P_h^{\widetilde{\phi}}(\cdot \mid m_f,a)\right) \right], \qquad (146)
\end{aligned}$$

where (i) follows from applying (132) and obtaining the fact as

$$\begin{aligned}
&\mathbb{E}_{s\sim d_h^{\mathsf{n}}(s)} \left[ \mathsf{KL}\left(P_h^{\star,\phi}(\cdot \mid s,a) \parallel P_h^{\star,\widetilde{\phi}}(\cdot \mid s,a)\right) \right] \\
&= \sum_s d_h^{\mathsf{n}}(s) \left\{ \sum_{a,s'} \widetilde{\mu}_h^{\mathsf{n}}(a \mid s) P_h^{\phi_h}(s' \mid s,a) \log \frac{\widetilde{\mu}_h^{\mathsf{n}}(a \mid s) P_h^{\phi_h}(s' \mid s,a)}{\widetilde{\mu}_h^{\mathsf{n}}(a \mid s) P_h^{\widetilde{\phi}_h}(s' \mid s,a)} \right\} \\
&= \frac{1}{2}\overline{\varrho}(m_f) \sum_a \sum_{s'} P_h^{\phi_h}(s' \mid m_f,a) \log \frac{P_h^{\phi_h}(s' \mid m_f,a)}{P_h^{\widetilde{\phi}_h}(s' \mid m_f,a)} \\
&= \frac{1}{2}\overline{\varrho}(m_f) \sum_a \mathsf{KL}\left(P_h^{\phi_h}(\cdot \mid m_f,a) \parallel P_h^{\widetilde{\phi}_h}(\cdot \mid m_f,a)\right).
\end{aligned}$$

Consequently, combining (145) and (146) leads to

$$p_e \geq \frac{1}{2} - \frac{4K}{H} \max_{(\phi, \widetilde{\phi}) \in \Phi, \phi \neq \widetilde{\phi}} \left[ \overline{\varrho}(m_f) \sum_{h=1}^{H} \sum_{a} \mathsf{KL}\big(P_h^{\phi_h}(\cdot \mid m_f, a) \parallel P_h^{\widetilde{\phi}_h}(\cdot \mid m_f, a)\big) \right]. \qquad (147)$$

Thus, we turn to focus on terms in (147) now in different cases of the uncertainty level $\sigma^+$.

- For $0 < \sigma^+ \leq \frac{c_2}{2H}$: If $\phi_h = \widetilde{\phi}_h$, it is obvious that

$$\sum_{a \in \{0,1\}} \mathsf{KL}\big(P_h^{\star, \phi}(\cdot \mid s, a) \parallel P_h^{\star, \widetilde{\phi}}(\cdot \mid s, a)\big) = 0. \qquad (148)$$

  Therefore, we consider the case of $\phi_h \neq \widetilde{\phi}_h$. Without loss of generality, we suppose $\phi_h = 0$ and $\widetilde{\phi}_h = 1$, which indicates

$$\mathsf{KL}\big(P_h^{\star, \phi}(0 \mid m_f, 0) \parallel P_h^{\star, \widetilde{\phi}}(0 \mid m_f, 0)\big) \leq \frac{(p-q)^2}{q(1-q)} \overset{(i)}{=} \frac{\Delta^2}{q(1-q)}$$

$$\overset{(ii)}{=} \frac{(c_5)^2 \varepsilon^2}{H^4 q(1-q)} \leq \frac{4(c_5)^2 \varepsilon^2}{c_2 H^3}, \qquad (149)$$

  where the first inequality exists by applying Lemma 7, (i) follows from the definitions in (126), (ii) holds due to the definition in (137), and the last inequality arises from $q = p - \Delta \geq \frac{c_2}{2H}$ (see (127)) and $1 - q \geq 1 - p \geq 1 - \frac{c_2}{H} \geq \frac{1}{2}$.

  Similarly, we can establish the same bound for $\mathsf{KL}\big(P_h^{\star, \phi}(0 \mid m_f, 1) \parallel P_h^{\star, \widetilde{\phi}}(0 \mid m_f, 1)\big)$.

  Summing up the results with the fact in (149), we arrive at

$$\sum_{a \in \{0,1\}} \mathsf{KL}\big(P_h^{\star, \phi}(\cdot \mid m_f, a) \parallel P_h^{\star, \widetilde{\phi}}(\cdot \mid m_f, a)\big) \leq \frac{16(c_5)^2 \varepsilon^2}{c_2 H^3}. \qquad (150)$$

- For $\frac{c_2}{2H} < \sigma^+ \leq 1 - c_0$: Following the same pipeline, it then boils down to control the main term as below:

$$\mathsf{KL}\big(P_h^{\star, \phi}(0 \mid m_f, 0) \parallel P_h^{\star, \widetilde{\phi}}(0 \mid m_f, 0)\big) \leq \frac{(p-q)^2}{q(1-q)} \overset{(i)}{=} \frac{\Delta^2}{q(1-q)}$$

$$\overset{(ii)}{=} \frac{(c_5)^2 \sigma^{+2} \varepsilon^2}{H^2 q(1-q)} \leq \frac{2(c_5)^2 \sigma^+ \varepsilon^2}{c_0 H^2}, \qquad (151)$$

  where (i) and (ii) follow from the definitions in (126) or (137). Here, the last inequality arises from

$$1 - q \geq 1 - p = 1 - (1 + \frac{c_1}{H})\sigma^+ \overset{(i)}{\geq} c_0 - \frac{c_1}{H} \overset{(ii)}{\geq} \frac{c_0}{2}$$

$$p \geq q = p - \Delta \overset{(iii)}{\geq} \sigma^+, \qquad (152)$$

  where (ii) holds by the definition of $c_1 = \frac{c_0}{2}$, and (iii) follows from (128). Consequently, we arrive at

$$\sum_{a \in \{0,1\}} \mathsf{KL}\big(P_h^{\star, \phi}(\cdot \mid s, a) \parallel P_h^{\star, \widetilde{\phi}}(\cdot \mid s, a)\big) \leq \frac{8(c_5)^2 \sigma^+ \varepsilon^2}{c_0 H^2}. \qquad (153)$$

Summing up (150) and (153), we achieve for any $(\phi, \widetilde{\phi}) \in \Phi$ with $\phi \neq \widetilde{\phi}$ and any time step $h \in [H]$

$$\sum_{a \in \{0,1\}} \mathsf{KL}\big(P_h^{\star, \phi}(\cdot \mid m_f, a) \parallel P_h^{\star, \widetilde{\phi}}(\cdot \mid m_f, a)\big) \leq \frac{16(c_5)^2 \varepsilon^2}{c_0 c_2 H^2} \max\{\sigma^+, 1/H\}. \qquad (154)$$

Plugging (154) back to (147), under the definition in (133), we obtain

$$
\begin{aligned}
p_{\mathrm{e}} &\geq \frac{1}{2} - \frac{4K}{H} \max_{(\phi,\widetilde{\phi}) \in \Phi, \phi \neq \widetilde{\phi}} \left[ \overline{\varrho}(m_f) \sum_{h=1}^{H} \sum_{a} \mathsf{KL}\big( P_h^{\phi_h}(\cdot \,|\, m_f, a) \,\|\, P_h^{\widetilde{\phi}_h}(\cdot \,|\, m_f, a) \big) \right] \\
&\geq \frac{1}{2} - \frac{4K}{H} \overline{\varrho}(m_f) \sum_{h=1}^{H} \frac{16(c_5)^2 \varepsilon^2}{c_0 c_2 H^2} \max\{\sigma^+, 1/H\} \\
&\geq \frac{1}{2} - \frac{64K(c_5)^2 \varepsilon^2}{c_0 c_2 CSAH^2} \max\{\sigma^+, 1/H\} \geq \frac{1}{4},
\end{aligned}
\tag{155}
$$

as long as the sample size $T = KH$ of the dataset is selected as

$$
T \leq \frac{c_0 c_2 CSAH^3 \min\{1/\sigma^+, H\}}{256(c_5)^2 \varepsilon^2} \leq \frac{c_0 c_2 C_{\mathrm{r}}^\star SAH^3 \min\{1/\sigma^+, H\}}{256(c_5)^2 \varepsilon^2}.
\tag{156}
$$

**Step 3: summing up the results together.** We suppose that there exists an estimator $\widetilde{\mu}$ such that

$$
\max_{(f,\phi \in \mathcal{F}) \times \Phi} \mathbb{P}_{f,\phi} \left[ \left\{ V_1^{\star, \sigma^+, f, \phi}(\varrho) - V_1^{\widetilde{\mu}, \sigma^+, f, \phi}(\varrho) \right\} \geq \varepsilon \right] < \frac{1}{4}.
\tag{157}
$$

Then according to (135), we need

$$
\forall w \in \mathcal{F}: \quad \max_{\phi \in \Phi} \mathbb{P}_{f,\phi} \left[ \left\{ V_1^{\star, \sigma^+, f, \phi}(m_f) - V_1^{\widetilde{\mu}, \sigma^+, f, \phi}(m_f) \right\} \geq \varepsilon \right] < \frac{1}{4}.
\tag{158}
$$

To meet (158) for any $w \in \mathcal{F}$, we require

$$
\forall \phi \in \Phi: \mathbb{P}_{f,\phi} \left\{ V_1^{\star, \sigma^+, f, \phi}(m_f) - V_1^{\widetilde{\mu}, \sigma^+, f, \phi}(m_f) < \varepsilon \right\} \geq \frac{3}{4},
\tag{159}
$$

which in view of (139) indicates that we necessarily have

$$
\forall \phi \in \Phi: \quad \mathbb{P}_{f,\phi} \left\{ \sum_{h=1}^{H} \left\| \widetilde{\mu}_h(\cdot \,|\, m_f) - \widetilde{\mu}_h^{\star, f, \phi}(\cdot \,|\, m_f) \right\|_1 < \frac{H}{8} \right\} \geq \frac{3}{4}.
\tag{160}
$$

As a consequence, (144) indicates

$$
\forall \phi \in \Phi: \mathbb{P}_{f,\phi} \left[ \widehat{\phi} = \phi \right] \geq \frac{3}{4}.
\tag{161}
$$

To achieve (157), we here apply the fact in (161) to all $w \in \mathcal{F}$, which leads to the fact that one necessarily has

$$
\forall (f, \phi) \in \mathcal{F} \times \Phi: \quad \mathbb{P}_{f,\phi} \left[ (\widehat{f}, \widehat{\phi}) = (f, \phi) \right] \geq \frac{3}{4}.
\tag{162}
$$

However, this would contract with (155) as long as the sample size condition in (156) is satisfied. Thus, if the sample size obeys the condition (156), we can't achieve an estimate $\widetilde{\mu}$ that satisfies (157), which completes the proof.

# E    AUXILIARY FACTS FOR THEOREM 2

## E.1    PROOF OF LEMMA 8

Since all RMDPs in $\mathcal{M}(\mathcal{F}, \Phi)$ are constructed similarly for each $w \in \mathcal{F}$ and $\phi \in \Phi$, we will focus on a specific RMDP $\mathcal{M}_f^\phi \in \mathcal{M}(\mathcal{F}, \Phi)$, with the results applicable to all other RMDPs in $\mathcal{M}(\mathcal{F}, \Phi)$.

**Part 1: ordering the robust value function over different states.** Before proceeding, we introduce several facts and notations that will be useful throughout this section. First, for any $\mathcal{M}_f^\phi$ and any policy $\widetilde{\mu}$, we observe the following at the final step $H+1$:

$$\forall s \in \mathcal{M} \cup \mathcal{N}: \quad V_{H+1}^{\widetilde{\mu},\sigma^+,f,\phi}(s) = 0. \tag{163}$$

Then for step $H$, we can easily verify that

$$\forall s \in \mathcal{N}: \quad V_H^{\widetilde{\mu},\sigma^+,f,\phi}(s) = \mathbb{E}_{a\sim\widetilde{\mu}_H(\cdot\,|\,s)}\left[r_H(s,a) + \inf_{\mathcal{P}\in\mathcal{U}^{\sigma^+}(P_{H,s,a}^{f,\phi})} \mathcal{P}V_{H+1}^{\widetilde{\mu},\sigma^+,f,\phi}\right] = 1 \tag{164a}$$

$$\forall s \in \mathcal{M}: \quad V_H^{\widetilde{\mu},\sigma^+,f,\phi}(s) = \mathbb{E}_{a\sim\widetilde{\mu}_H(\cdot\,|\,s)}\left[r_H(s,a) + \inf_{\mathcal{P}\in\mathcal{U}^{\sigma^+}(P_{H,s,a}^{f,\phi})} \mathcal{P}V_{H+1}^{\widetilde{\mu},\sigma^+,f,\phi}\right] = 0, \tag{164b}$$

which holds by (163) and the definition of the reward function (see (114)). The above fact directly indicates that

$$\forall (s,s') \in \mathcal{M} \times \mathcal{N}: \quad \min_{\widetilde{s}\in\mathcal{S}} V_H^{\widetilde{\mu},\sigma^+,f,\phi}(\widetilde{s}) = V_H^{\widetilde{\mu},\sigma^+,f,\phi}(m_f) \leq V_H^{\widetilde{\mu},\sigma^+,f,\phi}(s) < V_H^{\widetilde{\mu},\sigma^+,f,\phi}(s'), \tag{165a}$$

$$\forall (s,s') \in \mathcal{N} \times \mathcal{N}: \quad V_H^{\widetilde{\mu},\sigma^+,f,\phi}(s) = V_H^{\widetilde{\mu},\sigma^+,f,\phi}(s'). \tag{165b}$$

Then we introduce a claim which we will prove by induction in a moment as below:

$$\forall (h,s,s') \in [H] \times \mathcal{M} \times \mathcal{N}: \quad V_h^{\widetilde{\mu},\sigma^+,f,\phi}(m_f) \leq V_h^{\widetilde{\mu},\sigma^+,f,\phi}(s) < V_h^{\widetilde{\mu},\sigma^+,f,\phi}(s') \tag{166a}$$

$$\forall (s,s') \in \mathcal{N} \times \mathcal{N}: \quad V_h^{\widetilde{\mu},\sigma^+,f,\phi}(s) = V_h^{\widetilde{\mu},\sigma^+,f,\phi}(s'). \tag{166b}$$

Note that the base case when the time step is $H+1$ is verified in (165). Assume that the following fact at time step $h+1$ holds

$$\forall (s,s') \in \mathcal{M} \times \mathcal{N}: \quad \min_{\widetilde{s}\in\mathcal{S}} V_{h+1}^{\widetilde{\mu},\sigma^+,f,\phi}(\widetilde{s}) = V_{h+1}^{\widetilde{\mu},\sigma^+,f,\phi}(m_f) \leq V_{h+1}^{\widetilde{\mu},\sigma^+,f,\phi}(s) < V_{h+1}^{\widetilde{\mu},\sigma^+,f,\phi}(s'), \tag{167a}$$

$$\forall (s,s') \in \mathcal{N} \times \mathcal{N}: \quad V_{h+1}^{\widetilde{\mu},\sigma^+,f,\phi}(s) = V_{h+1}^{\widetilde{\mu},\sigma^+,f,\phi}(s'). \tag{167b}$$

Therefore, the rest of the proof focuses on proving the same property for time step $h$. For RMDP $\mathcal{M}_f^\phi \in \mathcal{M}(\mathcal{F},\Phi)$ and any policy $\widetilde{\mu}$, we characterize the robust value function of different states separately:

- *For state $s \in \mathcal{N}$:* we observe that for any $s \in \mathcal{N}$,

$$V_h^{\widetilde{\mu},\sigma^+,f,\phi}(s) = \mathbb{E}_{a\sim\widetilde{\mu}_h(\cdot\,|\,s)}\left[r_h(s,a) + \inf_{P\in\mathcal{U}^{\sigma^+}(P_{h,s,a}^{f,\phi})} PV_{h+1}^{\widetilde{\mu},\sigma^+,f,\phi}\right]$$

$$\overset{(i)}{=} 1 + \mathbb{E}_{a\sim\widetilde{\mu}_h(\cdot\,|\,s)}\left[P_h^{\inf,f,\phi}(s\,|\,s,a)V_{h+1}^{\widetilde{\mu},\sigma^+,f,\phi}(s)\right] + \sigma^+ V_{h+1}^{\widetilde{\mu},\sigma^+,f,\phi}(m_f)$$

$$= 1 + (1-\sigma^+)V_{h+1}^{\widetilde{\mu},\sigma^+,f,\phi}(s) + \sigma^+ V_{h+1}^{\widetilde{\mu},\sigma^+,f,\phi}(m_f), \tag{168}$$

where (i) holds by $r_h(s,a) = 1$ for all $s \in \mathcal{N}$ (see (114)), the fact that $\min_{\widetilde{s}\in\mathcal{S}} V_{h+1}^{\widetilde{\mu},\sigma^+,f,\phi}(\widetilde{s}) = V_{h+1}^{\widetilde{\mu},\sigma^+,f,\phi}(m_f)$ induced by the induction assumption (cf. (167)) and the definition of $P_h^{\inf,f,\phi}(s\,|\,s,a)$ in (117), and the last equality follows from $P^{f,\phi}(s\,|\,s,a) = 1$ for all $(s,a) \in \mathcal{N} \times \mathcal{A}_{\text{one}}$. Resorting to the induction assumption in (167), we have

$$\forall (s,s') \in \mathcal{N} \times \mathcal{N}: \quad V_h^{\widetilde{\mu},\sigma^+,f,\phi}(s) = V_h^{\widetilde{\mu},\sigma^+,f,\phi}(s'). \tag{169}$$

- *For state $m_f$:* first, the robust value function at state $m_f$ obeys

$$V_h^{\widetilde{\mu},\sigma^+,f,\phi}(m_f)$$

$$\stackrel{(i)}{=} \mathbb{E}_{a \sim \widetilde{\mu}_h(\cdot \mid m_f)}\left[r_h(m_f, a) + \inf_{P \in \mathcal{U}^{\sigma^+}(P_{h,m_f,a}^{f,\phi})} PV_{h+1}^{\widetilde{\mu},\sigma^+,f,\phi}\right]$$

$$\stackrel{(i)}{=} 0 + \widetilde{\mu}_h(\phi_h \mid m_f) \inf_{P \in \mathcal{U}^{\sigma^+}(P_{h,m_f,\phi_h}^{f,\phi})} PV_{h+1}^{\widetilde{\mu},\sigma^+,f,\phi}$$

$$\qquad + \widetilde{\mu}_h(1 - \phi_h \mid m_f) \inf_{P \in \mathcal{U}^{\sigma^+}(P_{h,m_f,1-\phi_h}^{f,\phi})} PV_{h+1}^{\widetilde{\mu},\sigma^+,f,\phi}$$

$$\stackrel{(ii)}{=} \widetilde{\mu}_h(\phi_h \mid m_f)\left[p^{\inf}V_{h+1}^{\widetilde{\mu},\sigma^+,f,\phi}(n_f) + \left(1 - p^{\inf}\right)V_{h+1}^{\widetilde{\mu},\sigma^+,f,\phi}(m_f)\right]$$

$$\qquad + \widetilde{\mu}_h(1 - \phi_h \mid m_f)\left[q^{\inf}V_{h+1}^{\widetilde{\mu},\sigma^+,f,\phi}(n_f) + \left(1 - q^{\inf}\right)V_{h+1}^{\widetilde{\mu},\sigma^+,f,\phi}(m_f)\right]$$

$$\stackrel{(iii)}{=} m_h^{\widetilde{\mu},f,\phi}V_{h+1}^{\widetilde{\mu},\sigma^+,f,\phi}(n_f) + (1 - m_h^{\widetilde{\mu},f,\phi})V_{h+1}^{\widetilde{\mu},\sigma^+,f,\phi}(m_f) \tag{170}$$

$$\leq (1 - \sigma^+)V_{h+1}^{\widetilde{\mu},\sigma^+,f,\phi}(n_f) + \sigma^+V_{h+1}^{\widetilde{\mu},\sigma^+,f,\phi}(m_f). \tag{171}$$

where (i) uses the definition of the robust value function and the reward function in (114), (ii) uses the induction assumption in (167) so that the minimum is attained by picking the choice specified in (118) to absorb probability mass to state $m_f$, and (iii) holds by plugging in the definition (120) of $m_h^{\widetilde{\mu},f,\phi}$. Finally, the last inequality follows from the fact that function $f(m) := mV_{h+1}^{\widetilde{\mu},\sigma^+,f,\phi}(n_f) + (1 - m)V_{h+1}^{\widetilde{\mu},\sigma^+,f,\phi}(m_f)$ is monotonically increasing with $m$ since $V_{h+1}^{\widetilde{\mu},\sigma^+,f,\phi}(n_f) > V_{h+1}^{\widetilde{\mu},\sigma^+,f,\phi}(m_f)$ (see the induction assumption (167)), and the fact $m_h^{\widetilde{\mu},f,\phi} \leq 1 - \sigma^+$.

Combining the above results with (169), we confirm the claim in (166).

**Part 2: deriving the optimal policy and optimal robust value function.** We shall characterize the optimal policy and corresponding optimal robust value function for different states separately:

- *For states in $\mathcal{M}$:* Recall (170)

$$V_h^{\widetilde{\mu},\sigma^+,f,\phi}(m_f) = m_h^{\widetilde{\mu},f,\phi}V_{h+1}^{\widetilde{\mu},\sigma^+,f,\phi}(n_f) + (1 - m_h^{\widetilde{\mu},f,\phi})V_{h+1}^{\widetilde{\mu},\sigma^+,f,\phi}(m_f) \tag{172}$$

and the fact $V_{h+1}^{\widetilde{\mu},\sigma^+,f,\phi}(n_f) > V_{h+1}^{\widetilde{\mu},\sigma^+,f,\phi}(m_f)$ in (166). We observe that (172) is monotonicity increasing with respect to $m_h^{\widetilde{\mu},f,\phi}$, and $m_h^{\widetilde{\mu},f,\phi}$ is also increasing in $\widetilde{\mu}_h(\phi_h \mid m_f)$ (refer to the fact $p^{\inf} \geq q^{\inf}$ since $p \geq q$; see (126) and (118)). Consequently, the optimal policy and optimal robust value function in state $m_f$ thus obey

$$\forall h \in [H]: \quad \widetilde{\mu}_h^{\star,f,\phi}(\phi_h \mid m_f) = 1,$$

$$V_h^{\star,\sigma^+,f,\phi}(m_f) = p^{\inf}V_{h+1}^{\star,\sigma^+,f,\phi}(n_f) + \left(1 - p^{\inf}\right)V_{h+1}^{\star,\sigma^+,f,\phi}(m_f). \tag{173}$$

- *For states $s \in \mathcal{N}$:* Recall the transitions in (125) and (113). Considering that the action does not influence the state transition for all states $s \in \mathcal{N}$, without loss of generality, we choose the robust optimal policy obeying

$$\forall s \in \mathcal{N}: \quad \widetilde{\mu}_h^{\star,f,\phi}(\phi_h \mid s) = 1. \tag{174}$$

### E.2 PROOF OF CLAIM (132) AND (134)

**Proof of the claim (132).** With the initial state distribution and behavior policy defined in (124), we have for any MDP $\mathcal{M}_\phi^f$,

$$d_1^{n,P^{\phi,f}}(s) = \varrho^n(s) = \overline{\varrho}(s),$$

which leads to

$$\forall (m_f, a) \in \mathcal{M} \times \mathcal{A}_{\mathsf{one}}: \quad d_1^{\mathsf{n}, P^{\phi, f}}(m_f, a) = \frac{1}{2}\overline{\varrho}(m_f). \tag{175}$$

Along with $d_1^{\mathsf{n}, P^{\phi, f}}(n_f, a) = \frac{1}{2}\overline{\varrho}(n_f) = 0$, the claim (132a) is proved.

In view of (125), the state occupancy distribution at any step $h = 2, 3, \cdots, H$ obeys

$$
\begin{aligned}
d_h^{\mathsf{n}, P^{\phi, f}}(m_f) &\geq \mathbb{P}\left\{ s_h = s' \mid s_{h-1} = m_f; \widetilde{\mu}^{\mathsf{n}} \right\} \\
&\geq d_{h-1}^{\mathsf{n}, P^{\phi, f}}(m_f) \left[ \widetilde{\mu}_{h-1}^{\mathsf{n}}(\phi_{h-1} \mid m_f)(1 - p - \Delta) + \widetilde{\mu}_{h-1}^{\mathsf{n}}(1 - \phi_{h-1} \mid m_f)(1 - p) \right] \\
&\geq d_{h-1}^{\mathsf{n}, P^{\phi, f}}(m_f)(1 - p - \Delta) \geq \cdots \geq d_1^{\mathsf{n}, P^{\phi}}(m_f) \prod_{j=0}^{h-1}(1 - p - \Delta) \\
&\geq d_1^{\mathsf{n}, P^{\phi}}(m_f)\left(1 - p - \Delta\right)^H > \frac{\overline{\varrho}(m_f)}{2},
\end{aligned} \tag{176}
$$

where the last line makes use of the properties $p$ and $\Delta$ in (128) and

$$\left(1 - p - \Delta\right)^H \geq \left(1 - \frac{c_2}{2H}\right)^H \geq \left(1 - \frac{1}{2H}\right)^H \geq \frac{1}{2},$$

provided that $0 < c_2 < 1$. In addition, as state $n_f$ is an absorbing state and state $m_f$ will only transfer to itself or state $n_f$ at each time step, we directly achieve that

$$d_h^{\mathsf{n}, P^{\phi, f}}(m_f) \leq d_{h-1}^{\mathsf{n}, P^{\phi, f}}(m_f) \leq \cdots \leq d_1^{\mathsf{n}, P^{\phi, f}}(m_f) \leq \overline{\varrho}(m_f). \tag{177}$$

For state $n_f$, as it is absorbing, we directly have

$$d_h^{\mathsf{n}, P^{\phi, f}}(n_f) = \mathbb{P}\left\{ s_h = n_f \mid s_{h-1} = n_f; \widetilde{\mu}^{\mathsf{n}} \right\} \geq d_{h-1}^{\mathsf{n}, P^{\phi, f}}(n_f) \geq \cdots \geq d_1^{\mathsf{n}, P^{\phi, f}}(n_f) = \overline{\varrho}(n_f). \tag{178}$$

According to the assumption in (131), it is easily verified that

$$d_h^{\mathsf{n}, P^{\phi, f}}(n_f) \leq 1 \leq 2\overline{\varrho}(n_f). \tag{179}$$

Finally, combining (176), (177, 178), (179), the definitions of $P_h^{\star}(\cdot \mid s, a)$ in (125) and the Markov property, we arrive at for any $(h, s) \in [H] \times \mathcal{S}$,

$$\frac{\overline{\varrho}(s)}{2} \leq d_h^{\mathsf{n}, P^{\phi, f}}(s) \leq 2\overline{\varrho}(s), \tag{180}$$

which directly leads to

$$\frac{\overline{\varrho}(s)}{4} \leq d_h^{\mathsf{n}, P^{\phi, f}}(s, a) = \widetilde{\mu}_1^{\mathsf{n}}(a \mid s) d_h^{\mathsf{n}, P^{\phi, f}}(s) \leq \overline{\varrho}(s). \tag{181}$$

**Proof of the claim (134).** Examining the definition of $C_{\mathsf{r}}^{\star}$ in (22), we make the following observations.

- For $h = 1$, we have

$$
\begin{aligned}
&\max_{(s, a, P) \in \mathcal{S}_{\mathsf{one}} \times \mathcal{A}_{\mathsf{one}} \times \mathcal{U}^\sigma(P^\phi)} \frac{\min\left\{ d_1^{\star, P}(s, a), \frac{1}{4SA} \right\}}{d_1^{\mathsf{n}, P^{\phi, f}}(s, a)} \\
&\stackrel{(\mathrm{i})}{=} \max_{(s, P) \in \mathcal{M} \times \mathcal{U}^\sigma(P^\phi)} \frac{\min\left\{ d_1^{\star, P}(s, \phi_1), \frac{1}{4SA} \right\}}{d_1^{\mathsf{n}, P^{\phi, f}}(s, \phi_1)} \\
&\stackrel{(\mathrm{ii})}{=} \max_{(s, P) \in \mathcal{M} \times \mathcal{U}^\sigma(P^\phi)} \frac{1}{4SA d_1^{\mathsf{n}, P^{\phi, f}}(s, \phi_1)} \\
&\stackrel{(\mathrm{iii})}{=} \max_{s \in \mathcal{M}} \frac{1}{2SA\overline{\varrho}(s)} = C,
\end{aligned} \tag{182}
$$

where (i) holds by $d_1^{\star, P}(s) = \rho(s) = 0$ for all $s \in \mathcal{N}$ (see (133)) and $\widetilde{\mu}_h^{\star, \phi}(\phi_h \mid s) = 1$ for all $(s, h) \in \mathcal{M} \times [H]$ (see (122)), (ii) follows from the fact $d_1^{\star, P}(s, \phi_1) = 1$ for all $s \in \mathcal{M}$, (iii) is verified in (132), and the last equality arises from the definition in (130).

- Similarly, for $h = 2, 3, \cdots, H$, we arrive at

$$\max_{(s,a,P)\in\mathcal{S}_{\mathrm{one}}\times\mathcal{A}_{\mathrm{one}}\times\mathcal{U}^\sigma(P^\phi)} \frac{\min\left\{d_h^{\star,P}(s,a), \frac{1}{4SA}\right\}}{d_h^{\mathsf{n},P^{\phi,f}}(s,a)}$$

$$\overset{(\mathrm{i})}{=} \max_{(s,P)\in\mathcal{S}\times\mathcal{U}^\sigma(P^\phi)} \frac{\min\left\{d_h^{\star,P}(s,\phi_h), \frac{1}{4SA}\right\}}{d_h^{\mathsf{n},P^{\phi,f}}(s,\phi_h)}$$

$$\leq \max_{(s,P)\in\mathcal{M}\times\mathcal{U}^\sigma(P^\phi)} \frac{1}{4SA d_h^{\mathsf{n},P^{\phi,f}}(s,\phi_h)}$$

$$\overset{(\mathrm{ii})}{\leq} \max_{s\in\mathcal{M}} \frac{1}{2SA\overline{\varrho}(s)} = 2C, \tag{183}$$

where (i) holds by the optimal policy in (122) and the trivial fact that $d_h^{\star,P}(s) = 0$ for all $s \in \mathcal{N}$ (see (133) and (125)), (ii) arises from (132), and the last equality comes from (130).

Combining the above cases, we complete the proof by

$$\frac{C}{2} \leq C_{\mathsf{r}}^\star = \max_{(h,s,a,P)\in[H]\times\mathcal{S}_{\mathrm{one}}\times\mathcal{A}_{\mathrm{one}}\times\mathcal{U}^\sigma(P^\phi)} \frac{\min\left\{d_h^{\star,P}(s,a), \frac{1}{4SA}\right\}}{d_h^{\mathsf{n},P^{\phi,f}}(s,a)} \leq C.$$

## E.3 PROOF OF CLAIM (139)

Recalling (121a) and (123), we first consider a more general form

$$V_h^{\star,\sigma^+,f,\phi}(m_f) - V_h^{\widetilde{\mu},\sigma^+,f,\phi}(m_f)$$

$$= p^{\mathrm{inf}} V_{h+1}^{\star,\sigma^+,f,\phi}(n_f) + (1 - p^{\mathrm{inf}}) V_{h+1}^{\star,\sigma^+,f,\phi}(m_f)$$

$$\quad - \left(m_h^{\widetilde{\mu},f,\phi} V_{h+1}^{\widetilde{\mu},\sigma^+,f,\phi}(n_f) + \left[1 - m_h^{\widetilde{\mu},f,\phi}\right] V_{h+1}^{\widetilde{\mu},\sigma^+,f,\phi}(m_f)\right)$$

$$= \left(p^{\mathrm{inf}} - m_h^{\widetilde{\mu},f,\phi}\right) V_{h+1}^{\star,\sigma^+,f,\phi}(n_f) + m_h^{\widetilde{\mu},f,\phi}\left(V_{h+1}^{\star,\sigma^+,f,\phi}(n_f) - V_{h+1}^{\widetilde{\mu},\sigma^+,f,\phi}(n_f)\right)$$

$$\quad + (1 - p^{\mathrm{inf}})\left(V_{h+1}^{\star,\sigma^+,f,\phi}(m_f) - V_{h+1}^{\widetilde{\mu},\sigma^+,f,\phi}(m_f)\right) - \left(p^{\mathrm{inf}} - m_h^{\widetilde{\mu},f,\phi}\right) V_{h+1}^{\widetilde{\mu},\sigma^+,f,\phi}(m_f)$$

$$= m_h^{\widetilde{\mu},f,\phi}\left(V_{h+1}^{\star,\sigma^+,f,\phi}(n_f) - V_{h+1}^{\widetilde{\mu},\sigma^+,f,\phi}(n_f)\right) + (1 - p^{\mathrm{inf}})\left(V_{h+1}^{\star,\sigma^+,f,\phi}(m_f) - V_{h+1}^{\widetilde{\mu},\sigma^+,f,\phi}(m_f)\right)$$

$$\quad + \left(p^{\mathrm{inf}} - m_h^{\widetilde{\mu},f,\phi}\right)\left(V_{h+1}^{\star,\sigma^+,f,\phi}(n_f) - V_{h+1}^{\star,\sigma^+,f,\phi}(m_f)\right)$$

$$\geq (1 - p^{\mathrm{inf}})\left(V_{h+1}^{\star,\sigma^+,f,\phi}(m_f) - V_{h+1}^{\widetilde{\mu},\sigma^+,f,\phi}(m_f)\right)$$

$$\quad + \left(p^{\mathrm{inf}} - m_h^{\widetilde{\mu},f,\phi}\right)\left(V_{h+1}^{\star,\sigma^+,f,\phi}(n_f) - V_{h+1}^{\star,\sigma^+,f,\phi}(m_f)\right)$$

$$\geq (1 - p^{\mathrm{inf}})\left(V_{h+1}^{\star,\sigma^+,f,\phi}(m_f) - V_{h+1}^{\widetilde{\mu},\sigma^+,f,\phi}(m_f)\right)$$

$$\quad + \frac{1}{2}(p - q)\left\|\widetilde{\mu}_h^{\star,f,\phi}(\cdot\,|\,m_f) - \widetilde{\mu}_h(\cdot\,|\,m_f)\right\|_1 \left(V_{h+1}^{\star,\sigma^+,f,\phi}(n_f) - V_{h+1}^{\star,\sigma^+,f,\phi}(m_f)\right), \tag{184}$$

where the last inequality holds since

$$p^{\mathrm{inf}} - m_h^{\widetilde{\mu},f,\phi} = \left(p^{\mathrm{inf}} - q^{\mathrm{inf}}\right)\left(1 - \widetilde{\mu}_h(\phi_h\,|\,m_f)\right)$$

$$= (p - q)\left(1 - \widetilde{\mu}_h(\phi_h\,|\,m_f)\right)$$

$$= \frac{1}{2}(p - q)\left(1 - \widetilde{\mu}_h(\phi_h\,|\,m_f) + \widetilde{\mu}_h(1 - \phi_h\,|\,m_f)\right)$$

$$= \frac{1}{2}(p - q)\left\|\widetilde{\mu}_h^{\star,f,\phi}(\cdot\,|\,m_f) - \widetilde{\mu}_h(\cdot\,|\,m_f)\right\|_1, \tag{185}$$

with the first equality holding by (120) and the second existing by (118).

To further control (184),

$$V_h^{\star,\sigma^+,f,\phi}(n_f) - V_h^{\star,\sigma^+,f,\phi}(m_f)$$

$$\overset{(i)}{=} 1 + (1-\sigma^+)V_{h+1}^{\star,\sigma^+,f,\phi}(n_f) + \sigma^+ V_{h+1}^{\star,\sigma^+,f,\phi}(m_f)$$

$$\quad - \left( p^{\inf} V_{h+1}^{\star,\sigma^+,f,\phi}(n_f) + (1-p^{\inf})V_{h+1}^{\star,\sigma^+,f,\phi}(m_f) \right)$$

$$= 1 + (1 - p^{\inf} - \sigma^+)\left( V_{h+1}^{\star,\sigma^+,f,\phi}(n_f) - V_{h+1}^{\star,\sigma^+,f,\phi}(m_f) \right)$$

$$\overset{(ii)}{=} 1 + (1-p)\left( V_{h+1}^{\star,\sigma^+,f,\phi}(n_f) - V_{h+1}^{\star,\sigma^+,f,\phi}(m_f) \right)$$

$$= \cdots = \sum_{j=0}^{H-h}(1-p)^j, \tag{186}$$

where (i) follows from Lemma 8 and (ii) holds by (118). Then, we consider two cases w.r.t. the uncertainty level $\sigma^+$ to control (186), respectively:

- *When* $0 < \sigma^+ \le \frac{c_2}{2H}$: Recall $p = \frac{c_2}{H}$ if $\sigma^+ \le \frac{c_2}{2H}$. In this case, applying (186), we have

$$V_h^{\star,\sigma^+,f,\phi}(n_f) - V_h^{\star,\sigma^+,f,\phi}(m_f)$$

$$= \sum_{j=0}^{H-h}(1-p)^j \ge \sum_{j=0}^{H-h}\left(1-\frac{c_2}{H}\right)^j = \frac{1-\left(1-\frac{c_2}{H}\right)^{H-h+1}}{c_2/H} \ge \frac{2c_2(H-h+1)}{3}. \tag{187}$$

Here, the final inequality holds by observing

$$\left(1-\frac{c_2}{H}\right)^{H-h+1} \le \exp\left(-\frac{c_2(H-h+1)}{H}\right) \le 1 - \frac{2c_2(H-h+1)}{3H}, \tag{188}$$

where the first inequality holds by noticing $c_2 < \frac{1}{2}$ and then $1-x \le \exp(-x)$, and the last inequality holds by $\exp(-x) \le 1 - \frac{2x}{3}$ for any $0 \le x \le \frac{1}{2}$.

Plugging above fact in (187) back to (184), we arrive at

$$V_h^{\star,\sigma^+,f,\phi}(m_f) - V_h^{\widetilde{\mu},\sigma^+,f,\phi}(m_f)$$

$$\ge (1-p^{\inf})\left( V_{h+1}^{\star,\sigma^+,f,\phi}(m_f) - V_{h+1}^{\widetilde{\mu},\sigma^+,f,\phi}(m_f) \right)$$

$$\quad + \frac{1}{2}(p-q)\big\|\widetilde{\mu}_h^{\star,f,\phi}(\cdot\,|\,m_f) - \widetilde{\mu}_h(\cdot\,|\,m_f)\big\|_1 \frac{2c_2(H-h+1)}{3}. \tag{189}$$

Then invoking the assumption

$$\sum_{h=1}^{H}\big\|\widetilde{\mu}_h(\cdot\,|\,m_f) - \widetilde{\mu}_h^{\star,f,\phi}(\cdot\,|\,m_f)\big\|_1 \ge \frac{H}{8} \tag{190}$$

in (138) and applying (189) recursively for $h = 1, 2, \cdots, H$ yields

$$V_1^{\star,\sigma^+,f,\phi}(m_f) - V_1^{\widetilde{\mu},\sigma^+,f,\phi}(m_f)$$

$$\ge \frac{c_2}{3}\sum_{h=1}^{H}(1-p^{\inf})^{h-1}(p-q)(H-h+1)\big\|\widetilde{\mu}_h^{\star,f,\phi}(\cdot\,|\,m_f) - \widetilde{\mu}_h(\cdot\,|\,m_f)\big\|_1$$

$$\overset{(i)}{\ge} \frac{c_2}{3}\sum_{h=1}^{H}(1-\frac{c_2}{H})^{h-1}(p-q)(H-h+1)\big\|\widetilde{\mu}_h^{\star,f,\phi}(\cdot\,|\,m_f) - \widetilde{\mu}_h(\cdot\,|\,m_f)\big\|_1$$

$$\overset{(ii)}{\ge} \frac{c_2}{6}\sum_{h=1}^{H}(p-q)(H-h+1)\big\|\widetilde{\mu}_h^{\star,f,\phi}(\cdot\,|\,m_f) - \widetilde{\mu}_h(\cdot\,|\,m_f)\big\|_1$$

$$\overset{(iii)}{=} \frac{c_2\Delta}{6}\sum_{h=1}^{H} h\big\|\widetilde{\mu}_{H-h+1}^{\star,f,\phi}(\cdot\,|\,m_f) - \widetilde{\mu}_{H-h+1}(\cdot\,|\,m_f)\big\|_1$$

$$\overset{(iv)}{\ge} \frac{c_2\Delta}{6}\sum_{h=1}^{\lfloor H/16 \rfloor} 2h \ge \frac{c_2\Delta}{6}\lfloor H/16 \rfloor\left(\lfloor H/16 \rfloor + 1\right), \tag{191}$$

where (i) follows from $1 - p^{\text{inf}} \geq 1 - p = 1 - \frac{c_2}{H}$, and (ii) holds by

$$\forall h \in [H] : \quad (1 - \frac{c_2}{H})^{h-1} \geq (1 - \frac{c_2}{H})^H \geq \frac{1}{2}b \tag{192}$$

as long as $c_2 \leq \frac{1}{2}$. Here, (iii) arises from the definition of $p, q$ in (126); (iv) can be verified by the fact that for any series $0 \leq m_1, m_2, \cdots, m_H \leq m_{\max}$ that obeys $\sum_{h=1}^H m_h \geq y$, one has

$$\sum_{h=1}^H m_h h \geq \sum_{h=1}^{\lfloor m_{\max}/n \rfloor} m_{\max} h, \tag{193}$$

and taking $m_h = \left\| \widetilde{\mu}_{H-h+1}(\cdot \,|\, m_f) - \widetilde{\mu}_{H-h+1}^{\star, f, \phi}(\cdot \,|\, m_f) \right\|_1 \leq 2 = m_{\max}$ and $n = \frac{H}{8}$. Consequently, observed from (191), the following inequality holds

$$V_1^{\star, \sigma^+, f, \phi}(m_f) - V_1^{\widetilde{\mu}, \sigma^+, f, \phi}(m_f) \geq \frac{c_2 \Delta}{6} \lfloor H/16 \rfloor (\lfloor H/16 \rfloor + 1) \geq c_3 \Delta H^2 > \varepsilon \tag{194}$$

for some small enough constant $c_3$ and letting $\Delta = \frac{\varepsilon}{c_3 H^2}$.

- *When $\frac{c_2}{2H} < \sigma^+ \leq 1 - c_0$:* Similarly, recalling $p = \left(1 + \frac{c_1}{H}\right) \sigma^+$ if $\sigma^+ > \frac{c_2}{2H}$ and invoking (186) gives

$$V_h^{\star, \sigma^+, f, \phi}(n_f) - V_h^{\star, \sigma^+, f, \phi}(m_f) = \sum_{j=0}^{H-h} (1 - p)^j = \sum_{j=0}^{H-h} \left(1 - \left(1 + \frac{c_1}{H}\right) \sigma^+\right)^j$$

$$\geq \frac{1 - \left(1 - (1 + \frac{c_1}{H})\sigma^+\right)^{H-h+1}}{(1 + \frac{c_1}{H})\sigma^+}$$

$$\geq \frac{c_2(H - h + 1)}{3\sigma^+ H}, \tag{195}$$

where the final inequality holds by observing

$$\left(1 - \left(1 + \frac{c_1}{H}\right) \sigma^+\right)^{H-h+1} \leq \exp\left(-\left(1 + \frac{c_1}{H}\right) \sigma^+ (H - h + 1)\right)$$

$$\overset{(i)}{\leq} \exp\left(-\frac{c_2}{2H} \left(1 + \frac{c_1}{H}\right) (H - h + 1)\right)$$

$$\leq 1 - \left(1 + \frac{c_1}{H}\right) \frac{c_2(H - h + 1)}{3H}. \tag{196}$$

Here, (i) holds by observing $\frac{c_2}{2H} < \sigma^+$, and the last inequality holds by $\left(1 + \frac{c_1}{H}\right) \leq 2$, $c_2 \leq \frac{1}{2}$, and the fact $\exp(-x) \leq 1 - \frac{2x}{3}$ for any $0 \leq x \leq \frac{1}{2}$.

Plugging above fact in (195) back to (184) gives

$$V_h^{\star, \sigma^+, f, \phi}(m_f) - V_h^{\widetilde{\mu}, \sigma^+, f, \phi}(m_f)$$

$$\geq (1 - p^{\text{inf}}) \left( V_{h+1}^{\star, \sigma^+, f, \phi}(m_f) - V_{h+1}^{\widetilde{\mu}, \sigma^+, f, \phi}(m_f) \right)$$

$$+ \frac{1}{2}(p - q) \left\| \widetilde{\mu}_h^{\star, f, \phi}(\cdot \,|\, m_f) - \widetilde{\mu}_h(\cdot \,|\, m_f) \right\|_1 \frac{c_2(H - h + 1)}{3\sigma^+ H}. \tag{197}$$

Following the same routine to achieve (191), applying (197) recursively for $h = 1, 2, \cdots, H$ gives

$$V_1^{\star, \sigma^+, f, \phi}(m_f) - V_1^{\widetilde{\mu}, \sigma^+, f, \phi}(m_f)$$

$$\geq \sum_{h=1}^H (1 - p^{\text{inf}})^{h-1} (p - q) \frac{c_2(H - h + 1)}{6\sigma^+ H} \left\| \widetilde{\mu}_h^{\star, f, \phi}(\cdot \,|\, m_f) - \widetilde{\mu}_h(\cdot \,|\, m_f) \right\|_1$$

$$\overset{(i)}{=} \frac{c_2(p - q)}{6\sigma^+ H} \sum_{h=1}^H (1 - \frac{c_1}{H})^{h-1} (H - h + 1) \left\| \widetilde{\mu}_h^{\star, f, \phi}(\cdot \,|\, m_f) - \widetilde{\mu}_h(\cdot \,|\, m_f) \right\|_1$$

$$\overset{(ii)}{\geq} \frac{c_2 \Delta}{12\sigma^+ H} \lfloor H/16 \rfloor (\lfloor H/16 \rfloor + 1), \tag{198}$$

where (i) follows from $1 - p^{\text{inf}} = 1 - (p - \sigma^+) = 1 - \frac{c_1}{H}\sigma^+$, and (ii) holds by letting $c_1 \leq \frac{1}{2}$ and following the same routine of (191).

Consequently, (198) yields

$$V_1^{\star,\sigma^+,f,\phi}(m_f) - V_1^{\widetilde{\mu},\sigma^+,f,\phi}(m_f) \geq \frac{c_2\Delta}{12\sigma^+H}\lfloor H/16 \rfloor(\lfloor H/16 \rfloor + 1) \geq \frac{c_4\Delta H}{\sigma^+} > \varepsilon, \tag{199}$$

which holds for some small enough constant $c_4$ and letting $\Delta = \frac{\sigma^+\varepsilon}{c_4H}$.

# F  MULTIPLAYER GENERAL-SUM MARKOV GAMES

In this section, we extend RTZ-VI-LCB to the setting of multi-player general-sum Markov games and present the corresponding theoretical guarantees.

## F.1  PROBLEM FORMULATION

A robust general-sum Markov game is a tuple $\mathcal{M}(\mathcal{S}, \{\mathcal{A}_i\}_{i=1}^m, H, \{\mathcal{U}_\rho^{\sigma_i}(P^0)\}_{i=1}^m, \{r_i\}_{i=1}^m)$ with $m$ players, where $\mathcal{S}$ denotes the state space and $H$ is the horizon length. We have $m$ different action spaces, where $\mathcal{A}_i$ is the action space for the $i^{\text{th}}$ player and $|\mathcal{A}_i| = A_i$. We let $\mathcal{A} = \mathcal{A}_1 \times \cdots \times \mathcal{A}_m$ denote the joint action space, and let $\boldsymbol{a} := (a_1, \cdots, a_m) \in \mathcal{A}$ denote the (tuple of) joint actions by all $m$ players. A notable deviation from standard MGs is that: for $1 \leq i \leq m$, instead of assuming a fixed transition kernel, each $i^{\text{th}}$ player anticipates that the transition kernel is allowed to be chosen arbitrarily from a prescribed uncertainty set $\mathcal{U}_\rho^{\sigma_i}(P^0)$. Here, the uncertainty set $\mathcal{U}_\rho^{\sigma_i}(P^0)$ is constructed centered on $P^0(\cdot|s,\boldsymbol{a})$, with its size and shape defined by a certain distance metric $\rho$ and a radius parameter $\sigma_i > 0$. $r_i = \{r_{h,i}\}_{h \in [H]}$ is a collection of reward functions for the $i^{\text{th}}$ player, so that $r_{h,i}(s,\boldsymbol{a})$ gives the reward received by the $i^{\text{th}}$ player if actions $\boldsymbol{a}$ are taken at state $s$ at step $h$.

The policy of the $i^{\text{th}}$ player is denoted as $\pi_i := \{\pi_{h,i} : \mathcal{S} \to \Delta_{\mathcal{A}_i}\}_{h \in [H]}$. We denote the product policy of all players as $\pi := \pi_1 \times \cdots \times \pi_M$, and denote the policy of all players except the $i^{\text{th}}$ player as $\pi_{-i}$. We define $V_{h,i}^\pi(s)$ as the expected cumulative reward that will be received by the $i^{\text{th}}$ player if starting at state $s$ at step $h$ and all players follow policy $\pi$. For any strategy $\pi_{-i}$, there also exists a *robust best response* of the $i^{\text{th}}$ player, which is a policy $\mu^\star(\pi_{-i})$ satisfying $V_{h,i}^{\mu^\star(\pi_{-i}),\pi_{-i},\sigma_i}(s) = \sup_{\pi_i} V_{h,i}^{\pi_i,\pi_{-i},\sigma_i}(s)$ for any $(s,h) \in \mathcal{S} \times [H]$. For convenience, we denote $V_{h,i}^{\star,\pi_{-i},\sigma_i} := V_{h,i}^{\mu^\star(\pi_{-i}),\pi_{-i},\sigma_i}$. The $Q$-functions of the robust best response can be defined similarly.

Similar to the definition of behavior policy $(\mu^{\text{n}}, \nu^{\text{n}})$, we use the short-hand notation for the occupancy distribution w.r.t. the behavior policy $\pi^{\text{n}} = (\pi_i^{\text{n}}, \pi_{-i}^{\text{n}})$ as: $\forall(h,s,\boldsymbol{a}) \in [H] \times \mathcal{S} \times \mathcal{A}$,

$$d_h^{\text{n},P^0}(s) = d_h^{\pi^{\text{n}},P^0}(s) := \mathbb{P}(s_h = s \,|\, s_1 \sim \varrho^{\text{n}}, \pi^{\text{n}}, P^0); \tag{200a}$$

$$d_h^{\text{n},P^0}(s,\boldsymbol{a}) = d_h^{\pi^{\text{n}},P^0}(s,\boldsymbol{a}) := \mathbb{P}(s_h = s \,|\, s_1 \sim \varrho^{\text{n}}, \pi^{\text{n}}, P^0)\,\pi^{\text{n}}(\boldsymbol{a}\,|\,s). \tag{200b}$$

Similarly, for any product policy $\pi = (\pi_i, \pi_{-i})$, there is, $\forall(h,s,\boldsymbol{a}) \in [H] \times \mathcal{S} \times \mathcal{A}$

$$d_h^{\pi_i,\pi_{-i},P}(s) := \mathbb{P}(s_h = s \,|\, s_1 \sim \varrho, \pi, P); \tag{201a}$$

$$d_h^{\pi_i,\pi_{-i},P}(s,\boldsymbol{a}) := \mathbb{P}(s_h = s \,|\, s_1 \sim \varrho, \pi, P)\,\pi_{i,h}(a_i\,|\,s)\,\pi_{-i,h}(\boldsymbol{a}_{-i}\,|\,s). \tag{201b}$$

Therefore, the robust variant of standard solution concepts—robust NE for Robust multi-player general-sum MGs is introuced as follows: A product policy $\pi$ is considered a *robust NE* if

$$\forall(s) \in \mathcal{S}, \quad V_1^{\pi,\sigma_i}(s) = V_h^{\star,\pi_{-i},\sigma^+}(s). \tag{202}$$

A robust NE signifies that given the product policy $(\pi)$ of the opponents, no player can enhance their outcome by deviating from their current policy unilaterally when each player accounts for the worst-case scenario within their uncertainty set $\mathcal{U}_\rho^{\sigma_i}(P^0)$ for all $i = 1, 2, \cdots, m$.

Since finding exact robust equilibria can be complex and may not always be feasible, practitioners often seek approximate equilibria. In this context, a product policy $\pi \in \Delta(\mathcal{A})$ can be termed an $\varepsilon$-*robust NE* if

$$\text{Gap}(\pi) := \max\left\{\left\{V_{i,1}^{\star,\pi_{-i},\sigma_i}(\varrho) - V_{i,1}^{\pi,\sigma_i}(\varrho)\right\}_{i=1}^m\right\} \leq \varepsilon, \tag{203}$$

where

$$V_1^{\star,\pi_{-i},\sigma_i}(\varrho) = \mathbb{E}_{s\sim\varrho}V_1^{\star,\pi_{-i},\sigma_i}(s), \qquad \text{and} \qquad V_1^{\star,\sigma_i}(\varrho) = \mathbb{E}_{s\sim\varrho}V_1^{\star,\sigma_i}(s).$$

The existence of robust NE has been established for general divergence functions used in the uncertainty set by Blanchet et al. (2024).

**Learning objective**    With a dataset collected from the nominal environment, our objective is to find a solution among the $\varepsilon$-robust NEs for the robust multi-player general-sum MG $\mathcal{MG}_r$ with respect to a specified uncertainty set $\mathcal{U}(P^0)$ around the nominal kernel, while minimizing the number of samples required under partial coverage of the state-action space.

### F.2    MULTI-RTZ-VI-LCB

Here we present the Multi-RTZ-VI-LCB algorithm in Algorithm 4, which is an extension of Algorithm 2 for multi-player general-sum Markov games.

According to the empirical frequencies of state transitions, we can naturally construct an empirical estimate $\widehat{P}^0 = \{\widehat{P}_h^0\}_{h=1}^H$ of $P^0$, where

$$\widehat{P}_h^0\left(s' \mid s, \boldsymbol{a}\right) = \begin{cases} \frac{1}{N_h(s,\boldsymbol{a})}\sum_{j=1}^N \mathbb{1}\left\{\left(s_j, \boldsymbol{a}_j, s_j'\right) = (s, \boldsymbol{a}, s')\right\}, & \text{if } N_h\left(s,\boldsymbol{a}\right) > 0; \\ \frac{1}{S}, & \text{if } N_h\left(s,\boldsymbol{a}\right) = 0, \end{cases} \tag{204}$$

$$\widehat{r}_{i,h}\left(s,\boldsymbol{a}\right) = \begin{cases} r_{i,h}\left(s,\boldsymbol{a}\right), & \text{if } N_h\left(s,\boldsymbol{a}\right) > 0; \\ 0, & \text{if } N_h\left(s,\boldsymbol{a}\right) = 0, \end{cases} \tag{205}$$

for any $(i, h, s, \boldsymbol{a}, s') \in [m] \times [H] \times \mathcal{S} \times \mathcal{A} \times \mathcal{B} \times \mathcal{S}$. Besides, $N_h(s, \boldsymbol{a})$ represents the total number of sample transitions from $(s, \boldsymbol{a})$ at step $h$, and

$$N_h(s, \boldsymbol{a}) := \sum_{j=1}^N \mathbb{1}\left\{(s_j, \boldsymbol{a}_j) = (s, \boldsymbol{a})\right\}. \tag{206}$$

Before the details of Multi-RTZ-VI-LCB, we extend Algorithm 1 as Algorithm 3, which reduces statistical dependencies and produces a distributionally equivalent dataset $\mathcal{D}_0$ with independent samples. Similar to Lemma 1, we present the following lemma concerning the dataset $\mathcal{D}_0$, whose proof is similar to the context in Appendix C.1.

---

**Algorithm 3:** Two-stage subsampling technique for Multi-RTZ-VI-LCB.

**1** **Input:** Dataset $\mathcal{D}$, probability $\delta$.

**2** **Step 1: Data Partitioning.** Split $\mathcal{D}$ into two equal-sized subsets, $\mathcal{D}^{\mathsf{m}}$ and $\mathcal{D}^{\mathsf{a}}$, each containing $K/2$ trajectories.

**3** **Step 2: Defining Transition Bounds.** For step $h$ and state $s$, denote the number of transitions from $\mathcal{D}^{\mathsf{m}}$ (resp. $\mathcal{D}^{\mathsf{a}}$) as $N_h^{\mathsf{m}}(s)$ (resp. $N_h^{\mathsf{a}}(s)$). Construct the trimmed count as:

$$N_h^{\mathsf{t}}(s) := \max\left\{N_h^{\mathsf{a}}(s) - 10\sqrt{N_h^{\mathsf{a}}(s)\log\frac{HS}{\delta}}, 0\right\}.$$

**4** **Step 3: Generating Subsampled Dataset.** Randomly sample transitions (quadruples of the form $(s, \boldsymbol{a}, h, s')$) from $\mathcal{D}^{\mathsf{m}}$ uniformly. For each $(s, h) \in \mathcal{S} \times [H]$, include $\min\{N_h^{\mathsf{t}}(s), N_h^{\mathsf{m}}(s)\}$ transitions in the new dataset $\mathcal{D}^{\mathsf{t}}$.

**5** **Output:** Set $\mathcal{D}_0 = \mathcal{D}^{\mathsf{t}}$.

---

**Lemma 9** *The dataset produced by the two-stage subsampling method is distributionally identical to $\mathcal{D}_0$ with probability at least $1 - 8\delta$, where $\{N_h(s, \boldsymbol{a})\}$ are independent of the sample transitions in $\mathcal{D}^0$ and obey: $\forall (h, s, \boldsymbol{a}) \in [H] \times \mathcal{S} \times \mathcal{A}$,*

$$N_h(s, \boldsymbol{a}) \geq \frac{K d_h^{\mathrm{n}}(s, \boldsymbol{a})}{8} - 5\sqrt{K d_h^{\mathrm{n}}(s, \boldsymbol{a}) \log \frac{KH}{\delta}}. \tag{207}$$

---

**Algorithm 4:** Multi-RTZ-VI-LCB.

---

**1 Initialization**: Set uncertainty levels $\sigma_i$ for $i = 1, 2, \cdots, m$; set $\widehat{V}_{i,h}^{\sigma_i}(s) = H$ and $\widehat{Q}_{i,h}^{\sigma_i}(s, \boldsymbol{a}) = H$ for all $(i, s, \boldsymbol{a}, h) \in [m] \times \mathcal{S} \times \mathcal{A} \times [H + 1]$.

**2 Compute** the empirical reward function $\widehat{r}$ using (13) and the empirical transition kernel $\widehat{P}_0$ using (12).

**3 for** $h = H, H - 1, \ldots, 1$ **do**

**4**     **for** *player* $i = 1, 2, \ldots, m$ **do**

**5**        **Update** the robust Q-value estimate as

$$\widehat{Q}_{i,h}^{\sigma_i}(s, \boldsymbol{a}) = \min\left\{\widehat{r}_{i,h}(s, \boldsymbol{a}) + \inf_{P \in \mathcal{U}^{\sigma_i}\left(\widehat{P}_{h,s,\boldsymbol{a}}^0\right)} P\widehat{V}_{i,h+1}^{\sigma_i} + \beta_{i,h}\left(s, \boldsymbol{a}, \widehat{V}_{i,h+1}^{\sigma_i}\right), H\right\},$$

with $\beta_{i,h}(s, \boldsymbol{a}, V) = \min\left\{\max\left\{\sqrt{\frac{C_{\mathrm{n}} \log \frac{KH}{\delta}}{N_h(s,\boldsymbol{a})} \mathsf{Var}_{\widehat{P}_{h,s,\boldsymbol{a}}^0}(V)}, \frac{2C_{\mathrm{n}} H \log \frac{KH}{\delta}}{N_h(s,\boldsymbol{a})}\right\}, H\right\}.$

**6**        **Compute** Nash policy for each $s \in \mathcal{S}$ as

$$\pi_h(s) = (\pi_{i,h}(s), \pi_{-i,h}(s)) = \mathsf{ComputNash}\left(\widehat{Q}_{i,h}^{\sigma_i}(s, \cdot)\right),$$

**7**        **Update** the robust value estimate for each $s \in \mathcal{S}$ as

$$\widehat{V}_{i,h}^{\sigma_i}(s) = \mathbb{E}_{\boldsymbol{a} \sim \pi_h(s)}\left[\widehat{Q}_{i,h}^{\sigma_i}(s, \boldsymbol{a})\right].$$

**8 Output**: The product policy $\hat{\pi}(s) = \{\pi_h(s)\}_{h=1}^{H}$ with $\pi_h(s) = \prod_{i=1}^{m} \pi_{i,h}(s)$.

---

Based on Algorithm 4, we propose a model-based approach for solving robust multi-player general-sum MGs using an approximate $\widehat{P}^0$ for $P^0$, as summarized in Algorithm 4.

Similar to (18), we can tackle the multi-player general-sum MGs problem as:

$$\inf_{P \in \mathcal{U}^{\sigma_i}\left(\widehat{P}_{h,s,\boldsymbol{a}}^0\right)} P\widehat{V}_{i,h+1}^{\sigma_i} = \max_{\alpha \in [\min_s \widehat{V}_{i,h+1}^{\sigma_i}, \max_s \widehat{V}_{i,h+1}^{\sigma_i}]} \left\{\widehat{P}_{h,s,\boldsymbol{a}}^0\left[\widehat{V}_{i,h+1}^{\sigma_i}\right]_\alpha - \sigma_i\left(\alpha - \min_{s'}\left[\widehat{V}_{i,h+1}^{\sigma_i}\right]_\alpha(s')\right)\right\}. \tag{208}$$

where $\left[\widehat{V}_{i,h+1}^{\sigma_i}\right]_\alpha$ respectively denote the clipped versions of $\widehat{V}_{i,h+1}^{\sigma_i} \in \mathbb{R}^S$ based on some level $\alpha \geq 0$, as follows.

$$\left[\widehat{V}_{i,h+1}^{\sigma_i}\right]_\alpha(s) := \begin{cases} \widehat{V}_{i,h+1}^{\sigma_i}(s), & \text{if } \widehat{V}_{i,h+1}^{\sigma_i}(s) > \alpha; \\ \alpha. & \text{otherwise;} \end{cases} \tag{209}$$

F.3    ANALYSIS OF MULTI-ME-NASH-QL

In this subsection, we prove Theorem 3, which can separated into three steps as the proof of Theorem 1.

First of all, similar to Assumption 1, we measure the distributional discrepancy between the historical data and the target data to assess the effectiveness of the historical dataset for achieving the desired goal. We propose a novel assumption for robust multi-agent general-sum MGs as:

**Assumption 2 (Robust multiple clipped concentrability)** *The behavior policies of the historical dataset $\mathcal{D}$ satisfies*

$$
\max\left\{\left\{\sup_{(\pi_{-i},s,\boldsymbol{a},h,P)\in\Delta(\mathcal{A}_{-i})\times\mathcal{S}\times\mathcal{A}\times[H]\times\mathcal{U}^{\sigma_i}(P^0)}\frac{\min\left\{d_h^{\pi_i^\star,\pi_{-i},P}(s,\boldsymbol{a}),\frac{1}{S\sum_{i=1}^m A_i}\right\}}{d_h^{\mathsf{n},P^0}(s,\boldsymbol{a})}\right\}_{i=1}^m\right\}\le C_{\mathrm{mr}}^\star
$$
(210)

**Step 1: decoupling statistical dependency**   Before bounding $\mathrm{Gap}(\widehat{\pi})$, we introduce an important lemma whose proof is similar to Lemma 3 in Appendix C.3, quantifying the difference between $\widehat{P}$ and $P$ when projected in the direction of the value function.

**Lemma 10** *Instate the assumptions in Theorem 3. Consider any vector $V\in\mathbb{R}^S$ with $\|V\|_\infty\le H$ for all $(i,h,s,\boldsymbol{a})\in[m]\times[H]\times\mathcal{S}\times\mathcal{A}$ satisfying $N_h(s,\boldsymbol{a})>0$. With probability at least $1-\delta$, one has*

$$
\left|\inf_{P\in\mathcal{U}^{\sigma_i}(\widehat{P}_{h,s,\boldsymbol{a}}^0)}PV-\inf_{P\in\mathcal{U}^{\sigma_i}(P_{h,s,\boldsymbol{a}}^0)}PV\right|\le C_4\sqrt{\frac{1}{N_h(s,\boldsymbol{a})}\mathsf{Var}_{\widehat{P}_{h,s,\boldsymbol{a}}^0}(V)\log\frac{KH}{\delta}}+C_4\frac{H\log\frac{KH}{\delta}}{N_h(s,\boldsymbol{a})}
$$
(211)

*for some sufficiently large constant $C_4>0$, and*

$$
\mathsf{Var}_{\widehat{P}_{h,s,\boldsymbol{a}}^0}(V)\le 2\mathsf{Var}_{P_{h,s,\boldsymbol{a}}^0}(V)+O\left(\frac{H^2}{N_h(s,\boldsymbol{a})}\log\frac{KH}{\delta}\right).
$$
(212)

With Lemma 10, we can now have

$$
\left|\inf_{\mathcal{P}\in\mathcal{U}^{\sigma_i}(\widehat{P}_{h,s,\boldsymbol{a}}^0)}PV-\inf_{\mathcal{P}\in\mathcal{U}^{\sigma_i}(P_{h,s,\boldsymbol{a}}^0)}PV\right|\le\beta_h(s,\boldsymbol{a},V)
$$
(213)

for any $(i,h,s,\boldsymbol{a})\in[m]\times[H]\times\mathcal{S}\times\mathcal{A}$ satisfying $N_h(s,\boldsymbol{a})\ge 1$.

Therefore, we conclude that $\widehat{Q}_{i,h}^{\sigma_i}(s,\boldsymbol{a})$ is an optimistic estimation of $\widehat{Q}_{i,h}^{\pi,\sigma_i}(s,\boldsymbol{a})$ for any $i=1,2,\cdots,m$, which is summarized below, whose proof is similar to Lemma 4 in Appendix C.4.

**Lemma 11** *With probability exceeding $1-\delta$, it holds that*

$$
\widehat{Q}_{i,h}^{\sigma_i}(s,\boldsymbol{a})\ge Q_{i,h}^{\star,\widehat{\pi}_{-i},\sigma_i}(s,\boldsymbol{a})\qquad and\qquad\widehat{V}_{i,h}^{\sigma_i}(s)\ge V_{i,h}^{\star,\widehat{\pi}_{-i},\sigma_i}(s).
$$
(214)

Besides, we introduce another key lemma highlighting the difference between robust multi-player general-sum MGs and standard multi-player general-sum MGs from the same idea of Lemma 5, as shown below.

**Lemma 12** *Consider any multi-player general-sum MGs $\mathcal{MG}_{\mathsf{r}}=\{\mathcal{S},\{\mathcal{A}_i\}_{i=1}^m,H,\{\mathcal{U}_\rho^{\sigma_i}(P^0)\}_{i=1}^m,\{r_i\}_{i=1}^m\}$ and the uncertainty set $\{\mathcal{U}_\rho^{\sigma_i}(P^0)\}_{i=1}^m(\cdot)$ with TV distance. The optimistic robust value function estimate $\widehat{V}_{i,h}^{\sigma_i}$:*

$$
\forall(i,h)\in[m]\times[H]:\quad\max_{s\in\mathcal{S}}\widehat{V}_{i,h}^{\sigma_i}-\min_{s\in\mathcal{S}}\widehat{V}_{i,h}^{\sigma_i}\le\min\left\{\frac{(H+1)\left(1-(1-\sigma_i)^{H-h}\right)}{\sigma_i},H\right\}.
$$

**Step 2: decomposing the error $\mathrm{Gap}(\widehat{\pi})$**   The goal of our algorithm is to output an $\varepsilon$-robust NE policy $(\widehat{\pi})$ satisfying $\mathrm{Gap}(\widehat{\pi})$ in (203), i.e.,

$$
\mathrm{Gap}(\widehat{\pi}):=\max\left\{\left\{V_{i,1}^{\star,\pi_{-i},\sigma_i}(\varrho)-V_{i,1}^{\pi,\sigma_i}(\varrho)\right\}_{i=1}^m\right\}\le\varepsilon.
$$

According to the relationship in Lemma 11, under the definition of $\mathcal{A}_{-i}:=\mathcal{A}_1\times\cdots\times\mathcal{A}_{i-1}\times\mathcal{A}_{i+1}\times\cdots\times\mathcal{A}_m$, we obtain

$$
V_h^{\star,\widehat{\pi}_{-i,h},\sigma^+}(s)\le\widehat{V}_{i,h}^{\sigma_i}(s)=\max_{\pi_i\in\Delta(\mathcal{A}_i)}\min_{\pi_{-i}\in\Delta(\mathcal{A}_{-i})}\mathbb{E}_{\boldsymbol{a}\sim\pi}\left[\widehat{Q}_{i,h}^{\sigma_i}(s,\boldsymbol{a})\right]
$$
$$
\le\min_{\max\pi_{-i}\in\Delta(\mathcal{A}_{-i})}\mathbb{E}_{\boldsymbol{a}\sim(\pi_i^\star(s),\pi_{-i}(s))}\left[Q_{i,h}^{\sigma_i}(s,\boldsymbol{a})\right],
$$
(215)

where the first equality comes from line 8 in Algorithm 4. Therefore, there exists a deterministic policy $\pi^{\mathsf{d}}_{-i} : \mathcal{S} \leftarrow \Delta(\mathcal{A}_{-i})$ satisfying that for any $s \in \mathcal{S}$

$$\pi^{\mathsf{d}}_{-i}(s) := \arg\min_{\pi_{-i} \in \Delta(\mathcal{A}_i)} \mathbb{E}_{\boldsymbol{a} \sim (\pi^{\star}_i(s), \pi_{-i}(s))} \left[ Q^{\sigma_i}_{i,h}(s, \boldsymbol{a}) \right]. \tag{216}$$

Before starting, we introduce several useful notations:

- The state-action space covered by the behavior policy $\pi^{\mathsf{n}}$ in the nominal transition kernel $P^0$ is denoted as

$$\mathcal{C}^{\mathsf{n}} = \{(h, s, \boldsymbol{a}) : d^{\mathsf{n}}_h(s, \boldsymbol{a}) > 0\}. \tag{217}$$

- The set of potential state occupancy distributions w.r.t. the policy $(\pi^{\star}_i(s), \pi^{\mathsf{b}}_{-i}(s))$ in a model within the uncertainty set $P \in \mathcal{U}^{\sigma_i}(P^0)$ for any $(i, h) \in [m] \times [H]$ is denoted as

$$\mathcal{D}^{\mathsf{pi}}_{i,h} := \left\{ \left[ d^{\pi^{\star}_i(s), \pi^{\mathsf{b}}_{-i}(s), P}_h(s) \right]_{s \in \mathcal{S}} : P \in \mathcal{U}^{\sigma_i}(P^0) \right\}; \tag{218}$$

$$\mathcal{D}^{\mathsf{pai}}_{i,h} := \left\{ \left[ d^{\pi^{\star}_i(s), \pi^{\mathsf{b}}_{-i}(s), P}_h(s, \boldsymbol{a}) \right]_{(s,\boldsymbol{a}) \in \mathcal{S} \times \mathcal{A}} : P \in \mathcal{U}^{\sigma_i}(P^0) \right\}. \tag{219}$$

- For convenience and without ambiguity, we introduce an additional notation for $(i, h) \in [m] \times [H]$ as

$$\beta^{\pi^{\star}_i, \pi^{\mathsf{b}}_{-i}}_{i,h}(s) = \mathbb{E}_{\boldsymbol{a} \sim (\pi^{\star}_i(s), \pi^{\mathsf{b}}_{-i}(s))} \beta_{i,h}\left(s, \boldsymbol{a}, \widehat{V}^{\sigma_i}_{i,h+1}\right).$$

In particular, the vector $\beta^{\pi^{\star}_i, \pi^{\mathsf{b}}_{-i}}_{i,h} \in \mathbb{R}^S$ is defined with its $s$-th item given by $\beta^{\pi^{\star}_i, \pi^{\mathsf{b}}_{-i}}_{i,h}(s)$.

- Similarly, we can define the notation related to rewards for $(i, h) \in [m] \times [H]$ as

$$\widehat{r}^{\pi^{\star}_i, \pi^{\mathsf{b}}_{-i}}_{i,h}(s) = \mathbb{E}_{\boldsymbol{a} \sim (\pi^{\star}_i(s), \pi^{\mathsf{b}}_{-i}(s))} \widehat{r}_{i,h}(s, \boldsymbol{a}).$$

According to the update rule in line 4 in Algorithm 4 and robust Bellman equality similar to (31), we derive

$$V^{\star, \widehat{\pi}_{-i}, \sigma^+}_{i,h}(s) - V^{\widehat{\pi}, \sigma^+}_{i,h}(s)$$

$$\leq \widehat{V}^{\sigma_i}_{i,h}(s) - V^{\pi^{\star}_i, \pi^{\mathsf{b}}_{-i}, \sigma^+}_h(s)$$

$$\leq \mathbb{E}_{\boldsymbol{a} \sim (\pi^{\star}_i(s), \pi^{\mathsf{b}}_{-i}(s))} \inf_{P \in \mathcal{U}^{\sigma_i}(\widehat{P}^0_{h,s,\boldsymbol{a}})} P\widehat{V}^{\sigma_i}_{i,h+1} + \beta^{\pi^{\star}_i, \pi^{\mathsf{b}}_{-i}}_{i,h}(s)$$

$$\quad - \mathbb{E}_{\boldsymbol{a} \sim (\pi^{\star}_i(s), \pi^{\mathsf{b}}_{-i}(s))} \inf_{P \in \mathcal{U}^{\sigma_i}(P^0_{h,s,\boldsymbol{a}})} P V^{\pi^{\mathsf{d}}_i, \pi^{\star}_{-i}, \sigma^+}_{i,h+1}$$

$$\leq \mathbb{E}_{\boldsymbol{a} \sim (\pi^{\star}_i(s), \pi^{\mathsf{b}}_{-i}(s))} \left[ \inf_{P \in \mathcal{U}^{\sigma_i}(P^0_{h,s,\boldsymbol{a}})} P\widehat{V}^{\sigma_i}_{i,h+1} - \inf_{P \in \mathcal{U}^{\sigma_i}(P^0_{h,s,\boldsymbol{a}})} P V^{\pi^{\mathsf{d}}_i, \pi^{\star}_{-i}, \sigma^+}_{i,h+1} \right.$$

$$\left. + \left| \inf_{P \in \mathcal{U}^{\sigma_i}(P^0_{h,s,\boldsymbol{a}})} P\widehat{V}^{\sigma_i}_{i,h+1} - \inf_{P \in \mathcal{U}^{\sigma_i}(\widehat{P}^0_{h,s,\boldsymbol{a}})} P\widehat{V}^{\sigma_i}_{i,h+1} \right| \right] + \beta^{\pi^{\star}_i, \pi^{\mathsf{b}}_{-i}}_{i,h}(s)$$

$$\overset{(i)}{\leq} \mathbb{E}_{\boldsymbol{a} \sim (\pi^{\star}_i(s), \pi^{\mathsf{b}}_{-i}(s))} \left[ \inf_{P \in \mathcal{U}^{\sigma_i}(P^0_{h,s,\boldsymbol{a}})} P\widehat{V}^{\sigma_i}_{i,h+1} - \inf_{P \in \mathcal{U}^{\sigma_i}(P^0_{h,s,\boldsymbol{a}})} P V^{\pi^{\mathsf{d}}_i, \pi^{\star}_{-i}, \sigma_i}_{i,h+1} \right] + 2\beta^{\pi^{\star}_i, \pi^{\mathsf{b}}_{-i}}_{i,h}(s)$$

$$\overset{(ii)}{\leq} \mathbb{E}_{\boldsymbol{a} \sim (\pi^{\star}_i(s), \pi^{\mathsf{b}}_{-i}(s))} \left[ P^{\mathsf{inf}, V}_{i,h,s,\boldsymbol{a}} \left( \widehat{V}^{\sigma_i}_{i,h+1} - V^{\pi^{\mathsf{d}}_i, \pi^{\star}_{-i}, \sigma_i}_{i,h+1} \right) \right] + 2\beta^{\pi^{\star}_i, \pi^{\mathsf{b}}_{-i}}_{i,h}(s). \tag{220}$$

Here, (ii) is valid under the notation

$$P^{\mathsf{inf}, V}_{i,h,s,\boldsymbol{a}} := \arg\min_{P \in \mathcal{U}^{\sigma^+}(P^0_{h,s,\boldsymbol{a}})} P V^{\pi^{\mathsf{d}}_i, \pi^{\star}_{-i}, \sigma^+}_{i,h+1} \tag{221}$$

and consequently,

$$\inf_{P\in\mathcal{U}^{\sigma_i}\left(P_{h,s,\boldsymbol{a}}^0\right)} PV_{i,h+1}^{\pi_i^{\mathsf{d}},\pi_{-i}^{\star},\sigma^+} = P_{i,h,s,\boldsymbol{a}}^{\inf,V} V_{i,h+1}^{\pi_i^{\mathsf{d}},\pi_{-i}^{\star},\sigma^+}, \text{ and } \inf_{P\in\mathcal{U}^{\sigma_i}\left(P_{h,s,\boldsymbol{a}}^0\right)} P\widehat{V}_{i,h+1}^{\sigma_i} \le P_{i,h,s,\boldsymbol{a}}^{\inf,V} \widehat{V}_{i,h+1}^{\sigma_i}.$$

Besides, (i) in (220) exists due to (213) in Lemma 10 for $N_h(s,\boldsymbol{a}) > 0$ and

$$\left| \inf_{P\in\mathcal{U}^{\sigma_i}\left(P_{h,s,\boldsymbol{a}}^0\right)} P\widehat{V}_{i,h+1}^{\sigma_i} - \inf_{P\in\mathcal{U}^{\sigma_i}\left(\widehat{P}_{h,s,\boldsymbol{a}}^0\right)} P\widehat{V}_{i,h+1}^{\sigma_i} \right| \le H = \beta_{i,h}^{\pi_i^{\star},\pi_{-i}^{\mathsf{b}}}(s) \tag{222}$$

for $N_h(s,\boldsymbol{a}) = 0$.

For ease of proof, we introduce a notation as $\check{P}_{i,h,s}^{\inf,V} := \mathbb{E}_{\boldsymbol{a}\sim(\pi_i^{\star}(s),\pi_{-i}^{\mathsf{b}}(s))} P_{i,h,s,\boldsymbol{a}}^{\inf,V}$. Furthermore, we define a sequence of matrices $\check{P}_{i,h}^{\inf,V} \in \mathbb{R}^{S\times S}$. We can utilizing (220) recursively over the time steps $h, h+1, \cdots, H$ and derive

$$V_{i,h}^{\star,\widehat{\pi}_{-i},\sigma_i}(s) - V_{i,h}^{\star,\sigma_i}(s) \le \widehat{V}_{i,h}^{\sigma_i}(s) - V_{i,h}^{\pi_i^{\mathsf{d}},\pi_{-i}^{\star},\sigma_i}(s)$$

$$\le \check{P}_{i,h}^{\inf,V}\left(\widehat{V}_{i,h+1}^{\sigma_i} - V_{i,h+1}^{\pi_i^{\mathsf{d}},\pi_{-i}^{\star},\sigma_i}\right) + 2\beta_{i,h}^{\pi_i^{\star},\pi_{-i}^{\mathsf{b}}}(s)$$

$$\le \check{P}_{i,h}^{\inf,V} \check{P}_{i,h+1}^{\inf,V}\left(\widehat{V}_{i,h+2}^{\sigma_i} - V_{i,h+2}^{\pi_i^{\mathsf{d}},\pi_{-i}^{\star},\sigma^+}\right) + 2\check{P}_{i,h}^{\inf,V}\beta_{i,h+1}^{\pi_i^{\mathsf{d}},\pi_{-i}^{\star}} + 2\beta_{i,h}^{\pi_i^{\star},\pi_{-i}^{\mathsf{b}}}(s)$$

$$\le \cdots \le 2\sum_{i'=h}^{H}\left(\prod_{j=h}^{i'-1}\check{P}_{i,j}^{\inf,V}\right)\beta_{i,i'}^{\pi_i^{\star},\pi_{-i}^{\mathsf{b}}}, \tag{223}$$

where we define $\left(\prod_{j=h}^{i'-1}\check{P}_{i,j}^{\inf,V}\right) = I$ for convenience.

For any $d_h^{\pi_i^{\star},\pi_{-i}^{\mathsf{b}}} \in \mathcal{D}_h^{\mathsf{pi}}$ (cf. (41)), taking inner product with (46) yields

$$\left\langle d_h^{\pi_i^{\star},\pi_{-i}^{\mathsf{b}}}, V_{i,h}^{\star,\widehat{\pi}_{-i},\sigma_i}(s) - V_{i,h}^{\star,\sigma_i}(s) \right\rangle \le \left\langle d_h^{\pi_i^{\star},\pi_{-i}^{\mathsf{b}}}, 2\sum_{i'=h}^{H}\left(\prod_{j=h}^{i'-1}\check{P}_{i,j}^{\inf,V}\right)\beta_{i,i'}^{\pi_i^{\star},\pi_{-i}^{\mathsf{b}}} \right\rangle$$

$$= 2\sum_{i'=h}^{H}\left\langle d_{i'}^{\mathsf{p},\pi_i^{\star},\pi_{-i}^{\mathsf{b}}}, \beta_{i,i'}^{\pi_i^{\star},\pi_{-i}^{\mathsf{b}}} \right\rangle, \tag{224}$$

where

$$d_{i'}^{\mathsf{p},\pi_i^{\mathsf{d}},\pi_{-i}^{\star}} := \left[ (d_h^{\pi_i^{\star},\pi_{-i}^{\mathsf{b}}})^{\top}\left(\prod_{j=h}^{i'-1}\check{P}_{i,j}^{\inf,V}\right) \right]^{\top} \in \mathcal{D}_{i'}^{\mathsf{pi}} \tag{225}$$

by the definition of $\mathcal{D}_{i'}^{\mathsf{pi}}$ (cf. (218)) for all $i' = h+1, \cdots, H$.

Next, we control $\langle d_{i'}^{\mathsf{p},\pi_i^{\star},\pi_{-i}^{\mathsf{b}}}, \beta_{i,i'}^{\pi_i^{\star},\pi_{-i}^{\mathsf{b}}} \rangle$ utilizing concentrability. First of all, according to the definition of penalty, we demonstrate that the pessimistic penalty satisfies

$$\beta_{i,i'}(s,\boldsymbol{a},\hat{V}) \le \max\left\{ \sqrt{\frac{C_{\mathsf{n}}\log\frac{KH}{\delta}}{N_i(s,\boldsymbol{a})}\mathsf{Var}_{\widehat{P}_{i,s,\boldsymbol{a}}^0}(\widehat{V})}, \frac{2C_{\mathsf{n}}H\log\frac{KH}{\delta}}{N_i(s,\boldsymbol{a})} \right\}$$

$$\le \sqrt{\frac{C_{\mathsf{n}}\log\frac{KH}{\delta}}{N_i(s,\boldsymbol{a})}\mathsf{Var}_{\widehat{P}_{i,s,\boldsymbol{a}}^0}(\widehat{V})} + \frac{2C_{\mathsf{n}}H\log\frac{KH}{\delta}}{N_i(s,\boldsymbol{a})}$$

$$\overset{\text{(i)}}{\le} \sqrt{\frac{C_{\mathsf{n}}\log\frac{KH}{\delta}}{N_i(s,\boldsymbol{a})}\left(2\mathsf{Var}_{P_{i,s,\boldsymbol{a}}^0}(\widehat{V}) + \frac{C_0 H^2}{N_i(s,\boldsymbol{a})}\log\frac{KH}{\delta}\right)} + \frac{2C_{\mathsf{n}}H\log\frac{KH}{\delta}}{N_i(s,\boldsymbol{a})}$$

$$\overset{\text{(ii)}}{\le} \sqrt{\frac{2C_{\mathsf{n}}\log\frac{KH}{\delta}}{N_i(s,\boldsymbol{a})}\mathsf{Var}_{P_{i,s,\boldsymbol{a}}^0}(\widehat{V})} + \frac{(2C_{\mathsf{n}}+\sqrt{C_{\mathsf{n}}C_0})H\log\frac{KH}{\delta}}{N_i(s,\boldsymbol{a})} \tag{226}$$

where (i) holds by applying (212) for some sufficiently large $C_0$ and (ii) exists follows from the Cauchy-Schwarz inequality. Therefore, combining the definition of $\beta_{i,i'}^{\pi_i^\star, \pi_{-i}^b}(s)$, we obtain

$$\langle d_{i'}^{p, \pi_i^\star, \pi_{-i}^b}, \beta_{i,i'}^{\pi_i^\star, \pi_{-i}^b} \rangle = \sum_{s \in \mathcal{S}} d_{i'}^{p, \pi_i^\star, \pi_{-i}^b}(s) \beta_{i,i'}^{\pi_i^\star, \pi_{-i}^b}(s)$$

$$= \sum_{s \in \mathcal{S}} d_{i'}^{p, \pi_i^\star, \pi_{-i}^b}(s) \mathbb{E}_{\boldsymbol{a} \sim (\pi_i^\star(s), \pi_{-i}^b(s))} \beta_{i,i'}(s, \boldsymbol{a}, \hat{V})$$

$$= \sum_{(s, \boldsymbol{a}) \in \mathcal{S} \times \mathcal{A} \times \mathcal{B}} d_{i'}^{p, \pi_i^\star, \pi_{-i}^b}(s) \mathbb{1}\{a_i = \pi_i^\star(s)\} \pi_{-i}^d(\boldsymbol{a}_{-i}|s) \beta_{i,i'}(s, \boldsymbol{a}, \hat{V})$$

$$= \sum_{(s, a_i) \in \mathcal{S} \times \mathcal{A}} d_{i'}^{p, \pi_i^\star, \pi_{-i}^b}(s, a_i, \pi_{-i}^b(s)) \beta_{i,i'}(s, \pi_i^d(s), \boldsymbol{a}_{-i}, \hat{V}), \tag{227}$$

where the last equation holds due to the definition in (201b). Then, we observe $d_h^{p, \pi_i^\star, \pi_{-i}^b}(s, \boldsymbol{a}) \in \mathcal{D}_h^{\text{pai}}$ (cf. (219)). Thereafter, we divide the bound (227) into two cases.

**For the first case**, i.e., $s \in S$ where $\max_{P \in \mathcal{U}^{\sigma_i}(P^0)} d_{i'}^{\pi_i^\star, \pi_{-i}^b, P}(s, a_i, \pi_{-i}^b(s)) = 0$, it follows from the definition (cf. (218)) that for any $d_{i'}^{p, \pi_i^\star, \pi_{-i}^b}(s, a_i, \pi_{-i}^b(s)) \in \mathcal{D}_i^{\text{pai}}$, it satisfies that

$$d_{i'}^{p, \pi_i^\star, \pi_{-i}^b}(s, a_i, \pi_{-i}^b(s)) = 0. \tag{228}$$

**For the second case**, i.e., $s \in S$ where $\max_{P \in \mathcal{U}^{\sigma^+}(P^0)} d_{i'}^{\pi_i^\star, \pi_{-i}^b, P}(s, a_i, \pi_{-i}^b(s)) > 0$, by the assumption in (210)

$$\max_{P \in \mathcal{U}^{\sigma_i}(P^0)} \frac{\min \left\{ d_{i'}^{\pi_i^\star, \pi_{-i}^b, P}(s, a_i, \pi_{-i}^b(s)), \frac{1}{S \sum_{i=1}^{} A_i} \right\}}{d_{i'}^n(s, a_i, \pi_{-i}^b(s))} \leq C_r^\star < \infty.$$

It implies that

$$d_{i'}^n(s, a_i, \pi_{-i}^b(s)) > 0 \quad \text{and} \quad (i', s, a_i, \pi_{-i}^b(s)) \in \mathcal{C}^n. \tag{229}$$

Lemma 9 tells that with probability at least $1 - 8\delta$,

$$N_{i'}(s, a_i, \pi_{-i}^b(s)) \geq \frac{K d_{i'}^n(s, a_i, \pi_{-i}^b(s))}{8} - 5\sqrt{K d_{i'}^n(s, a_i, \pi_{-i}^b(s)) \log \frac{KH}{\delta}}$$

$$\overset{(i)}{\geq} \frac{K d_{i'}^n(s, a_i, \pi_{-i}^b(s))}{16}$$

$$\overset{(ii)}{\geq} \frac{K \max_{P \in \mathcal{U}^{\sigma_i}(P^0)} \min \left\{ d_{i'}^{\pi_i^\star, \pi_{-i}^b, P}(s, a_i, \pi_{-i}^b(s)), \frac{1}{S \sum_{i=1}^{} A_i} \right\}}{16 C_r^\star}$$

$$\geq \frac{K \min \left\{ d_{i'}^{p, \pi_i^\star, \pi_{-i}^b}(s, a_i, \pi_{-i}^b(s)), \frac{1}{S \sum_{i=1}^{} A_i} \right\}}{16 C_r^\star}, \tag{230}$$

where (ii) comes from Assumption 2 and (i) holds due to

$$K d_{i'}^n(s, a_i, \pi_{-i}^b(s)) \geq c_0 \frac{HS \sum_{i=1}^{} A_i}{d_m^n} \log \frac{KH}{\delta} f(\{\sigma_i\}_{i=1}^m, H) d_{i'}^n(s, a_i, \pi_{-i}^b(s))$$

$$\geq c_0 HS \sum_{i=1}^{} A_i \log \frac{KH}{\delta} f(\{\sigma_i\}_{i=1}^m, H) \geq 1600 \log \frac{KH}{\delta}, \tag{231}$$

where $f(\{\sigma_i\}_{i=1}^m, H) = \min \left\{ \left\{ \frac{(H\sigma_i - 1 + (1 - \sigma_i)^H)}{(\sigma_i)^2} \right\}_{i=1}^m, H \right\}$, the first inequality follows from condition (29), and the second inequality follows from

$$d_m^n = \min_{h, s, a_i, \pi_{-i}^b(s)} \left\{ d_h^n(s, \pi_i^d(s), \boldsymbol{a}_{-i}) : d_h^n(s, \pi_i^d(s), \boldsymbol{a}_{-i}) > 0 \right\} \leq d_{i'}^n(s, a_i, \pi_{-i}^b(s)). \tag{232}$$

Combining the results in (49) and (50), we arrive at

$$
\langle d_{i'}^{\mathsf{p},\pi_i^\star,\pi_{-i}^{\mathsf{b}}}, \beta_{i,i'}^{\pi_i^\star,\pi_{-i}^{\mathsf{b}}} \rangle
$$

$$
= \sum_{(s,a_i)\in\mathcal{S}\times\mathcal{A}_i} d_{i'}^{\mathsf{p},\pi_i^\star,\pi_{-i}^{\mathsf{b}}}(s,a_i,\pi_{-i}^{\mathsf{b}}(s))\beta_{i,i'}(s,a_i,\pi_{-i}^{\mathsf{b}}(s),\hat{V})
$$

$$
\leq \sum_{(s,a_i)\in\mathcal{S}\times\mathcal{A}_i} d_{i'}^{\mathsf{p},\pi_i^\star,\pi_{-i}^{\mathsf{b}}}(s,a_i,\pi_{-i}^{\mathsf{b}}(s))\sqrt{\frac{2C_{\mathsf{n}}\log\frac{KH}{\delta}}{N_i\left(s,a_i,\pi_{-i}^{\mathsf{b}}(s)\right)}\mathsf{Var}_{P^0_{i,s,a_i,\pi_{-i}^{\mathsf{b}}(s)}}\left(\widehat{V}\right)}
$$

$$
+ \sum_{(s,a_i)\in\mathcal{S}\times\mathcal{A}_i} d_{i'}^{\mathsf{p},\pi_i^\star,\pi_{-i}^{\mathsf{b}}}(s,a_i,\pi_{-i}^{\mathsf{b}}(s))\frac{\left(2C_{\mathsf{n}}+\sqrt{C_{\mathsf{n}}C_0}\right)H\log\frac{KH}{\delta}}{N_i\left(s,a_i,\pi_{-i}^{\mathsf{b}}(s)\right)}
$$

$$
\leq \sum_{(s,a_i)\in\mathcal{S}\times\mathcal{A}_i} d_{i'}^{\mathsf{p},\pi_i^\star,\pi_{-i}^{\mathsf{b}}}(s,a_i,\pi_{-i}^{\mathsf{b}}(s))\left(\frac{16C_{\mathsf{r}}^\star\left(2C_{\mathsf{n}}+\sqrt{C_{\mathsf{n}}C_0}\right)H\log\frac{KH}{\delta}}{K\min\left\{d_{i'}^{\mathsf{p},\pi_i^\star,\pi_{-i}^{\mathsf{b}}}(s,a_i,\pi_{-i}^{\mathsf{b}}(s)),\frac{1}{S\sum_{i=1}A_i}\right\}}\right.
$$

$$
\left. + \sqrt{\frac{32C_{\mathsf{r}}^\star C_{\mathsf{n}}\log\frac{KH}{\delta}}{K\min\left\{d_{i'}^{\mathsf{p},\pi_i^\star,\pi_{-i}^{\mathsf{b}}}(s,a_i,\pi_{-i}^{\mathsf{b}}(s)),\frac{1}{S\sum_{i=1}A_i}\right\}}\mathsf{Var}_{P^0_{i,s,a_i,\pi_{-i}^{\mathsf{b}}(s)}}\left(\widehat{V}\right)}\right). \tag{233}
$$

Similar to the proof in Appendix B.2, we are ready to bound $V_{i,1}^{\star,\sigma_i}(\varrho) - V_{i,1}^{\widehat{\pi}_i,\star,\sigma_i}(\varrho)$. There exists some sufficiently large constants $C_1, C_2, C_3 > 0$, and

$$
V_{i,1}^{\star,\widehat{\pi}_{-i},\sigma_i}(\varrho) - V_{i,1}^{\star,\sigma_i}(\varrho) \leq \sqrt{\frac{C_{\mathsf{r}}^\star C_1 H^3 S\sum_{i=1}A_i\log\frac{KH}{\delta}}{K}\min\left\{\frac{2(H\sigma_i-1+(1-\sigma_i)^H)}{(\sigma_i)^2},H\right\}}
$$

$$
+ \frac{C_{\mathsf{r}}^\star C_2 H^2 S\sum_{i=1}A_i\log\frac{KH}{\delta}}{K}\min\left\{\frac{2(H\sigma_i-1+(1-\sigma_i)^H)}{(\sigma_i)^2},H\right\}
$$

$$
\leq \sqrt{\frac{C_{\mathsf{r}}^\star C_3 H^3 S\sum_{i=1}A_i\log\frac{KH}{\delta}}{K}\min\left\{\frac{2(H\sigma_i-1+(1-\sigma_i)^H)}{(\sigma_i)^2},H\right\}}, \tag{234}
$$

where the last inequality follows from condition (29).

**Step 3: summing up the results** Consequently, we obtain the upper bound of $V_{i,1}^{\star,\widehat{\pi}_{-i},\sigma_i}(\varrho) - V_{i,1}^{\widehat{\pi},\sigma_i}(\varrho)$ in (234). which directly leads to

$$
\mathrm{Gap}(\widehat{\pi}) \leq c_1 \sqrt{\frac{C_{\mathsf{r}}^\star H^2 S\sum_{i=1}^m A_i\log\frac{KH}{\delta}}{K}\min\left\{\left\{\frac{2(H\sigma_i-1+(1-\sigma_i)^H)}{(\sigma_i)^2}\right\}_{i=1}^m,H\right\}}, \tag{235}
$$

for some sufficiently large $c_1$ and

$$
K \geq HS\sum_{i=1}A_i\log\frac{KH}{\delta}\min\left\{\left\{\frac{2(H\sigma_i-1+(1-\sigma_i)^H)}{(\sigma_i)^2}\right\}_{i=1}^m,H\right\}.
$$

