# OpenReview forum: "Sample Efficient Robust Offline Self-Play for Model-based Reinforcement Learning"
_ICLR.cc/2025/Conference — Submitted to ICLR 2025_

### Official Review · Reviewer_RJEh · 2024-10-20

**Soundness:** 3
**Presentation:** 3
**Contribution:** 2
**Rating:** 5
**Confidence:** 3

**Summary:**

This paper presents a robust model-based algorithm for offline two-player zero-sum Markov games (RTZMGs), effectively addressing the challenges of learning under partial coverage and environmental uncertainty. The key contributions of the paper are as follows:

- The authors introduce the robust tabular zero-sum Markov game framework by extending the standard tabular zero-sum Markov game to a robust setting. Under this framework, they propose a new algorithm, RTZ-VI-LCB, which integrates robust value iteration with a data-informed penalty term to estimate robust Nash equilibria.
- The authors provide a finite-sample complexity analysis for RTZ-VI-LCB, demonstrating its optimal dependency on the number of actions. This represents the first set of optimal sample complexity bounds for RTZMGs.
- The authors establish a lower bound on the sample complexity for learning RTZMGs, confirming the tightness of their upper bound and demonstrating the near-optimality of RTZ-VI-LCB across varying levels of uncertainty.

**Strengths:**

The primary strengths of this paper can be summarized in the following two aspects:

1. This paper introduces the first algorithm that achieves optimal sample complexity with respect to the dependence on action spaces.
2. The paper offers a comprehensive analysis of robust tabular zero-sum Markov games, presenting both upper and lower bounds on the sample complexity.

**Weaknesses:**

There are two major weaknesses from my perspective:

1. The authors do not discuss whether a Nash equilibrium exists under their definition of the robust zero-sum Markov game. It is well known that in robust Markov games, the existence of a Nash equilibrium can be affected by the choice of uncertainty sets and specific problem settings. Therefore, I believe it is essential to provide a discussion on the existence of Nash equilibrium within their framework.
2. Another weakness is the limited technical novelty of the work. The presentation in Sections 3.1 and 3.2 closely resembles that of [1], and the overall methodology appears to be a direct combination of [1] and [2]. The primary contribution seems to be the incorporation of the two-fold subsampling trick from [3] to sharpen the sample complexity bounds.

**Questions:**

Based on the discussion of the paper's strengths and weaknesses, I have the following questions for the authors:

1. The authors focus on the finite-horizon setting. Can the methodology presented in the paper be extended to analyze the infinite-horizon setting, as in [1]? Additionally, why did the authors choose to focus on the finite-horizon case rather than the infinite-horizon scenario?

2. The algorithmic framework follows from [1]. What are the specific technical challenges in extending the techniques of [1] from standard zero-sum Markov games to robust zero-sum Markov games?


 [1] Yan Y, Li G, Chen Y, et al. Model-based reinforcement learning is minimax-optimal for offline zero-sum markov games[J]. arXiv preprint arXiv:2206.04044, 2022.

   [2]Shi, Laixi, and Yuejie Chi. "Distributionally robust model-based offline reinforcement learning with near-optimal sample complexity." Journal of Machine Learning Research 25.200 (2024): 1-91.

   [3]Li G, Shi L, Chen Y, et al. Settling the sample complexity of model-based offline reinforcement learning[J]. The Annals of Statistics, 2024, 52(1): 233-260.

---

> ### Author Response · Authors · 2024-11-23
> **Part I**
>
> **Weakness 1**: Thank you for your valuable feedback. The existence of a Nash equilibrium policy has been proved in Ref. [1]. As suggested, we have highlighted this conclusion in line 250 in the revised version.
>
> **Weakness 2**: We would like to express our sincere gratitude for your valuable comments. While Sections 3.1 and 3.2 may resemble parts of [1], this similarity arises due to the fact that our work builds on their theoretical foundation but aims to address a fundamentally different problem: how to enhance sample efficiency with robustness for two-player zero-sum Markov games under partial coverage. We shall clarify the distinct contributions and innovations introduced in our paper.
>
> *Contributions:* We design the RTZ-VI-LCB algorithm and prove that it offers **the best sample complexity** for offline robust two-player zero-sum Markov games. Our algorithm achieves the theoretical lower bound in terms of state space and action space, achieving **optimal dependency** on these factors considering uncertainty level. Our result is significantly better than the result obtained by Blanchet et. al., particularly in state space and action space. Notably, the result obtained by Blanchet et. al. does not account for the influence of uncertainty level. Moreover, our algorithm extends beyond the scope of the work [1] and [2], by considering the joint robustness of two-player zero-sum Markov games and addressing partial dataset coverage in a principled manner. The techniques developed in this paper offer practical value, demonstrating how robust RL methods can operate under realistic constraints, such as partial observability and limited data coverage.
>
> *Innovations:* Unlike [1] or [2], we specifically focus on the challenges associated with adversarial uncertainty in both players’ policies, which have never been addressed in prior works. While adopting the two-fold sub-sampling trick originating from [3], we integrate it in a novel context to derive tighter sample complexity bounds for robust offline algorithms. This integration is non-trivial and requires careful adaptation to the setting of robust Markov games. Furthermore, our analysis provides new insights into the interaction between partial coverage and robust policy optimization, which are missing in prior works.
>
> Collectively, these contributions and innovations advance the state of the art in robust reinforcement learning for Markov games. We have revised the manuscript to better highlight the technical novelty and the distinctions from related works.
>
> **Q1**: Thank you for this insightful feedback. Our current work focuses on the finite-horizon setting primarily because it aligns with many real-world applications where decision-making processes are naturally bounded within a fixed time horizon, e.g., episodic tasks in reinforcement learning (See Ref. [1]), time-constrained planning (See Ref. [2]), and certain competitive environments (See Ref. [3]).
>
> Regarding the extension to an infinite-horizon setting, while our methodology is specifically tailored to the finite-horizon case, our algorithm could be adapted to the infinite-horizon setting with appropriate modifications, which will be part of our future work.
>
> **Q2**: Sincere thanks to the reviewer for this astute comment. Compared to [1], in our work, the technical challenges introduced by environment uncertainty stem from the need to incorporate and manage uncertainty in both the environment and the data, as well as the additional complexities introduced by the requirement of robustness in the value function and sample complexity analysis.
>
> On the one hand, in robust two-player zero-sum Markov games, players must account for uncertainty in the environment, often modeled through adversarial distributions. This requires the adaptation of the two-player zero-sum Markov games to handle the worst-case scenarios, leading to a new robust value function. However, we cannot learn the robust Q-function directly, since it could be computationally intensive with requirement of optimizing over an $S$-dimensional probability simplex.
>
> On the other hand, the robustness aspect of the problem requires a more sophisticated analysis of sample complexity, as players must learn policies that are resilient to uncertainty in the environment. Unlike standard zero-sum Markov games, with sample complexity bounds typically depending on the number of states, actions, and the horizon, our robustness setting introduces additional dependence on the uncertainty set.
>
> For these reasons, the robust zero-sum Markov games poses significant new challenges of incorporating and managing uncertainty, and the additional complexities introduced by the requirement of robustness, compared to the standard zero-sum Markov games like the one studied in [1].

---

> ### Author Response · Authors · 2024-11-23
> **Reference**
>
> **Reference**
>
> [1]. Blanchet, Jose, et al. "Double pessimism is provably efficient for distributionally robust offline reinforcement learning: Generic algorithm and robust partial coverage." Advances in Neural Information Processing Systems 36 (2024).
>
> [2]. Li, Jialian, et al. "Exploration analysis in finite-horizon turn-based stochastic games." Conference on Uncertainty in Artificial Intelligence. PMLR, 2020.
>
> [3]. Clempner, Julio B. "A Bayesian reinforcement learning approach in markov games for computing near-optimal policies." Annals of Mathematics and Artificial Intelligence 91.5 (2023): 675-690.
>
> [4]. Guo, Wenbo, et al. "Adversarial policy learning in two-player competitive games." International conference on machine learning. PMLR, 2021.

---

> ### Comment · Reviewer_RJEh · 2024-11-25
> **Thank You**
>
> Thank you for the response. I have no additional questions.

---

> > ### Author Response · Authors · 2024-11-25
> > **Thank you**
> >
> > Sincere thanks for your acknowledgment and kind approval of our response and revision. If there are no further questions or concerns, we kindly hope you might consider raising the score of our submission. If any additional issues arise, we would be more than happy to provide further clarification.

---

### Official Review · Reviewer_DR6T · 2024-10-30

**Soundness:** 3
**Presentation:** 2
**Contribution:** 3
**Rating:** 6
**Confidence:** 3

**Summary:**

This paper studies robust two-player zero-sum Markov games.  Recent papers have provided near optimal sample complexity bounds in this setting under partial and limited coverage of historical data individually, but cannot handle both settings simultaneously.  This work provides an algorithm that can achieve near-optimal sample complexity under partial and limited coverage simultaneously while also providing information theoretic lower bounds.

**Strengths:**

The question of robust learning in strategic multi-agent settings has received considerable attention in recent years. This paper builds on recent work, providing a clear contribution by combining technical and algorithmic ideas from recent work to provide tighter and more general results than the state-of-the-art.

Additionally, the work provides interesting lower bounds, whose proofs provide important new insight for the area.

**Weaknesses:**

The algorithmic novelty in the work is not clear.  A core component of the algorithm is a natural extension of work by Li et al the setting in this paper.  From my read, the key algorithmic novelty is primarily in the penalty term.

The technical novelty of the paper is not clearly presented.  It seems that key components of the proof follow recent work (e.g. much of the work in step 1 of the proof follows closely the approach of Shi et al.).  That said, there are certainly new ideas in the proof, it is just that the paper does not do a good job of highlighting the new techniques in the analysis.

Numerical comparisons to the work of Blanchet et al and Shi et al are not included.  Such comparisons would increase the potential impact of the paper and highlight the extent to which the improvement in theoretical bounds represent empirical improvements.

**Questions:**

Can you please clarify the new technical ideas in the proof as compared to the work of Blanchet et al and She et al?

Can you please clarify the relationship of the algorithmic ideas to prior work, highlighting which components are natural extensions and which represent new algorithmic ideas?

---

> ### Author Response · Authors · 2024-11-23
>
> **Weakness 3**: Sincere thanks to you for this valuable suggestion. We would highlight our algorithm is the first of theoretical endeavor that establish the best sample complexity in state $S$ and action spaces $\\{A, B\\}$ considering the uncertainty levels. We hope that our algorithm can motivate more applications and studies in the future. On the other hand, like the existing theoretical study of RTZMGs in Ref. [1], we do not include numerical experiments. Nevertheless, numerically simulating our algorithm on practical applications is currently non-trivial. We are conducting experiments using a toy example.
>
> **Novelty (Weakness 1 \& 2, Question 1 \& 2)**:
> We would like to express our sincere gratitude for your feedback regarding the novelty of our work. As suggested, the distinct contributions and innovations of our paper are clarified, as follows.
>
> *Contributions:* We design the RTZ-VI-LCB algorithm and prove that it offers **the best sample complexity** for offline robust two-player zero-sum Markov games. Our algorithm achieves the theoretical lower bound in terms of state space and action space, achieving **optimal dependency** on these factors considering uncertainty level. Our result is significantly better than the result obtained by Blanchet et. al., particularly in state space and action space. Notably, the result obtained by Blanchet et. al. does not account for the influence of uncertainty level. Moreover, our algorithm extends beyond the scope of the work by Shi et. al., by considering the joint robustness of both players and addressing partial dataset coverage in a principled manner. The techniques developed in this paper offer practical value, demonstrating how robust RL methods can operate under realistic constraints, such as partial observability and limited data coverage.
>
> *Innovations:* Unlike the work by Shi et. al., we specifically focus on the challenges associated with adversarial uncertainty in both players’ policies, which have never been addressed in prior works. While adopting the two-fold sub-sampling trick originating from Li et. al., we integrate it in a novel context to derive tighter sample complexity bounds for robust offline algorithms. This integration is non-trivial and requires careful adaptation to the setting of robust Markov games. Furthermore, our analysis provides new insights into the interaction between partial coverage and robust policy optimization, which are missing in prior works.
>
> Collectively, these contributions and innovations advance the state of the art in robust reinforcement learning for Markov games. We have revised the manuscript to better highlight the technical novelty and distinctions from related works.
>
> **Reference**
>
> [1]. Blanchet, Jose, et al. "Double pessimism is provably efficient for distributionally robust offline reinforcement learning: Generic algorithm and robust partial coverage." Advances in Neural Information Processing Systems 36 (2024).

---

> > ### Comment · Reviewer_DR6T · 2024-11-25
> >
> > Thank you for your response and clarifications.  I maintain my positive score.

---

> > > ### Author Response · Authors · 2024-11-26
> > > **Thank you**
> > >
> > > Thank you for acknowledging our response. If you have any further concerns or questions, we would be more than happy to provide additional clarification.

---

### Official Review · Reviewer_GTfX · 2024-11-02

**Soundness:** 3
**Presentation:** 3
**Contribution:** 3
**Rating:** 6
**Confidence:** 2

**Summary:**

The paper proposed an algorithm RTZ-VI-LCB, designed to efficiently find the robust Nash Equilibrium(NE) in Robust Two-player Zero-sum Markov Games (RTZMGs). The authors employ confidence bounds innovatively in the algorithm, enabling it to achieve a sample complexity close to the lower bound except for the order of the horizon.

**Strengths:**

The paper is well-written and easy to follow. The problem of efficiently finding the robust NE in RTZMGs is significant for the field. The theoretical results are strong, as the sample complexity of the RTZ-VI-LCB algorithm nearly matches the lower bound for this problem class. Additionally, the lower bound analysis indicates that RTZMGs are not notably easier than traditional TZMGs.

**Weaknesses:**

There are still some technical problems to justify, which will be discussed in the Questions section.

**Questions:**

1. In remark 1, the paper mentions that the coefficient $C_r^\star$ could be $\frac{AB}{A + B}$. Given that the sample complexity result is $\tilde{O}\left(\frac{C_r^*(A + B)}{\epsilon^2}\right)$, does this imply that the complexity is reduced to $\tilde{O}\left(\frac{A B}{\epsilon^2}\right)$ in terms of $A$ and $B$, which is the same as the result in DR-NVI?
2. How should we compare the term $\min\left(f(\sigma^+,\sigma^-),H\right)$ in the upper bound and the term $\min\left(1/\min(\sigma^+,\sigma^-),H\right)$ in the lower bound? Additional discussion on this comparison would clarify the practical implications of the upper bound's tightness relative to the lower bound.
3. From the similarity between the lower bounds of RTZMGs and RZMGs (Shi et al., 2024b), I assume that RTZMGs are not significantly easier than RZMGs. Given this, is it feasible to extend the RTZ-VI-LCB algorithm to more than two players?

---

> ### Author Response · Authors · 2024-11-23
> **Part I**
>
> **Weakness**: Thank you for your comments. We have carefully reviewed the specific issues mentioned in the Questions section and have provided detailed responses to each point below.
>
> **Q1**: We would like to apologize for our inaccurate description of DR-NVI. Our work focuses on the offline RTZMG setting. By contrast, DR-NVI considers online setting which is beyond the scope of our paper.
> In the revised version, we have strengthened the misleading description. Specifically, $C_{\mathrm{r}}^{\star}$ measures the distributional discrepancy between the historical dataset and the target data. This is a distinct factor in the offline RTZMG setting, setting it apart from the online setting involving $\\{A, B\\}$. Compared state-of-the-art algorithm $\mathrm{P}^2\mathrm{M}^2\mathrm{PO}$, our algorithm is not only optimal in the dependency of the state space and action spaces, but also has a lower concentrability coefficient by unprecedentedly capturing uncertainty levels. Please see our changes to Section 1 in the revised version.
>
> **Q2**: Thank you for your feedback and suggestions.
>
> * For the term $ T_1 = \min\\{f(\sigma^+, \sigma^-), H\\} $: Being the uncertainty levels of the two players, $\sigma^+$ and $\sigma^-$ are independent and can be analyzed separately using a similar approach. Taking $\sigma^+$ as an example, we define $ g(\sigma^+, H) = H\sigma^+ - H(1-\sigma^+)^H - (\sigma^+)^2H $. For $ H \geq 2 $, the first derivative of $g(\sigma^+, H)$ with respect to $\sigma^+$ is $ \frac{\partial g(\sigma^+, H)}{\partial \sigma^+} = H + H^2(1-\sigma^+)^{H-1} - 2H\sigma^+ $. The second derivative is $ \frac{\partial^2 g(\sigma^+, H)}{\partial (\sigma^+)^2} = -H^2(H-1)(1-\sigma^+)^{H-2} - 2H < 0 $, indicating that $ g(\sigma^+, H) $ is concave. By evaluating the first derivative at the boundaries, we find $ \frac{\partial g(\sigma^+, H)}{\partial \sigma^+} |_ {\sigma^+ \to 0} \to H^2 + H > 0 $ and $ \frac{\partial g(\sigma^+, H)}{\partial \sigma^+} |_ {\sigma^+ = 1} = -H < 0 $, which indicates that $ g(\sigma^+, H) $ first increases monotonically, reaches its maximum at some point $\sigma^\star$, and then decreases monotonically. Since $ g(\sigma^+ \to 0, H) \to -H < 0 $ and $ g(\sigma^+ = 1, H) = 0 $, there exists $ 0 < \sigma^0 < 1 $ such that $ g(\sigma^0, H) = 0 $.
> Thus, when $ \sigma^0 \le \min\\{\sigma^+, \sigma^-\\} \le 1 $, we have $ T_1 = H $. Otherwise, $ T_1 = \min \\{\frac{(H\sigma^+ - 1 + (1-\sigma^+)^H)}{(\sigma^+)^2}, \frac{(H\sigma^- - 1 + (1-\sigma^-)^H)}{(\sigma^-)^2}\\} $.
>
> * For the term $ T_2 = \min\\{1 / \min(\sigma^+, \sigma^-), H\\} $: When $ \min\\{\sigma^+, \sigma^-\\} \gtrsim 1/H $, we have $ T_2 = 1 / \min\\{\sigma^+, \sigma^-\\} $. Otherwise, $ T_2 = H $.
>
> In summary, the behavior of $ T_1 $ and $ T_2 $ depends on the values of $\sigma^+$, $\sigma^-$, and $ H $. We have added the above discussion into Appendix B.3 in the revised version.

---

> ### Author Response · Authors · 2024-11-23
> **Part II**
>
> **Q3**: Sincere thanks to you for your inspirational comment and question. We would like to clarify that robust two-player zero-sum Markov games (RTZMGs) present significant challenges compared to robust Markov decision processes (RMDPs) due to the interplay between two players with opposing objectives and the added complexity of robustness requirements.
>
> In RTZMGs, the optimal policies of both players are interdependent, necessitating the solution of a saddle-point problem, where one player’s actions directly influence the other’s reward. Additionally, adversarial uncertainty increases computational difficulty, as the robust value function must account for the dynamics of both the players’ policies and environmental uncertainty. Each player in RTZMGs must hedge against the worst-case outcomes of both the environment's uncertainty and the opposing player’s actions, creating a two-layer robustness problem. Furthermore, accurately estimating the robust value function demands careful calibration of uncertainty effects on both players' policies, which complicates exploration of the joint state and action spaces. The theoretical analysis of robust Nash equilibria in RTZMGs is also more intricate than in single-agent RMDPs. While single-agent settings focus on optimizing a single policy, RTZMGs require proving the existence and robustness of Nash equilibria under uncertainty. Extending theoretical frameworks from RMDPs to RTZMGs demands novel tools to analyze stability and convergence in adversarial environments. Therefore, RTZMGs are inherently more challenging due to their multi-agent structure, the interaction of adversarial uncertainties, and the need for advanced theoretical and computational methodologies.
>
> We would also clarify that our current algorithm can be extended to robust multi-agent general-sum Markov games, referred to as Multi-RTZ-VI-LCB. We have added Theorem 3 and its detailed information and proof in Appendix F for Multi-RTZ-VI-LCB to the revised version. Specifically, Theorem 3 asserts that the proposed Multi-RTZ-VI-LCB algorithm can attain an $\varepsilon$-robust NE solution when the total sample size exceeds $\widetilde{O}(\frac{C^\star_\mathrm{r} H^4 S\sum_{i=1}^m A_i}{\varepsilon^2} {\min \\{\\{\frac{(H\sigma_i-1+(1-\sigma_i)^H)}{(\sigma_i)^2}\\}_{i=1}^m, H\\}})$, breaking the curse of multiagency.

---

### Official Review · Reviewer_uNA9 · 2024-11-03

**Soundness:** 2
**Presentation:** 2
**Contribution:** 2
**Rating:** 5
**Confidence:** 4

**Summary:**

This work focuses on developing provable algorithm for distributionally robust multi-agent reinforcement learning in the face of environmental shift, in offline setting using only a history dataset. Considering two-player zero-sum games, it proposes RTZ-VI-LCB with an upper bound and a lower bound for this problem.

**Strengths:**

1. This is the first work that targets offline settings for robust MARL problems, which is an interesting topic.
2. It provides both upper and lower bounds for understanding this problem.

**Weaknesses:**

1. The writing and presentation need to be revised a lot. A lot of parts of the paper are similar to prior art. For instance, the two-fold sampling method in Algorithm 1 is almost the same as Algorithm 3 in [1]. Although cited the prior works, the algorithm needs to be rewritten entirely.
2. The contributions are a little bit overclaimed from the reviewer's viewpoint. In line 104, this work claims that "To the best of our knowledge, this is the first time optimal dependency on actions {A, B} has been achieved". While the concentrability coefficient also involves potential terms of A and B. So it is better to also say this is only for offline settings.
3. Some writing issues such as in line 107. The "transition kernel" does not need to be solved, it seems to need to be revised to "RTZMG". In line 113, "across a range of uncertainty levels", it seems there is something missing in this half sentence.
4. In the discussion part after showing the theorems, the reviewer highly suggests that the author check the claims again. For instance, in line 511-512, it seems the upper bound and lower bound do not match in $H$ even if $\min\\{\sigma^+, \sigma^- \\} \geq \frac{1}{H}$. The upper bound has $O(H^5)$, while the lower bound has $O(H^4)$? So it is not optimal yet, which is also claimed in the second paragraph of the discussion.

[1] Li, Gen, et al. "Settling the sample complexity of model-based offline reinforcement learning." The Annals of Statistics 52.1 (2024): 233-260.

**Questions:**

1. Why target two-player zero-sum games? Is there any special structure that helps the results, which hinders the authors from considering more general general-sum multi-agent games?

Other minors:
1) For presentation, as actually the max-player and min-player enjoys very similar formulation, algorithm update rules, and others, the presentation is a little bit redundant. It will be better to only write one time of them, such as equations 8(a), 8(b) can be represented as one if we let the min-player's everything be its negative version. The same for equation 9, 10, 18, the two terms in 22, and etc.

---

> ### Author Response · Authors · 2024-11-23
>
> **Weakness 1**: Thank you for your feedback and suggestions. We have rewritten Algorithm 1 in the revised version.
>
> **Weakness 2**: Thank you for your valuable feedback and suggestions. For the revised version, we have modified the claim pointed out by the reviewer to *"To the best of our knowledge, this is the first time optimal dependency on state $S$ and actions $\{A, B\}$ has been achieved for **offline RTZMGs**"*. Please see our changes to Line 97 - 98 in the revised version.
>
> **Weakness 3**: We apologize for this typo. We have corrected "transition kernel" into "*an RTZMG algorithm*" and rewritten sentence in line 106 to "*Besides, we confirm the optimality of RTZ-VI-LCB across different uncertainty levels of the  critical parameters, i.e., state $S$ and actions $\\{A, B\\}$, except for the finite-horizon $H$*".
>
> **Weakness 4**: We apologize for this typo. Our algorithm indeed matches the lower bound in the key factors, including the state $S$ and action $\{A, B\}$, except for $H$. We have thoroughly checked the related parts, i.e., abstract, contribution in Section 1, statement of Theorems in Section 4, and conclusion in Section 5, and corrected the typos in the revised version.
>
> **Q1**: Thank you for your helpful question. The reason for us focusing on our research on two-player zero-sum Markov games (TZMGs) is that TZMGs are more closely aligned with real-world problems. In many practical scenarios, such as adversarial security and Atari games (see Refs. [1-3] for details), the interactions between the two parties inherently exhibit zero-sum characteristics.
>
> Our current algorithm can be extended to robust multi-agent general-sum Markov games, referred to as Multi-RTZ-VI-LCB. We have added Theorem 3 and its detailed information and proof in Appendix F for Multi-RTZ-VI-LCB to the revised version. Specifically, Theorem 3 asserts that the proposed Multi-RTZ-VI-LCB algorithm can attain an $\varepsilon$-robust NE solution when the total sample size exceeds $\widetilde{O}(\frac{C^\star_\mathrm{r} H^4 S\sum_{i=1}^m A_i}{\varepsilon^2} {\min \\{\\{\frac{(H\sigma_i-1+(1-\sigma_i)^H)}{(\sigma_i)^2}\\}_{i=1}^m, H\\}})$, breaking the curse of multiagency.
>
> **Minor** Thank you for your valuable feedback and suggestions. As suggested, we have suppressed the redundant equations in the revised version.
>
> **Reference**
>
> [1]. EO, OO Ibidunmoye, B. K. Alese, and O. S. Ogundele. "Modeling attacker-defender interaction as a zero-sum stochastic game." Journal of Computer Sciences and Applications 1.2 (2013): 27-32.
>
> [2]. Guo, Wenbo, et al. "Adversarial policy learning in two-player competitive games." International conference on machine learning. PMLR, 2021.
>
> [3]. Sieusahai, Alexander, and Matthew Guzdial. "Explaining deep reinforcement learning agents in the atari domain through a surrogate model." Proceedings of the AAAI Conference on Artificial Intelligence and Interactive Digital Entertainment. Vol. 17. No. 1. 2021.

---

> > ### Comment · Reviewer_uNA9 · 2024-12-03
> > **Response to the authors**
> >
> > Thank the authors for the detailed response. I apologize for the delayed reply. Most of my concerns have been addressed. While I will maintain my score, I am open to acceptance if other reviewers champion it.

---

> > > ### Author Response · Authors · 2024-12-03
> > >
> > > Thank you for your openness to the acceptance of this paper. Could you please consider increasing the score from 5 (borderline rejection) to 6 (borderline acceptance). In other words, increasing your score to above the average score of the current paper. Sincere thanks for your generous support and invaluable feedback.

---

### Official Review · Reviewer_dphg · 2024-11-05

**Soundness:** 3
**Presentation:** 3
**Contribution:** 2
**Rating:** 6
**Confidence:** 3

**Summary:**

This paper addresses robust multi-agent reinforcement learning (MARL) in two-player zero-sum Markov games (TZMGs) by introducing the RTZ-VI-LCB algorithm, a sample-efficient approach to handle offline settings with environmental uncertainties. The algorithm improves robustness by applying value iteration with data-driven penalties and establishing sample complexity bounds without requiring full state-action space coverage.

**Strengths:**

1. The paper provides a rigorous theoretical framework, including upper and lower sample complexity bounds, which supports the robustness and efficiency claims of the RTZ-VI-LCB algorithm.

2. The design of RTZ-VI-LCB is explained in a step-by-step manner, making it easy to follow the rationale behind each component, such as the use of lower confidence bounds and two-player-wise rectangularity.

3. The paper adapts the robust Bellman equations specifically for two-player games, enhancing the clarity and relevance of the methodology in the context of MARL.

**Weaknesses:**

1. The paper assumes that both players in the two-player zero-sum game have identical uncertainty sets (same divergence function for both players). This simplifies the model but may limit its applicability to real-world scenarios where players could have different levels of uncertainty.

2. The penalty term introduced in the RTZ-VI-LCB algorithm is crucial for the robust value estimation, but the paper does not clearly explain how the penalty is calibrated or how different choices of penalty function influence the algorithm’s performance.

3. The paper assumes that historical datasets can be treated as independent samples after applying subsampling techniques, but it does not fully address the potential temporal dependencies within offline data.

**Questions:**

1. How does the algorithm's performance vary with different types of divergence functions beyond total variation, such as Kullback-Leibler divergence?

2. Would the RTZ-VI-LCB framework be adaptable to handle more complex multi-agent settings with more than two players?

3. How sensitive is the model’s performance to variations in the clipping parameter $C_r^*$, and what guidelines can be provided for choosing this parameter effectively?

---

> ### Author Response · Authors · 2024-11-23
> **Part I**
>
> **Weakness 1**: Thank you for your insightful feedback. On the one hand, the assumption of identical uncertainty levels serves as a reasonable approximation in some applications. For instance, in Atari games, both players operate under similar environmental uncertainties. On the other hand, the assumption of identical uncertainty sets significantly enhances analytical tractability. This assumption may not hold universally. We will further investigate more general cases in our future work.
>
> **Weakness 2**: Thank you for your feedback and suggestions. Our penalty term enforces optimistic estimation amid uncertainty. As anticipated, the properties of the fixed point of equalities in line 4 of Algorithm 2 rely heavily upon the choice of the penalty, often derived based on certain concentration bounds. In our work, we consider the Bernstein-style penalty to prioritize certain variance statistics.
>
> To clarify the penalty defined in RTZ-VI-LCB, the following has been added to the revised version
> "*We adopt the Bernstein-style penalty to better capture the variance structure over time*" in lines 363-364 and "*Note that we choose $\widehat{P}^0$, as opposed to ${P}^0$ (i.e., $\mathsf{Var}_ {\widehat{P}^0_ {h,s,a,b}}(\widehat{V})$) in the variance term, since we have no access to the true transition kernel ${P}^0$*" in lines 373-374.
>
> **Weakness 3**: Thank you for this astute comment. We would like to clarify that the samples in dataset $\mathcal{D}_0$ produced by two-stage subsampling technique are independent. This independence has already been proved for single-agent cases in Ref. [1]. This is not an assumption.
>
> In this paper, we extend the proof in Ref. [1] to robust two-player zero-sum Markov games. To clarify this, we examine two distinct data-generation mechanisms, where a sample transition quadruple $(s, a, b, h, s')$ represents a transition from state $s$ with actions $\{a, b\}$ to state $s'$ at step $h$.
>
> * Step 1: Augmenting $\mathcal{D}^{\mathrm{t}}$ to create $\mathcal{D}^{\mathrm{t,a}}$.
> To construct the augmented dataset $\mathcal{D}^{\mathrm{t,a}}$, for each $(s, h) \in \mathcal{S} \times [H]$, (i) we define $\mathcal{D}^{\mathrm{t,a}}$ to collect all $\min\\{N^{\mathrm{t}}_ h(s), N^{\mathrm{m}}_ h(s)\\}$ sample transitions in $\mathcal{D}^{\mathrm{t}}$ originating from state $s$ at step $h$; and (ii) if $N^{\mathrm{t}}_ h(s) > N^{\mathrm{m}}_ h(s)$, we supplement $\mathcal{D}^{\mathrm{t,a}}$ with an additional $N^{\mathrm{t}}_ h(s) - N^{\mathrm{m}}_ h(s)$ independent sample transitions $\\{ (s, a^{(i)}_ {h,s}, b^{(i)}_ {h,s}, h, s^{\prime\,(i)}_ {h,s} ) \\}$ with
> $$a^{(i)}_ {h,s} \overset{\mathrm{i.i.d.}}{\sim} \mu^\mathrm{b}_ h(\cdot | s), \quad
> b^{(i)}_ {h,s} \overset{\mathrm{i.i.d.}}{\sim} \nu^\mathrm{b}_ h(\cdot | s), \quad
> s^{\prime\,(i)}_ {h,s} \overset{\mathrm{i.i.d.}}{\sim} P_ h\big(\cdot | s, a^{(i)}_ {h,s}, b^{(i)}_ {h,s} \big), \quad
> N^{\mathrm{m}}_ h(s) < i \leq N^{\mathrm{t}}_ h(s).
> $$
>
> * Step 2: Constructing $\mathcal{D}^{\mathrm{iid}}$.
> For each $(s, h) \in \mathcal{S} \times [H]$, we generate $N^{\mathrm{t}}_ h(s)$ independent sample transitions $\\{ \big(s, a^{(i)}_ {h,s}, b^{(i)}_ {h,s}, h, s^{\prime\,(i)}_ {h,s} \big) \\}$ with
> $$
> a^{(i)}_ {h,s} \overset{\mathrm{i.i.d.}}{\sim} \mu^{\mathrm{b}}_ h(\cdot | s), \quad
> b^{(i)}_ {h,s} \overset{\mathrm{i.i.d.}}{\sim} \nu^{\mathrm{b}}_ h(\cdot | s), \quad
> s^{\prime\,(i)}_ {h,s} \overset{\mathrm{i.i.d.}}{\sim} P_ h\big(\cdot | s, a, b \big), \quad
> 1 \leq i \leq N^{\mathrm{t}}_ h(s).
> $$
> The resulting dataset is given by
> $$
> \mathcal{D}^{\mathrm{iid}} \coloneqq \\{ \big( s, a^{(i)}_ {h,s}, b^{(i)}_ {h,s}, h, s^{\prime\,(i)}_ {h,s} \big) \mid s \in \mathcal{S}, 1 \leq h \leq H, 1 \leq i \leq N^{\mathrm{t}}_ h(s) \\}.
> $$
>
> * Establishing independence property.
> The dataset $\mathcal{D}^{\mathrm{t,a}}$ deviates from $\mathcal{D}^{\mathrm{t}}$ only when $N^{\mathrm{t}}_ h(s) > N^{\mathrm{m}}_ h(s)$. This augmentation ensures that $\mathcal{D}^{\mathrm{t,a}}$ contains precisely $N^{\mathrm{t}}_ h(s)$ sample transitions from state $s$ at step $h$. Both $\mathcal{D}^{\mathrm{t,a}}$ and $\mathcal{D}^{\mathrm{iid}}$ comprise exactly $N^{\mathrm{t}}_ h(s)$ sample transitions from state $s$ at step $h$, with $\\{N^{\mathrm{t}}_ h(s)\\}$ being statistically independent of the random sample generation.
> Consequently, given $\{N^{\mathrm{t}}_ h(s)\}$, the sample transitions in $\mathcal{D}^{\mathrm{t,a}}$ across different steps are statistically independent. Both $\mathcal{D}^{\mathrm{t}}$ and $\mathcal{D}^{\mathrm{iid}}$ can be regarded as collections of independent samples.
>
> We have added the analysis above into Appendix C.1 in the revised version.

---

> ### Author Response · Authors · 2024-11-23
> **Part II**
>
> **Q1**: Thank you for insightful feedback. While our work focuses on total variation distance, the RTZ-VI-LCB algorithm is inherently flexible and can be adapted to alternative divergence measures, including KL divergence, by appropriately redefining the uncertainty sets. In the context of RTZMG research, Ref. [2] has explored TV distance and KL divergence, demonstrating that the sample complexity exhibits identical dependencies on the horizon, state space, and action spaces under these two metrics. This indicates that the choice of the divergence measure has a relatively minor impact on sample complexity. Investigating whether our algorithm achieves similar performance with KL divergence is an important direction of our future research.
>
> **Q2**: Our current algorithm can be extended to robust multi-agent general-sum Markov games, referred to as Multi-RTZ-VI-LCB. We have added Theorem 3 and its detailed information and proof in Appendix F for Multi-RTZ-VI-LCB to the revised version. Specifically, Theorem 3 asserts that the proposed Multi-RTZ-VI-LCB algorithm can attain an $\varepsilon$-robust NE solution when the total sample size exceeds $\widetilde{O}(\frac{C^\star_ \mathrm{r} H^4 S\sum_ {i=1}^m A_i}{\varepsilon^2} {\min \{\\{\frac{(H\sigma_ i-1+(1-\sigma_ i)^H)}{(\sigma_i)^2}\\}_ {i=1}^m, H\}})$, breaking the curse of multiagency.
>
> **Q3**: Thank you for your valuable feedback.
>
> For the first question: As revised in Theorem 1, the robust NE policy gap error $\varepsilon$ in our analysis depends on $C_{\mathrm{r}}^*$, with higher values of $C_{\mathrm{r}}^*$ leading to greater errors.
>
> For the second question: Generally, the coefficient $C_{\mathrm{r}}^*$ cannot be reliably estimated from the existing dataset, making it inherently challenging in the offline RTZMG settings, i.e., the settings considered in this paper to determine the required sample size or provide formal guarantees. Nevertheless, our algorithm does not rely on prior knowledge of this coefficient. Once provided with a batch dataset, the algorithm can be executed, and succeeds when the task becomes feasible. Therefore, our algorithm retains significant practical value.
>
> **Reference**
>
> [1]. Li, Gen, et al. "Settling the sample complexity of model-based offline reinforcement learning." The Annals of Statistics 52.1 (2024): 233-260.
>
> [2]. Blanchet, Jose, et al. "Double pessimism is provably efficient for distributionally robust offline reinforcement learning: Generic algorithm and robust partial coverage." Advances in Neural Information Processing Systems 36 (2024).

---

> > ### Comment · Reviewer_dphg · 2024-11-26
> >
> > Thank you for the detailed response. I appreciate it and decide to raise my score.

---

> > > ### Author Response · Authors · 2024-11-26
> > >
> > > Thank you for your time, expertise, and positive acknowledgment. We are particularly grateful for the additional score you allocated, and truly appreciate your efforts in advancing the quality and impact of our work.

---

### Meta-Review · Area_Chair_JNTd · 2024-12-21

**Metareview:**

This paper introduces RTZ-VI-LCB for robust MARL in two-player zero-sum Markov games, offering theoretical guarantees and near-optimal sample complexity. However, the contributions rely heavily on existing methods from a few prior works, with limited novelty in algorithmic design or theoretical insights.

**Additional Comments On Reviewer Discussion:**

During the discussion, reviewers raised concerns about limited novelty, scalability to more general settings, and the lack of empirical comparisons with relevant baselines. The authors clarified their contributions, highlighted distinctions from prior work, and addressed scalability partially but did not fully resolve concerns about novelty or practical impact. These limitations were significant in the final decision to reject.

---

### Decision · Program_Chairs · 2025-01-22

Reject